# Functional coupling between TRPV4 channel and TMEM16F modulates human trophoblast fusion

**Yang Zhang[1], Pengfei Liang[1], Liheng Yang[2], Ke Zoe Shan[1], Liping Feng[3,4], Yong Chen[5], Wolfgang Liedtke[5,6,7,8†], Carolyn B Coyne[2,9], Huanghe Yang[1,7]\***

[1]Department of Biochemistry, Duke University Medical Center, Durham, United States; [2]Department of Molecular Genetics and Microbiology, Duke University Medical Center, Durham, United States; [3]Department of Obstetrics and Gynecology, Duke University Medical Centre, Durham, United States; [4]MOE-Shanghai Key Laboratory of Children's Environmental Health, Xinhua Hospital, Shanghai, China; [5]Department of Neurology, Duke University Medical Center, Durham, United States; [6]Department of Anesthesiology, Duke University Medical Center, Durham, United States; [7]Department of Neurobiology, Duke University Medical Center, Durham, United States; [8]College of Dentistry, Department of Molecular Pathobiology, NYU, New York, United States; [9]Duke Human Vaccine Institute, Duke University, Durham, United States

**\*For correspondence:**
huanghe.yang@duke.edu

**Present address:** [†]Global Development Scientific Council, Regeneron Pharmaceuticals, Tarrytown, United States

**Abstract** TMEM16F, a $Ca^{2+}$-activated phospholipid scramblase (CaPLSase), is critical for placental trophoblast syncytialization, HIV infection, and SARS-CoV2-mediated syncytialization, however, how TMEM16F is activated during cell fusion is unclear. Here, using trophoblasts as a model for cell fusion, we demonstrate that $Ca^{2+}$ influx through the $Ca^{2+}$ permeable transient receptor potential vanilloid channel TRPV4 is critical for TMEM16F activation and plays a role in subsequent human trophoblast fusion. GSK1016790A, a TRPV4 specific agonist, robustly activates TMEM16F in trophoblasts. We also show that TRPV4 and TMEM16F are functionally coupled within $Ca^{2+}$ microdomains in a human trophoblast cell line using patch-clamp electrophysiology. Pharmacological inhibition or gene silencing of TRPV4 hinders TMEM16F activation and subsequent trophoblast syncytialization. Our study uncovers the functional expression of TRPV4 and one of the physiological activation mechanisms of TMEM16F in human trophoblasts, thus providing us with novel strategies to regulate CaPLSase activity as a critical checkpoint of physiologically and disease-relevant cell fusion events.

## Editor's evaluation

This study by Yang Zhang and collaborators shows that calcium entry caused by TRPV4 channel activation in human trophoblasts results in TMEM16F lipid scramblase activation and subsequent phosphatidylserine exposure in the outer leaflet of the plasma membrane. Multiple lines of evidence are provided to support the hypothesis that TRPV4 are required for TMEM16F activity in human trophoblasts, and thus that both membrane proteins contribute to trophoblast cell fusion in the human placenta.

## Introduction

Phospholipids are essential building blocks of mammalian cell membranes and are asymmetrically distributed between the inner and outer leaflets (*Bevers and Williamson, 2016*; *Nagata et al., 2016*). TMEM16F is a $Ca^{2+}$-activated phospholipid scramblase (CaPLSase) that catalyzes trans-bilayer movement of the phospholipids down their concentration gradients (*Suzuki et al., 2010*; *Le et al., 2021*). In response to increased intracellular $Ca^{2+}$, phosphatidylserine (PS), an anionic phospholipid that is usually sequestered in the inner membrane leaflet, can rapidly permeate through a hydrophilic pathway of TMEM16F and become exposed on the cell surface (*Le et al., 2019*; *Feng et al., 2019*; *Alvadia et al., 2019*; *Yu et al., 2015*). In platelets, TMEM16F CaPLSase-mediated PS exposure facilitates clotting factor assembly to catalyze thrombin generation and blood coagulation (*Bevers and Williamson, 2016*; *Suzuki et al., 2010*; *Yang et al., 2012*). Consistent with its important role in blood coagulation, TMEM16F deficiency leads to defective thrombin generation and prolonged bleeding in Scott syndrome patients (*Suzuki et al., 2010*; *Boisseau et al., 2018*; *Castoldi et al., 2011*) and in TMEM16F knockout (KO) mice (*Yang et al., 2012*). Besides blood coagulation, TMEM16F-mediated PS exposure has been increasingly reported to be important in cell-cell fusion and viral-cell fusion events including placental trophoblast syncytialization (*Zhang et al., 2020*), SARS-CoV2-mediated syncytialization (*Braga et al., 2021*) and HIV infection (*Zaitseva et al., 2017*). To elucidate the physiological and pathological roles of TMEM16F in cell-cell and viral-cell fusion events, one hitherto missing prerequisite is to understand the cellular mechanisms of TMEM16F activation.

$Ca^{2+}$ is required for TMEM16 activation (*Le et al., 2021*; *Tien et al., 2014*; *Yu et al., 2012*; *Brunner et al., 2014*; *Pedemonte and Galietta, 2014*). $Ca^{2+}$ binds to the highly conserved $Ca^{2+}$ binding sites to activate TMEM16 proteins including TMEM16F CaPLSase and TMEM16A and TMEM16B $Ca^{2+}$-activated $Cl^-$ channels (CaCCs) (*Le et al., 2021*; *Yang et al., 2012*; *Pedemonte and Galietta, 2014*). However, the apparent $Ca^{2+}$ sensitivity of TMEM16F is lower than that of TMEM16A and TMEM16B. Differing from CaCCs that are readily activated by elevated cytosolic $Ca^{2+}$ from either cell surface $Ca^{2+}$ channels or from internal stores (*Yang et al., 2012*; *Cabrita et al., 2017*; *Genovese et al., 2019*; *Jin et al., 2013*; *Shah et al., 2020*; *Zhang et al., 2017*; *Takayama et al., 2014*), TMEM16F activation requires more robust and sustained intracellular $Ca^{2+}$ elevation (*Suzuki et al., 2010*; *Le et al., 2021*; *Yu et al., 2015*; *Yang et al., 2012*; *Grubb et al., 2013*; *Liang and Yang, 2021*; *Shimizu et al., 2013*; *Ye et al., 2018*; *Ye et al., 2019*). $Ca^{2+}$ ionophores are widely used to artificially trigger TMEM16F activation and these pharmacological tools allow us to study the biophysical properties or functional expression of TMEM16F in native cells (*Suzuki et al., 2010*; *Le et al., 2019*; *Yu et al., 2015*; *Zhang et al., 2020*). Nevertheless, how TMEM16F is activated under physiological conditions remains unclear, greatly hindering our understanding of TMEM16F biology and subsequent modulation strategies.

We recently showed that TMEM16F is highly expressed in placenta trophoblasts and plays an essential role in human trophoblast syncytialization both in vitro and in mouse placental development in vivo (*Zhang et al., 2020*). Our TMEM16F KO mice displayed deficient trophoblast syncytialization, signs of intrauterine growth restriction, and partial perinatal lethality. Our animal experiments suggested that the extremely rare incidence of Scott syndrome patients exhibiting only mild bleeding tendencies (*Zwaal et al., 2004*; *Millington-Burgess and Harper, 2020*) could be derived from pregnancy complications due to *TMEM16F* loss-of-function in trophoblasts. To further understand the underlying mechanism of TMEM16F-mediated trophoblast fusion, it is critical to elucidate how TMEM16F is activated under physiological conditions. $Ca^{2+}$-permeable TRPV6 and L-type voltage-gated $Ca^{2+}$ ($Ca_V$) channel expression has been reported in trophoblasts (*Fecher-Trost et al., 2019*; *Meuris et al., 1994*; *Miura et al., 2015*; *Niger et al., 2004*; *Suzuki et al., 2008*). Due to its fast inactivation kinetics upon $Ca^{2+}$ entry (*Singh et al., 2018*), TRPV6 involves in trans-placental $Ca^{2+}$ transport (CaT) instead of sustained $Ca^{2+}$ signaling (*Fecher-Trost et al., 2017*; *van Goor et al., 2017*; *Moreau et al., 2002*; *Stumpf et al., 2008*). L-type Cav channels, on the other hand, require strong membrane depolarization (*Hille, 2001*), which can hardly be achieved in non-excitable trophoblasts. Therefore, these known trophoblast $Ca^{2+}$ channels are unlikely to play a major role in activating TMEM16F during trophoblast fusion.

Here, we report that $Ca^{2+}$ influx through $Ca^{2+}$-permeable TRPV4 channels in human trophoblasts play a key role in activating TMEM16F CaPLSase to modulate trophoblast fusion. Using $Ca^{2+}$ imaging and patch-clamp electrophysiology, we showed that TRPV4 channel is functionally expressed in human trophoblasts. GSK-1016790A (GSK101), a TRPV4 specific agonist, causes

robust Ca²⁺ entry, which triggers TMEM16F CaPLSase activation and subsequent PS exposure. Furthermore, using the different Ca²⁺ chelating kinetics of EGTA and BAPTA, our patch-clamp recording further demonstrated that trophoblast TPRV4 and TMEM16F are functionally coupled within microdomains. We also show that gene silencing of *TRPV4* or pharmacological inhibition of TRPV4 channels hinders trophoblast syncytialization in a human trophoblast cell line and a human trophoblast stem cell-derived organoid model. Thus, manipulating Ca²⁺ channels can serve as an effective approach to control TMEM16F activities, thereby helping to dissect the biological functions of the CaPLSases. We anticipate that these findings will advance our understanding of the cellular activation mechanisms of TMEM16 CaPLSases in different physiological and pathological fusion processes, while also inspiring new therapeutic strategies to target TMEM16F-mediated diseases.

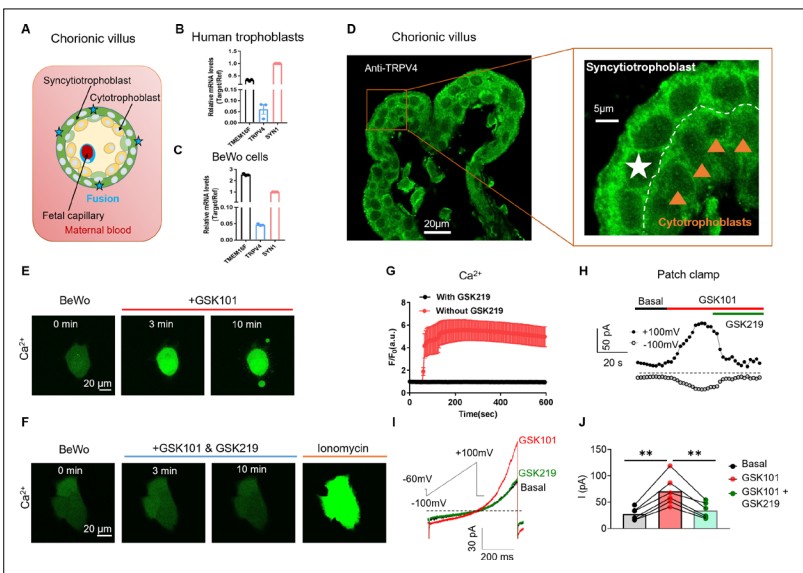

**Figure 1.** TRPV4 is functionally expressed in human trophoblasts. (**A**) Schematic of the first trimester placental villus and trophoblast fusion. (**B, C**) qRT-PCR of *TRPV4* in primary human trophoblasts (**B**) and BeWo cells (**C**). All genes were normalized to GAPDH and then normalized to Syncytin-1 (SYN1) (n=3). (**D**) Representative immunofluorescence of TRPV4 (green) and nuclei (blue) in a human first trimester placenta villus (cross-section). TRPV4 is expressed in both cytotrophoblasts and syncytiotrophoblasts. (**E**) GSK1016790A (GSK101, 20 nM), a specific TRPV4 agonist, triggers robust intracellular Ca²⁺ elevation in BeWo cells. (**F**) GSK2193874 (GSK219, 500 nM), a selective TRPV4 antagonist, abolishes GSK101-induced Ca²⁺ influx through TRPV4 channels in BeWo cells. Ca²⁺ elevation through ionomycin is intact in the presence of GSK219. All fluorescence images in (**D–F**) are the representatives of at least three biological replicates. (**G**) Summary of GSK101 (n=4) and GSK219 (n=5) effects on BeWo cell Ca²⁺ dynamics measured by Ca²⁺ dye (Calbryte 520). (**H**) Time course of outside-out currents elicited in response to 30 nM GSK101 with and without 500 nM GSK219. Current was elicited by a 500-ms ramp voltage protocol from –100 to +100 mV. The holding potential was set at –60 mV. (**I**) Representative current traces in the presence of GSK101 and GSK219+GSK101 from (**H**). (**J**) Quantification of GSK101-induced current in BeWo cells using the same voltage protocol in (**H**). Values represent mean ± SEM and statistics were done using Student's t-test (n=5 for each group, **: p<0.01).

The online version of this article includes the following source data and figure supplement(s) for figure 1:

**Source data 1.** This data is from the Human Protein Atlas V20.1.proteinatlas.org.V20.1.proteinatlas.org.

**Figure supplement 1.** Single-cell RNA sequencing results of *TRPV* channel genes in different cell types from human placentas.

**Figure supplement 2.** Validation of the TRPV4 antibody for immunofluorescence.

**Figure supplement 3.** 4α-phorbol-12, 13-didecanoate (4α-PDD, 20 µM), a TRPV4 agonist, triggers intracellular Ca²⁺ elevation in BeWo cells.

**Figure supplement 4.** Immunofluorescence of TRPV4 in BeWo cells transfected with *TRPV4* siRNAs.

**Figure supplement 5.** siRNA knockdown supports TRPV4 functional expression in BeWo cells.

## Results

### TRPV4 is functionally expressed in human trophoblasts

As a contiguous multinucleated cell layer in the chorionic villi, the placental syncytiotrophoblast establish the most important materno-fetal exchange interface and perform vital barrier, transport, and secretion functions to sustain fetal development and maternal health (*Figure 1A*; *Benirschke and Driscoll, 1967*; *Huppertz and Borges, 2008*). The syncytiotrophoblast is formed and maintained by continuous cell fusion of underlying mononucleated cytotrophoblasts into the syncytial layer during placental development (*Huppertz and Borges, 2008*). Besides the trophoblast CaT encoded by *TRPV6* (*Fecher-Trost et al., 2017*, *van Goor et al., 2017*), the single-cell RNA sequencing results from the Human Protein Atlas show that *TRPV4* is also highly expressed in cytotrophoblasts and syncytiotrophoblasts in the human placenta among TRPV channels (*Figure 1—figure supplement 1*; *Uhlen et al., 2019*; *Sjöstedt et al., 2020*). Consistent with the database, our RT-qPCR experiment also showed that *TRPV4* transcripts are expressed in human primary trophoblasts and BeWo cells, a human choriocarcinoma trophoblast cell line (*Figure 1B–C*). Immunofluorescence with a specific anti-TRPV4 antibody (*Figure 1—figure supplement 2A, B*) further demonstrates that TRPV4 is expressed in the multinucleated syncytiotrophoblasts and single-nucleated cytotrophoblasts in human placenta villi (*Figure 1D*), as well as in BeWo cells (*Figure 1—figure supplement 2C*).

TRPV4 is a $Ca^{2+}$ permeable, non-selective cation channel with less pronounced inactivation than TRPV6 (*Singh et al., 2018*, *Jara-Oseguera et al., 2019*, *Nilius et al., 2004*). Thus, TRPV4 activation might lead to sustained $Ca^{2+}$ elevation in trophoblasts, which would subsequently activate TMEM16F. The availability of selective and potent TRPV4 agonists such as GSK101 and antagonists including GSK-2193874 (GSK219) (*Cheung et al., 2017*; *Thorneloe et al., 2008*) enabled us to explicitly confirm TRPV4's functional expression in human trophoblasts and examine its capability to activate TMEM16F. We found that 20 nM GSK101 robustly increases intracellular $Ca^{2+}$ in BeWo cells, sustaining for no less than 10 min (*Figure 1E and G*). Application of 20 µM 4α-phorbol-12,13-didecanoate (4α-PDD), another TRPV4 agonist, also induced sustained intracellular $Ca^{2+}$ increase in BeWo cells (*Figure 1—figure supplement 3*). On the other hand, 500 nM TRPV4 antagonist GSK219 abolished the intracellular $Ca^{2+}$ elevation induced by 20 nM GSK101, yet it failed to inhibit $Ca^{2+}$ increase from ionomycin (*Figure 1F–G*). Our $Ca^{2+}$ imaging experiments using specific pharmacological tools thus support TRPV4 functional expression in human trophoblasts and that TRPV4 activation can lead to sustained intracellular $Ca^{2+}$ increase.

To further validate our $Ca^{2+}$ imaging results, we used outside-out patch clamp to record GSK101-induced current from BeWo cell membranes. Indeed, GSK101 robustly elicits an outward rectifying cation current that is rapidly inhibited by co-application of GSK219 (*Figure 1H–J*). In addition to pharmacological inhibition, silencing *TRPV4* with siRNAs (*Figure 1—figure supplement 4*) also greatly reduced GSK101-induced $Ca^{2+}$ elevation (*Figure 1—figure supplement 5A, B*) and the non-selective, outward rectifying current (*Figure 1—figure supplement 5C, E*). Taken together, our $Ca^{2+}$ imaging and patch-clamp results established that $Ca^{2+}$ permeable TRPV4 is functionally expressed in human trophoblasts and that TRPV4 activation can lead to sustained $Ca^{2+}$ elevation in these cells.

### $Ca^{2+}$ entry through TRPV4 activates TMEM16F in human trophoblasts in vitro

Using TMEM16F's dual function as a $Ca^{2+}$-activated (i) phospholipid scramblase and (ii) a non-selective ion channel, we applied a fluorescence imaging-based CaPLSase assay (*Le et al., 2019*; *Yu et al., 2015*; *Zhang et al., 2020*) and patch-clamp technique to investigate whether TRPV4-mediated $Ca^{2+}$ influx activates endogenous TMEM16F in human trophoblasts. Upon GSK101 application, $Ca^{2+}$ increased in both BeWo cells (*Figure 2A and C*) and primary human term placental trophoblasts (*Figure 2—figure supplement 1*), followed by phospholipid scrambling as evidenced by robust cell surface accumulation of the fluorescently tagged PS sensor Annexin V (AnV) (*Figure 2A and D*, *Video 1*). In addition, the plasma membrane was dramatically remodeled and released numerous PS positive microparticles, which is consistent with the essential role of TMEM16F in mediating microparticle release in platelets (*Fujii et al., 2015*). In stark contrast, TMEM16F deficient BeWo (KO) cells did not exhibit CaPLSase activity, despite robust $Ca^{2+}$ elevation in response to GSK101 application (*Figure 2B and C–D*, *Video 2*). Neither membrane remodeling nor microparticle release occurred in the TMEM16F KO

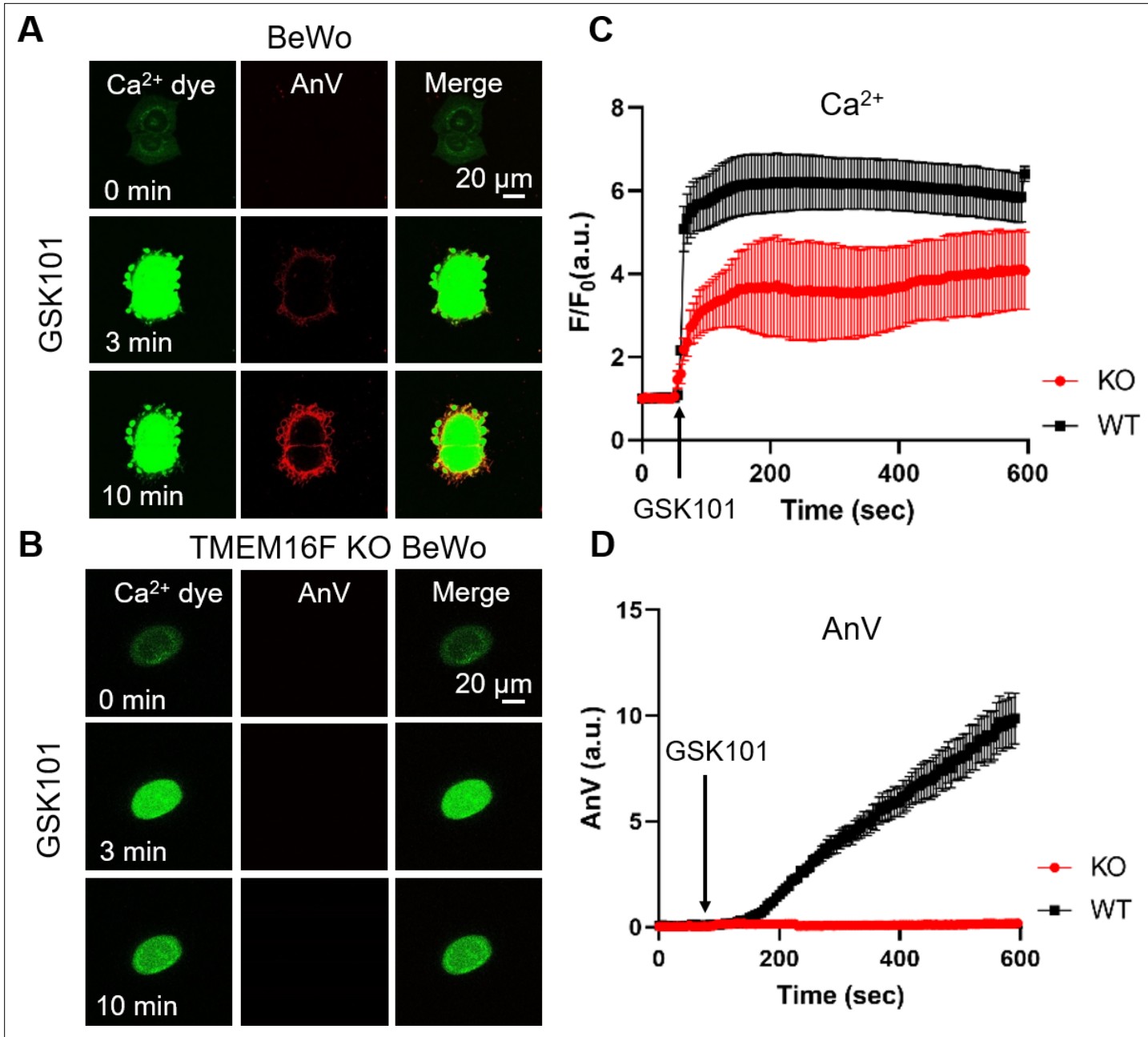

**Figure 2.** $Ca^{2+}$ influx through TRPV4 activates TMEM16F scramblase in BeWo cells. (**A**) 20 nM GSK101 triggers $Ca^{2+}$ influx and PS exposure (labeled by AnV) in wild-type (WT) BeWo cells. (**B**) 20 nM GSK101 induces $Ca^{2+}$ influx, but fails to trigger PS exposure in TMEM16F knockout (KO) BeWo cells. $Ca^{2+}$ dye (Calbryte 520, green) and fluorescently tagged AnV (AnV-CF594, red) were used to monitor the dynamics of intracellular $Ca^{2+}$ and PS externalization, respectively. (**C–D**) Time course of GSK101-triggered $Ca^{2+}$ influx (**C**) and PS exposure (**D**) in BeWo WT (n=5) and TMEM16F KO cells (n=5). AnV, Annexin V.

The online version of this article includes the following source data and figure supplement(s) for figure 2:

**Source data 1.** Source data for *Figure 2* and *Figure 2—figure supplement 1B*.

**Figure supplement 1.** 20 nM GSK101 triggers $Ca^{2+}$ increase and subsequent CaPLSase activities in primary human placental trophoblasts.

BeWo cells. Our fluorescence imaging experiments thus indicate that TRPV4 activation can robustly activate TMEM16F CaPLSases in human trophoblasts.

We also used patch clamp to monitor whole-cell current in BeWo cells in response to GSK101 stimulation. GSK101 rapidly triggered outward-rectifying TRPV4 current that plateaus in both wild-type (WT) and TMEM16F KO BeWo cells (*Figure 3A–C*). A much larger voltage- and time-dependent

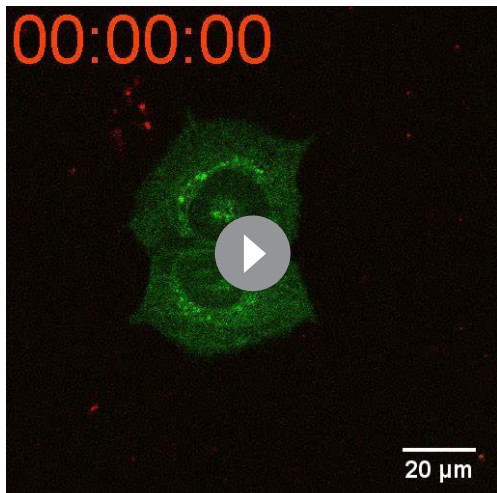

**Video 1.** Time lapse video showing wild-type BeWo cells in response to 20 nM GSK101 stimulation. $Ca^{2+}$ dye (Calbryte 520, green) and fluorescently tagged AnV (AnV-CF594, red) were used to monitor the dynamics of intracellular $Ca^{2+}$ and PS externalization, respectively. Related to Figure 2A. AnV, Annexin V.

https://elifesciences.org/articles/78840/figures#video1

conductance developed after a delay of 6–8 min. Distinct from the initial TRPV4 current, a larger voltage- and time-dependent conductance developed after a 6- to 8-min delay (*Figure 3A and D & E*), showing slow activation and deactivation kinetics (*Figure 3B–C*). The biophysical characteristics of the second conductance closely resemble TMEM16F current recorded in heterologous expression systems (*Grubb et al., 2013*; *Liang and Yang, 2021*; *Shimizu et al., 2013*) and endogenous TMEM16F current recorded in WT BeWo cells under whole-cell (*Figure 3—figure supplement 1A, C*) or outside-out configurations (*Figure 3—figure supplement 1D-F*). TMEM16F KO BeWo cells lack $Ca^{2+}$-, voltage- and time-dependent TMEM16F conductance (*Figure 3—figure supplement 1*); and GSK101 only triggers small, outward-rectifying TRPV4 conductance (*Figure 3*). It is worth noting that, for reasons unclear, there is always a long delay to activate TMEM16F CaPLSase and channel activities in the whole-cell configuration (*Le et al., 2021*; *Yu et al., 2015*; *Grubb et al., 2013*; *Shimizu et al., 2013*; *Lin et al., 2018*). Using whole-cell patch clamp-lipid scrambling fluorometry (PCLSF) to monitor both ionic current and PS exposure (*Yu et al., 2015*; *Liang and Yang, 2021*), we confirm that TMEM16F-mediated ion and phospholipid permeation happen almost simultaneously 7–10 min after GSK101 application (*Figure 3—figure supplement 2*). Our imaging and patch-clamp experiments therefore demonstrate that strong stimulation of TRPV4 activation can robustly activate TMEM16F channels in human trophoblasts.

To further support this conclusion, we used *TRPV4* siRNA and GSK219 to examine if suppressing *TRPV4* expression or TRPV4 activation prevents GSK101-induced TMEM16F activation. GSK219 indeed completely inhibits GSK101-induced $Ca^{2+}$ elevation, eliminating TMEM16F-CaPLSase activation (*Figure 4—figure supplement 1*). Similarly, siRNA knockdown of *TRPV4* not only greatly suppresses the endogenous TRPV4 current and TRPV4-mediated $Ca^{2+}$ influx in BeWo cells (*Figure 4A and C–E* and *Figure 1—figure supplement 5*), but also prevents TMEM16F-mediated PS exposure (*Figure 4A–B*) and ion permeation (*Figure 4C–E*). We still observed robust ionomycin-induced phospholipid scrambling and TMEM16F current in *TRPV4* knockdown BeWo cells (*Figure 4—figure supplement 2*), suggesting that TMEM16F CaPLSases are not affected by *TRPV4* knockdown. Our CaPLSase imaging and patch-clamp experiments thus demonstrate that TRPV4, a previously underappreciated $Ca^{2+}$-permeable channel in human trophoblasts, can serve as an upstream $Ca^{2+}$-source for trophoblast TMEM16F activation.

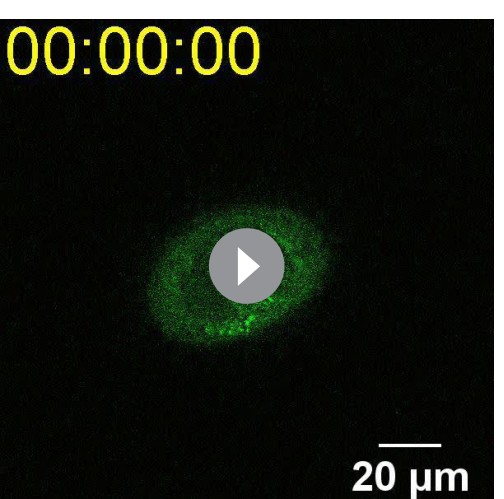

**Video 2.** Time lapse video showing TMEM16F knockout (KO) BeWo cells in response to 20 nM GSK101 stimulation. $Ca^{2+}$ dye (Calbryte 520, green) and fluorescently tagged AnV (AnV-CF594, red) were used to monitor the dynamics of intracellular $Ca^{2+}$ and PS externalization, respectively. Related to Figure 2B. AnV, Annexin V.

https://elifesciences.org/articles/78840/figures#video2

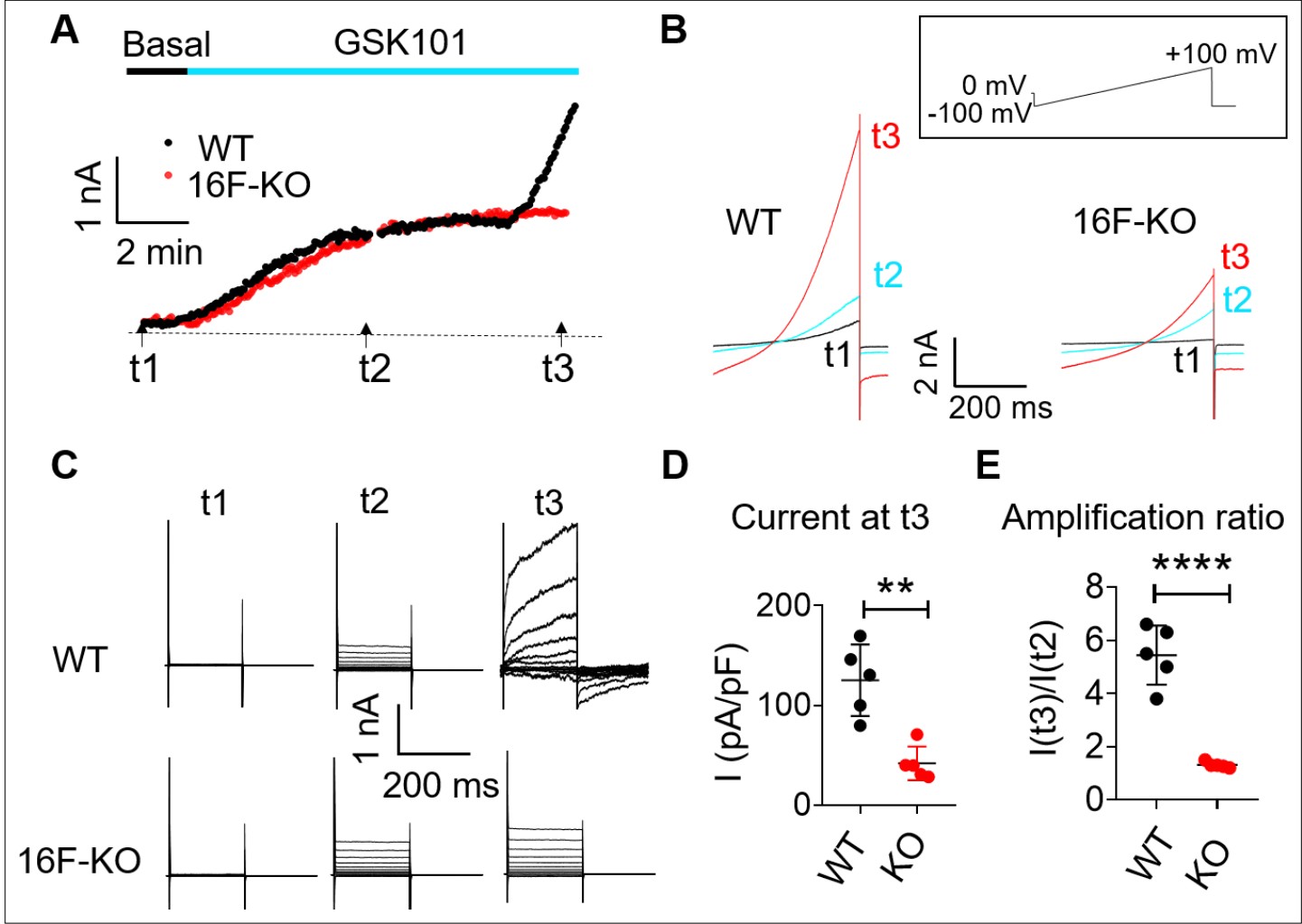

**Figure 3.** TRPV4 activation elicits TMEM16F current in BeWo cells. (**A, B**) Time course of whole-cell currents in response to 30 nM GSK101 stimulation in WT and TMEM16F-KO BeWo cells. The currents were elicited with the ramp protocol shown in (**B**) (top). (**A**) The current amplitudes at +100 mV were plotted every 5 s. (**B**) Representative currents at three different time points as shown in (**A**). (**C**) Representative current traces elicited by a voltage step protocol (200 ms) from −100 to +140 mV at three different time points t1, t2, and t3 as indicated in (**A**). (**D**) Statistical analysis of current density (+100 mV) at t3 in WT and TMEM16F-KO BeWo cells. The current was elicited by the voltage steps shown in (**C**). Values represent mean ± SEM and statistics were done using Student's t-test (n=5 for each group, **: p<0.01). (**E**) Statistical analysis of amplification ratio (current amplitude ratio at t3 and t2 in (**C**)) in WT and TMEM16F-KO BeWo cells. Values represent mean ± SEM and statistics were done using Student's t-test (n=5 for each group, ****: p<0.0001). WT, wild-type.

The online version of this article includes the following source data and figure supplement(s) for figure 3:

**Source data 1.** Source data for *Figure 3* and *Figure 3—figure supplement 1B,C,E,F*.

**Figure supplement 1.** $Ca^{2+}$-activated current in BeWo WT and TMEM16F KO BeWo cells.

**Figure supplement 2.** Simultaneous monitoring of GSK101-triggered channel and lipid scramblase activities in BeWo cells.

## TRPV4 and TMEM16F are functionally coupled within microdomains

To overcome TMEM16F's relatively low $Ca^{2+}$ sensitivity (*Yu et al., 2015*; *Yang et al., 2012*; *Grubb et al., 2013*), its $Ca^{2+}$ sources must be nearby. Unfortunately, the validated antibodies for TRPV4 (*Figure 1—figure supplement 2*) and TMEM16F (*Zhang et al., 2020*) are incompatible, which prevents us from using standard biochemical or imaging analyses to demonstrate their spatial relationship under native conditions. Instead, we used a quantitative patch-clamp approach to estimate the proximity between endogenous TRPV4 and TMEM16F in trophoblasts, taking advantage of EGTA and BAPTA's differential $Ca^{2+}$ buffering kinetics to distinguish between TRPV4 $Ca^{2+}$ micro-domains and nano-domains (*Augustine et al., 2003*; *Dargan and Parker, 2003*; *Fakler and Adelman, 2008*;

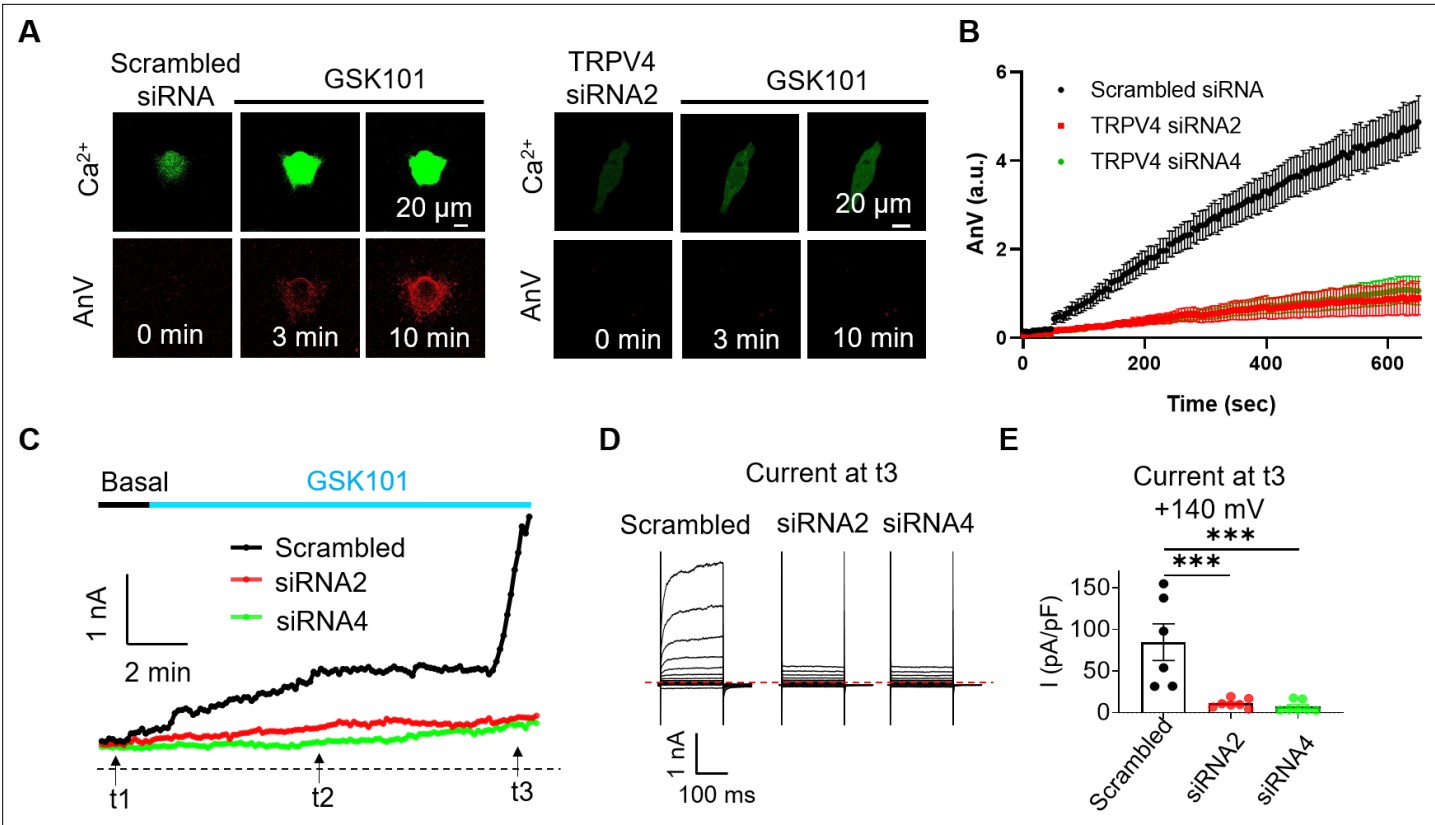

**Figure 4.** siRNA knockdown of *TRPV4* abolishes GSK101-induced Ca²⁺ influx and subsequent TMEM16F CaPLSase activation in BeWo cells. (**A**) Representative images of Ca²⁺ and AnV in scrambled siRNA and *TRPV4* siRNA2 treated BeWo cells in response to 20 nM GSK101 stimulation. Fluorescently tagged AnV labels exposed PS, serving as an indicator of CaPLSase activity. (**B**) Time course of CaPLSase activities for scrambled siRNA (n=50), *TRPV4* siRNA2 (n=55), and siRNA4 (n=51) treated BeWo cells. (**C**) Time course of whole-cell currents elicited in response to 30 nM GSK101 in scrambled siRNA or *TRPV4* siRNAs treated BeWo cells. The currents were elicited with a ramp protocol from –100 to +100 mV and plotted every 5 s at +100 mV. (**D**) Representative current traces elicited by a voltage step protocol (200 ms) from –100 to +140 mV at three different time points t1, t2, and t3 as indicated in (**C**). (**E**) Statistical analysis of current density at t3 (+140 mV) in WT and *TRPV4*-siRNA knockdown BeWo cells. Current densities after scrambled siRNA, *TRPV4*-siRNA2, and *TRPV4*-siRNA4 treatment are 84.72±21.97, 11.12±2.03, and 7.32±2.53 pA/pF, respectively. Values represent mean ± SEM and statistics were done using Student's t-test (n=6 for scrambled group and n=7 for TRPV4 siRNA groups, ***: p<0.001). AnV, Annexin V; WT, wild-type.

The online version of this article includes the following source data and figure supplement(s) for figure 4:

**Source data 1.** Source data for *Figure 4* and *Figure 4—figure supplement 1*.

**Figure supplement 1.** Pharmacological inhibition of TRPV4 abolishes GSK101-induced Ca²⁺ influx and subsequent TMEM16F activation in BeWo cells.

**Figure supplement 2.** TMEM16F CaPLSase and channel activity in BeWo cells is not affected by *TRPV4* knockdown.

*Neher, 1998*; *Eggermann et al., 2011*; *Adler et al., 1991*). Because we and others have shown that TMEM16F ionic current is instantaneously elicited by Ca²⁺ without delay in excised inside-out (*Yang et al., 2012*; *Lin et al., 2018*), we conducted outside-out patch recording to (i) enable extracellular application of GSK101 and, (ii) avoid complications derived from the long delay of TMEM16F activation in the whole-cell configuration (*Figure 3A* and *Figure 3—figure supplement 2*; *Yu et al., 2015*; *Grubb et al., 2013*; *Shimizu et al., 2013*; *Lin et al., 2018*).

In the presence of GSK101, we varied the duration of a –100 mV pre-pulse to facilitate Ca²⁺ influx through TRPV4 channels (*Figure 5A*). 30 nM GSK101 elicited a small outward-rectifying TRPV4 current in the absence of extracellular Ca²⁺ (*Figure 5B*, #1) or without TMEM16F expression (*Figure 5B*, #5). As this current depends on GSK101 but not on the duration of the pre-pulse (*Figure 5—figure supplement 1*), it is largely mediated by TRPV4. Under 2.5 mM extracellular Ca²⁺ and in the presence of 0.2 mM intracellular EGTA from the electrode, GSK101 rapidly elicited a large, time-dependent current in WT (*Figure 5B–C*, #2) but not TMEM16F-KO BeWo cells (*Figure 5B*, #5), implying that TMEM16F is responsible for the GSK101-induced time-dependent current. Interestingly, TMEM16F

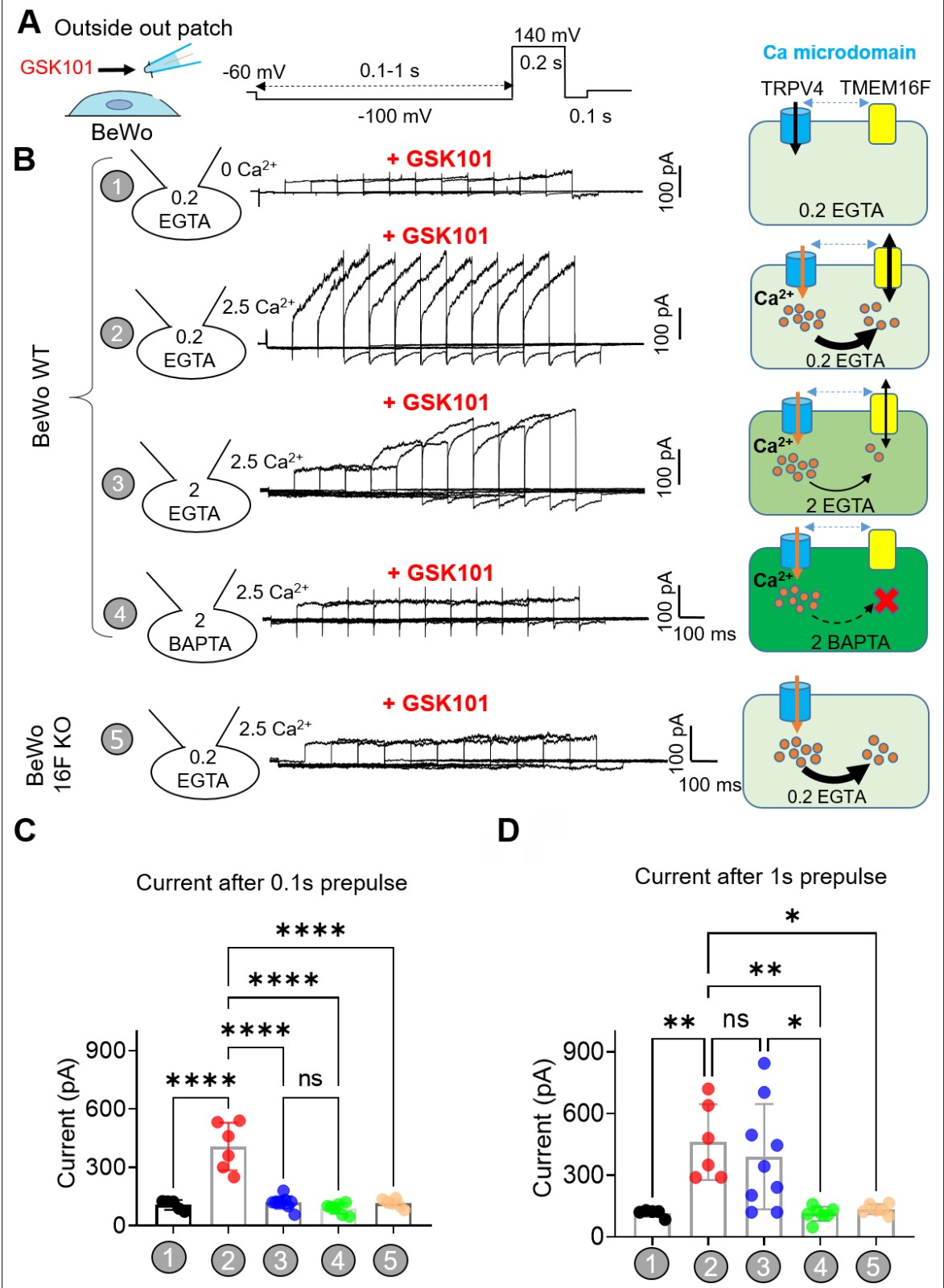

**Figure 5.** TRPV4 and TMEM16F are functionally coupled within microdomain in BeWo cells. (**A**) Outside-out patch configuration and voltage protocols used to demonstrate TRPV4-TMEM16F coupling. The holding potential was set at –60 mV. (**A**) –100 mV pre-pulse with varied length from 0.1 to 1 s with 0.1-s increment was applied along with perfusion of 30 nM GSK101 to induce Ca²⁺ influx. Following the pre-pulse, a depolarized pulse with 0.2 s duration and 140 mV amplitude was applied to record TMEM16F current. (**B**) Left: Representative outside-out patch recordings from wild-type (WT)

*Figure 5 continued*

BeWo cells under different conditions: (1) intracellular 0.2 mM EGTA, extracellular 0 $Ca^{2+}$ with 30 nM GSK101 (n=5); (2) intracellular 0.2 mM EGTA, extracellular 2.5 mM $Ca^{2+}$ with 30 nM GSK101 (n=6); (3) intracellular 2 mM EGTA, extracellular 2.5 mM $Ca^{2+}$ with 30 nM GSK101 (n=9); (4) intracellular 2 mM BAPTA, extracellular 2.5 mM $Ca^{2+}$ with 30 nM GSK101 (n=7); and (5) Intracellular 0.2 mM EGTA, extracellular 2.5 mM $Ca^{2+}$ with 30 nM GSK101 for TMEM16F-KO BeWo cells (n=6). Right: Diagrams demonstrating TRPV4-TMEM16F coupling under each condition on the left. The intensity of green color depicts $Ca^{2+}$ chelating capacity and kinetics with BAPTA as the most efficient $Ca^{2+}$ chalator. (**C, D**) Quantification of peak current amplitudes at +140 mV after 0.1 s pre-pulse (**C**) and 1 s pre-pulse (**D**). Values represent mean ± SEM and statistics were done using Student's t-test (****: $p<0.0001$, **: $p<0.01$, *: $p<0.05$, ns: not significant).

The online version of this article includes the following source data and figure supplement(s) for figure 5:

**Source data 1.** Source data for *Figure 5* and *Figure 5—figure supplement 1B-C*.

**Figure supplement 1.** Lack of TRPV4-TMEM16F coupling in BeWo trophoblast cells in the absence of GSK101.

current amplitudes remain stable regardless of pre-pulse duration, suggesting that $Ca^{2+}$ influx through TRPV4 can activate nearby TMEM16F channels and then rapidly diffuse into the bulk pipette solution without obvious accumulation (*Figure 5B*, #2). When the intracellular EGTA concentration was increased to 2 mM, TMEM16F current after GSK101 stimulation was drastically delayed, suggesting that a higher concentration of EGTA in the electrode efficiently buffers $Ca^{2+}$ influx through TRPV4 channels, hindering TMEM16F activation (*Figure 5B–C*, #3). When the –100 mV pre-pulse was prolonged, more $Ca^{2+}$ entered through TRPV4, overcoming EGTA's buffer capacity within $Ca^{2+}$ microdomains, ultimately activating nearby TMEM16F (*Figure 5B and D*, #3). In stark contrast, 2 mM BAPTA, which has much faster $Ca^{2+}$ chelating kinetics (*Dargan and Parker, 2003*; *Fakler and Adelman, 2008*; *Neher, 1998*), completely ablates TMEM16F current development; only TRPV4 current is observed under this condition (*Figure 5B–D*, #4). The effectiveness of BAPTA but not EGTA to abolish GSK101-induced TMEM16F channel activation demonstrates that TMEM16F and TRPV4 function together in shared microdomain.

## TRPV4-TMEM16F coupling enables spatiotemporal CaPLSase activity

TRPV4-TMEM16F microdomain coupling suggests that $Ca^{2+}$ influx through TRPV4 can efficiently activate TMEM16F CaPLSases without requiring sustained global $Ca^{2+}$ increase. To test this hypothesis,

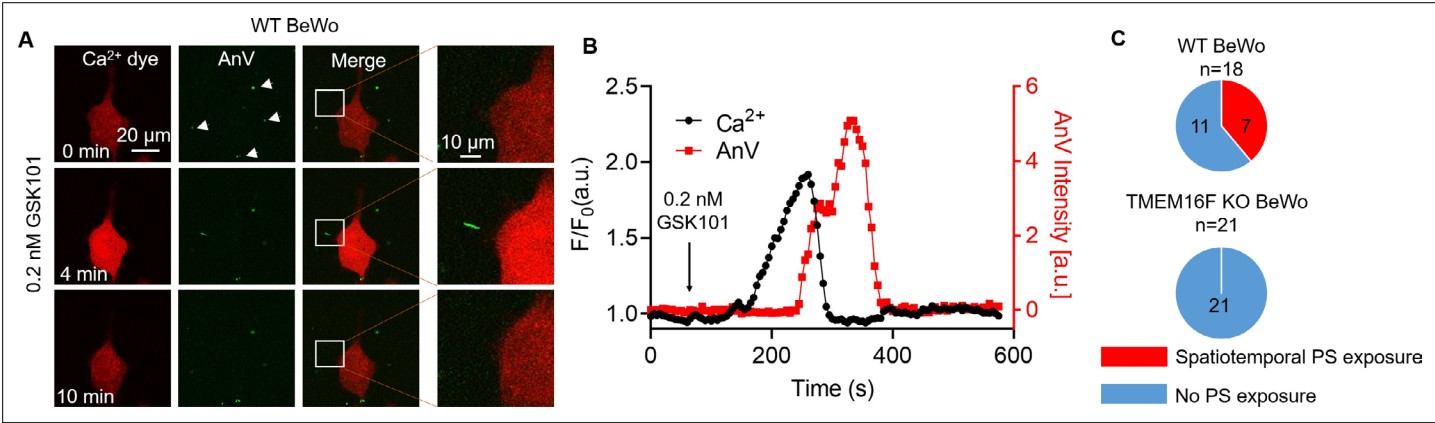

**Figure 6.** Transient $Ca^{2+}$ influx through TRPV4 induces local PS exposure in BeWo cells. (**A**) Representative images of $Ca^{2+}$ and PS exposure from the BeWo cells stimulated with low concentration of GSK101 (0.2 nM). $Ca^{2+}$ dye (Calbryte 594) and fluorescently tagged AnV proteins (AnV-CF488) were used to measure the dynamics of intracellular $Ca^{2+}$ and PS externalization, respectively. The white box highlights transient and reversible PS exposure in a membrane process in response to TRPV4 stimulation. The arrow heads label the PS positive debris in the cell culture, which existed before TRPV4 stimulation. All fluorescence images are the representatives of at least three biological replicates. (**B**) The dynamics of global $Ca^{2+}$ (black) and local AnV signal (red) upon 0.2 nM GSK101 stimulation in WT BeWo cells. (**C**) Quantification of WT (7/18) and TMEM16F KO (0/21) BeWo cells that showed local PS exposure in response to 0.2 nM GSK101. AnV, Annexin V; WT, wild-type.

The online version of this article includes the following source data and figure supplement(s) for figure 6:

**Source data 1.** Source data for *Figure 6* and *Figure 6—figure supplement 1B*.

**Figure supplement 1.** Simultaneous imaging of $Ca^{2+}$ increase and phospholipid scrambling in TMEM16F knockout (KO) BeWo cells in response to low concentration GSK101.

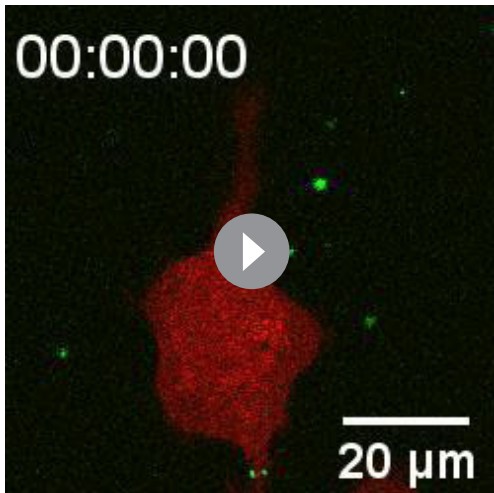

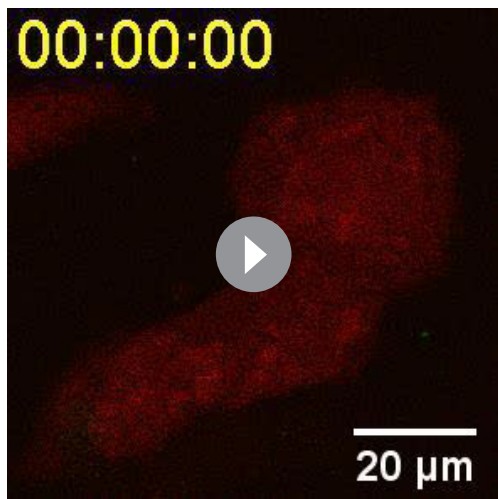

**Video 3.** Time lapse video showing wild-type BeWo cells in response to 0.2 nM GSK101 stimulation. Ca²⁺ dye (Calbryte 594, red) and fluorescently tagged AnV (AnV-CF488, green) were used to monitor the dynamics of intracellular Ca²⁺ and PS externalization, respectively. Related to Figure 6A. AnV, Annexin V.

https://elifesciences.org/articles/78840/figures#video3

**Video 4.** Time lapse video showing TMEM16F knockout (KO) BeWo cells in response to 0.2 nM GSK101 stimulation. Ca²⁺ dye (Calbryte 594, red) and fluorescently tagged AnV (AnV-CF488, green) were used to monitor the dynamics of intracellular Ca²⁺ and PS externalization, respectively. Related to Figure 6—figure supplement 1. AnV, Annexin V.

https://elifesciences.org/articles/78840/figures#video4

we applied an extremely low concentration of GSK101 to BeWo cells. We found that 0.2 nM GSK101 triggered a mild global increase of Ca²⁺, which rapidly decayed. Following the Ca²⁺ transient, AnV fluorescence accumulated on a small membrane process and then rapidly disappeared (*Figure 6A–B* and *Video 3*). As opposed to strong 20–30 nM GSK101 induced irreversible PS exposure (*Figure 2* and *Video 1*), 0.2 nM GSK101 induced local CaPLSase activity that can be fully reversible (*Figure 6*), reflecting the synergistic action of CaPLSases and lipid flippases in maintaining membrane phospholipid asymmetry under physiological conditions (*Nagata et al., 2016*). In addition, mild TRPV4 stimulation-induced CaPLSase activity did not result in obvious membrane remodeling and microparticle release, which are usually evident after strong CaPL-Sase activation (*Figure 2* and *Video 1*). Comparing with global PS exposure in response to 20 nM GSK101 (*Figure 2A, C and D* and *Figure 2—figure supplement 1*), 0.2 nM GSK101 triggered local PS exposure in 7 out of 18 WT BeWo cells tested (*Figure 6C*). In stark contrast, 0.2 nM GSK101 did not stimulate any local CaPLSase activity in TMEM16F KO BeWo cells, albeit normal transient increase of global Ca²⁺ as WT BeWo cells (*Figure 6C*, *Figure 6—figure supplement 1* and *Video 4*). It is worth noting that no local and transient PS exposure was observed in the *TRPV4* siRNA-treated BeWo cells even in the presence of saturating 20 nM GSK101 (*Figure 1—figure supplement 5A, B*), albeit weak increase of intracellular Ca²⁺ through the residual TRPV4 channels. Our experiment thus demonstrates that the transient opening of TRPV4 is sufficient to activate TMEM16F CaPLSase in some subcellular locations. TRPV4-TMEM16F coupling likely plays a key role in spatiotemporal activation of TMEM16F CaPLSase in trophoblast under physiological conditions. Our experiment also shows that TMEM16F CaPLSase can be activated under physiologically relevant Ca²⁺ concentrations without requiring extremely high Ca²⁺ concentration of 20–200 μM as estimated by whole-cell patch-clamp studies.

## Inhibiting TRPV4 hinders human trophoblast fusion in vitro

We recently showed that TMEM16F plays an indispensable role in controlling trophoblast fusion (*Zhang et al., 2020*). To elucidate a physiological function of TRPV4-TMEM16F coupling in trophoblasts, we examined the impacts of TRPV4 inhibition on forskolin-induced BeWo cell fusion in vitro. Consistent with the critical role of TRPV4 in mediating Ca²⁺ influx and TMEM16F-mediated PS externalization, the fusion index (FI) of BeWo cells was significantly decreased when cells were exposed to the TRPV4 inhibitor, GSK219 (*Figure 7A–B*). Furthermore, *TRPV4* siRNA knockdown also significantly

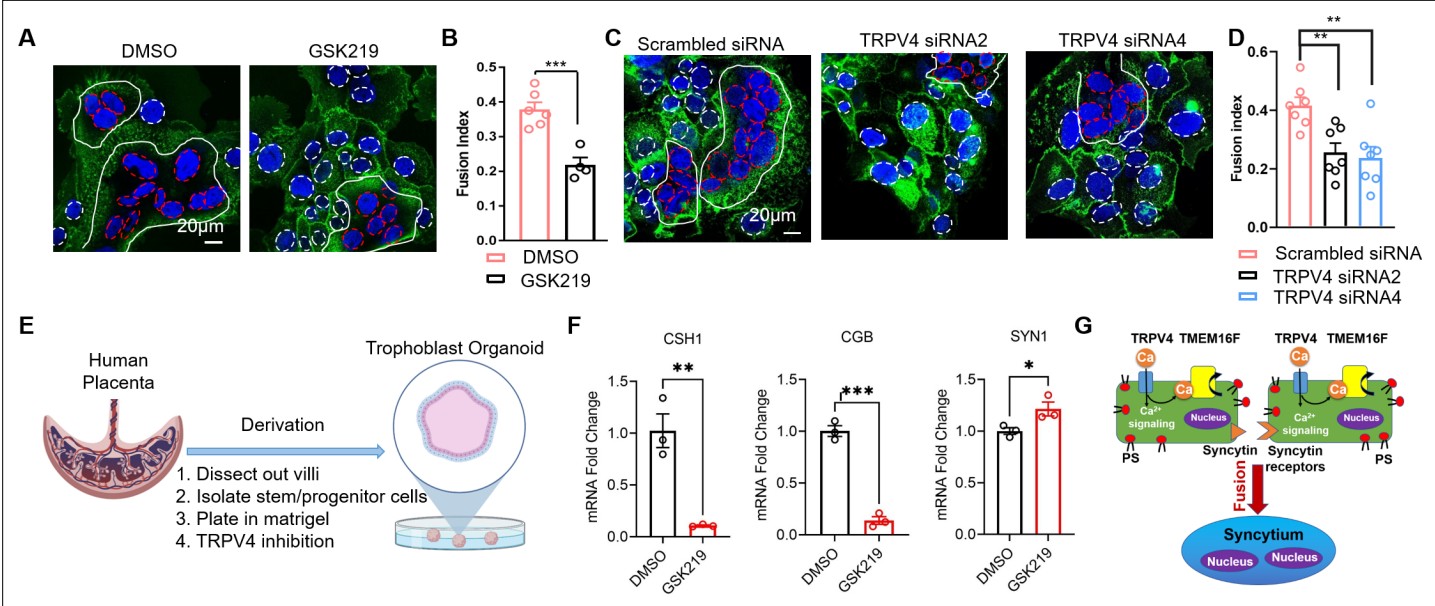

**Figure 7.** TRPV4 inhibition hinders human trophoblast fusion in vitro. (**A**) Representative images of control and GSK219 treated BeWo cells after 48-hr forskolin treatment. Nuclei and membranes were labeled with Hoechst (blue) and Di-8-ANEPPS (green), respectively. (**B**) GSK219 inhibits forskolin-induced BeWo cell fusion. Unpaired two-sided Student's t-test. ***: p<0.001. Error bars indicate ± SEM. Each dot represents the average of fusion indexes (FIs) of six random fields from one cover glass. (**C**) Representative images of control (scrambled siRNA) and *TRPV4* knockdown (*TRPV4* siRNA 2 and 4) BeWo cells after 48-hr forskolin treatment. (**D**) *TRPV4* siRNA knockdown inhibits forskolin-induced BeWo cell fusion. Unpaired two-sided Student's t-test. **: p<0.01. Error bars represent ± SEM. Each dot represents the average of FIs of six random fields from one cover glass (see Materials and methods for details). All fluorescence images are the representatives of at least three biological replicates. (**E**) Schematic of 3-D culture of human placenta trophoblast-derived organoid. (**F**) The mRNA changes of placental syncytiotrophoblast marker genes *CSH1, CGB*, and the fusogenic gene *SYNCITIN-1(SYN1)* after TRPV4 inhibition. n=3 for each group, Unpaired two-sided Student's t-test. ***: p<0.001, **: p<0.01. *: p<0.05. Error bars indicate ± SEM. (**G**) A cartoon demonstration of TRPV4-TMEM16F coupling in regulating trophoblast fusion.

The online version of this article includes the following source data and figure supplement(s) for figure 7:

**Source data 1.** This is IF staining results.

**Figure supplement 1.** TRPV4 knockout (KO) placentas do not show obvious defects in trophoblast syncytialization in mice.

hinders BeWo cell fusion (*Figure 7C–D*). Our cell-based fusion measurements suggest that TRPV4-TMEM16F coupling plays a critical role in controlling trophoblast fusion.

Next, we attempted to understand the physiological function of TRPV4-TMEM16F coupling in vivo by examining the histology of the *Trpv4* deficient (KO) mouse placentas. We do not observe obvious defects on the gross morphology of the placentas (*Figure 7—figure supplement 1A, B*). Using MCT1 and MCT2 as markers for the SynT-1 and SynT-2 syncytiotrophoblast layers in the mouse labyrinth (*Figure 7—figure supplement 1C*), no obvious change in syncytiotrophoblast layer integrity was observed between the WT and *Trpv4* KO placentas (*Figure 7—figure supplement 1D, E*). The lack of fusion defect in mouse placentas is likely due to distinct expression profiles of *TRPV4* in human and mouse placentas. According to recent systematic mapping of Trp channel expression in the mouse placenta (*De Clercq et al., 2021*), *Trpv4* is only expressed in the maternal decidua but not the junctional zone nor the labyrinth, where cytotrophoblasts fuse and form the two syncytiotrophoblast layers (*De Clercq et al., 2021*). Interestingly, the species-specific differential expression profiles are not just limited to *TRPV4*. *Trpv6*, encoding a long-established Ca²⁺ transporter in human trophoblasts (*Suzuki et al., 2008*; *Figure 1—figure supplement 1*), is only weakly expressed in the mouse labyrinth and more prominently expressed in the intraplacental yolk sac (*De Clercq et al., 2021*; *Suzuki et al., 2008*). *Tprv2*, on the other hand, is highly expressed in the mouse labyrinth (*De Clercq et al., 2021*), yet only weakly expressed in human cytotrophoblasts and the syncytiotrophoblast according to the Human Protein Atlas (*Figure 1—figure supplement 1*). The species-specific differential expression of TRPV channels is possibly rooted in the fact that the placenta is a recently evolved organ (*Roberts et al., 2016*). Therefore, TRPV4 might play a key role in the human placenta and a more marginal role

in the mouse placenta. Future studies are therefore needed to systematically compare the physiological functions of other TRPV channels in human and murine trophoblasts.

The species-dependent expression of *Trpv4* in the murine placenta prevents us from using mice as an animal model. Therefore, we used a recently developed human trophoblast organoid (TO) ex vivo model (*Yang et al., 2021*) to further examine TRPV4-TMEM16F coupling in human trophoblast fusion (*Figure 7E*). An imaging method to directly quantify FIs in 3D human TOs have yet to be developed. Therefore, we evaluate the fusion efficiency by measuring the expression of the placental syncytiotrophoblast marker genes, *CSH1* and *CGB*, which encode human placental lactogen (hPL) and hormone-specific β-subunit of human chorionic gonadotropin (βhCG), respectively (*Figure 7F*). As both hPL and βhCG are specifically secreted by the syncytiotrophoblast (*Yabe et al., 2016*), these markers have been widely used as the biomarkers of trophoblast fusion (*Pidoux et al., 2007*; *Deglincerti et al., 2016*; *Gaspard et al., 1980*). We found that incubation of human TOs derived from two unique donor tissues with GSK219 abolished *CSH1* and *CGB* expression, indicating that pharmacological inhibition of TRPV4 can hinder human trophoblast fusion in the organoid model. Interestingly, TRPV4 inhibition even resulted in slight upregulation of *SYNCYTIN-1* (*SYN1*) in the human TOs (*Figure 7F*). This suggests that the trophoblast fusion defect in the presence of TRPV4 inhibitor GSK219 is not derived from the reduction of fusogene SYN1 expression. Our experiments using human TOs and BeWo cells thus demonstrate that TRPV4 plays an important role in human trophoblast fusion in vitro.

## Discussion

Using TRPV4-specific pharmacological and gene silencing tools, fluorescence imaging, and patch-clamp recording with $Ca^{2+}$ chelators, we show that TRPV4 and TMEM16F are functionally coupled within local $Ca^{2+}$ microdomains in a human trophoblast cell line and their coupling plays an important role in trophoblast fusion. The spatial proximity between TRPV4 and TMEM16F ensures higher and more sustained $Ca^{2+}$ elevation, which is required for spatiotemporal TMEM16F activation. When TRPV4 is strongly stimulated by 20–30 nM of GSK101, BeWo cells show a sustained global increase of $Ca^{2+}$ and robust $Ca^{2+}$-dependent phospholipid scrambling and membrane remodeling (*Figure 2*). Distinct from delayed TMEM16F channel and scramblase activation under the invasive whole-cell patch-clamp configuration (*Figures 3A and 4C* and *Figure 3—figure supplement 2*), there is a minimal delay in TMEM16F CaPLSase activation when intact BeWo cells are treated with the same concentration of GSK101 (*Figures 2D and 4B* and *Figure 4—figure supplement 2*). This observation suggests that (1) membrane break-in to form the whole-cell configuration likely disrupts the intracellular environment, alters some cellular factors or structures, which subsequently hinder TMEM16F activation; and (2) the previously reported low $Ca^{2+}$ sensitivity (20–200 μM) for TMEM16F channel activation in whole-cell patches may underestimate TMEM16F CaPLSase's $Ca^{2+}$ affinity (*Alvadia et al., 2019*). Consistent with this hypothesis, our observation of spatiotemporal activation of CaPLSases in local membrane processes under 0.2 nM GSK101 stimulation further supports that extremely high, global increase of $Ca^{2+}$ is not required for TMEM16F CaPLSase activation under physiological conditions. Future studies are needed to accurately measure the $Ca^{2+}$ sensitivity of native TMEM16F CaPLSase in different cell types and pinpoint the exact cellular factors and/or structures that hamper TMEM16F activation.

The transient and reversible CaPLSase activity observed under mild TRPV4 stimulation suggests that TMEM16F CaPLSase activity is likely dynamically regulated in a spatiotemporal fashion under physiological conditions. This highlights the importance of further understanding TMEM16F activation mechanisms to decipher its physiological and pathological functions. We anticipate that our study will inspire future investigations to examine if the functional coupling between $Ca^{2+}$ permeable channels and TMEM16F is a generalized mechanism for TMEM16F activation across different cell types. $Ca^{2+}$-permeable channels other than TRPV4 may be identified as new $Ca^{2+}$ sources for TMEM16F activation. TMEM16A and TMEM16B CaCCs are readily activated by $Ca^{2+}$ entry and $Ca^{2+}$ release from internal stores in both excitable and non-excitable cells (*Genovese et al., 2019*; *Jin et al., 2013*; *Shah et al., 2020*; *Zhang et al., 2017*). Future studies are also needed to examine if $Ca^{2+}$ store release is sufficient to transiently activate TMEM16F at the plasma membrane.

As a polymodal temperature, chemical and osmotic sensors in vertebrates, TRPV4 $Ca^{2+}$ permeable channel is widely expressed and has been reported to be important for vascular function, joint biology and disease, nociception, itch, central nervous system regulation of systemic tonicity, neuroprotection, skin barrier, immune and neurosensory function, lung fibrosis, and skeletal integrity (*Rosenbaum*

et al., 2020). TRPV4's importance for reproduction was first demonstrated in an elegant study of the uterus, in which *Ying et al., 2015* demonstrated that TRPV4 in the myometrium modulates uterine tone during pregnancy and contributes to preterm birth (*Ying et al., 2015*). However, TRPV4's expression and physiological function in the human placenta remained unexplored. In our present study, we demonstrate for the first time that TRPV4 is functionally expressed in human trophoblasts and plays an important role in regulating trophoblast $Ca^{2+}$ signaling, activating TMEM16F to subsequently mediate trophoblast fusion (*Figure 7G*).

Our discovery of TRPV4 expression in human trophoblasts also opens a new avenue to further understand TRPV4 in reproductive biology and pregnancy complications. Future studies are needed to understand how TRPV4 is activated under physiological and pathological conditions and whether TRPV4 contributes to trophoblast functions besides activating TMEM16F (*Figure 7G*). As a polymodal sensor, TRPV4 senses temperature, osmolarity, and it can be activated by endogenous signaling molecules including bioactive lipid, such as glycerophospholipids (*Chen et al., 2021*) and endocannabinoids (*Rosenbaum et al., 2020*). It is plausible that TRPV4 may sense physiological cues, and respond to changing concentrations of endogenous lipid mediators, to maintain trophoblasts function. On the other hand, inappropriate activation mechanisms of TRPV4 might contribute pregnancy complications. Moreover, some dominant gain-of-function (GOF) mutations of *TRPV4* are lethal, up to this point, were considered as critically severe skeletal malformation. Our data suggest that GOF of TRPV4 may contribute to placental malfunction, resulting perinatal lethality (*Kang et al., 2012*; *Camacho et al., 2010*), another interesting subject for future studies.

Cell fusion, including cell-cell fusion and viral-cell fusion, requires the synergistic action of multiple fusion machineries and is tightly regulated (*Brukman et al., 2019*; *Whitlock and Chernomordik, 2021*; *Chen and Olson, 2005*). In addition to known fusogens, recent studies from our and others show that TMEM16F CaPLSase plays essential roles in trophoblast fusion (*Zhang et al., 2020*), HIV-host cell fusion (*Zaitseva et al., 2017*), and SARS-CoV2-mediated syncytialization (*Braga et al., 2021*). Although a detailed mechanism for TMEM16F-mediated cell-cell fusion remains to be determined, it is plausible that PS exposure mediated by TMEM16F CaPLSase serves as a 'fuse-me' signal to prime the fusogenic sites or fusogenic synapses to facilitate these cell fusion events. Our current finding that TMEM16F is functionally coupled to TRPV4 channels raises a provocative question of the molecular identities of $Ca^{2+}$ sources that activate TMEM16F in viral diseases where TMEM16F plays a significant role in syncytium-formation, namely HIV infection (*Zaitseva et al., 2017*) and SARS-CoV2-mediated pathological syncytialization (*Braga et al., 2021*). Targeting functional coupling between $Ca^{2+}$-permeable channels (*Kuebler et al., 2020*) and TMEM16F might complement current therapeutic strategies for HIV-AIDS and COVID19.

Taken together, we reveal functional expression of a previously unrecognized $Ca^{2+}$-permeable channel, TRPV4 in human trophoblasts and demonstrate that microdomain-coupled $Ca^{2+}$ entry through TRPV4 directly activates TMEM16F CaPLSase and contributes to trophoblast fusion. Our study thus uncovers a new physiological activation mechanism for TMEM16F, demonstrating one approach to understand TMEM16F's biological functions in wide-ranging physiological and pathological processes, including blood coagulation, bone mineralization, HIV infection, trophoblast fusion, and SARS-CoV2-mediated syncytialization.

## Materials and methods
### Cell lines
The BeWo cell line was a gift from Dr. Sallie Permar at Duke University and was authenticated by the Duke University DNA Analysis Facility. The TMEM16F KO BeWo cell line was generated by sgRNAs targeting exon 2 as described previously (*Zhang et al., 2020*). The HEK293T cells are authenticated by Duke Cell Culture Facility. BeWo cells were cultured in Dulbecco's modified Eagle's medium-Hams F12 (DMEM/F12) medium (Gibco, REF 11320-033) and HEK293T cells were cultured in DMEM (Gibco, REF 11995-065). Both media were supplemented with 10% fetal bovine serum (FBS) (Sigma-Aldrich, Cat. F2442) and 1% penicillin/streptomycin (Gibco, REF 15-140-122) and cells were cultured in a 5% $CO_2$-95% air incubator at 37°C.

## Human placenta tissue and primary cultured human trophoblast cells

Placental tissues were collected under the Institutional Review Board approval (IRB# PRO00014627 and XHEC-C-2018-089). Informed consent was obtained following the IRBs. Placental cytotrophoblast cells from human term placenta were prepared using a modified method of Kliman as described previously (*Zhang et al., 2020*). The cytotrophoblasts were plated in fibronectin coated cover glass and supplied with DMEM/F12 medium, 10% FBS, and 1% penicillin/streptomycin.

## siRNA transfection

BeWo cells were seeded on coverslips coated with poly-L-lysine (Sigma-Aldrich, #P2636). One day after seeding, BeWo cells were transfected with *TRPV4* siRNAs (Horizon, #MQ-004195-00-0002) using Lipofectamine RNAiMAX Transfection Reagent (Invitrogen, #13778075) following the manufacturer's instructions. After siRNA transfection for 24 hr, the medium was changed to fresh DMEM-F12 and the cells were cultured for another 24 hr before confocal imaging.

## TOs culture

TOs lines used in this study were derived as described (*Yang et al., 2021*). Briefly, TOs lines were plated into the Matrigel (Corning 356231) 'domes', then submerged with prewarmed complete growth media as described (*Yang et al., 2021*). TO cultures were maintained in a 37°C humidified incubator with 5% $CO_2$. The medium was renewed every 2–3 days. About 5–7 days after seeding, TOs were collected from Matrigel 'domes', digested in prewarmed Stem Pro Accutase (Gibco, A11105-01) at 37°C for 5–7 min, then mechanically dissociated into small fragments using electronic automatic pipette. If necessary, digested TOs can be further pipetted manually and then re-seeded into fresh Matrigel 'domes' in 24-well tissue culture plates (Corning 3526). Propagation was performed at 1:3–6 splitting ratio once every 5–7 days. During the first 4 days after re-seeding, the complete growth media was supplemented with additional 5 µM Y-27632 (Sigma-Aldrich, Y0503). For TRPV4 inhibition, the dissociated TOs were directly plated onto the top of 20 µl of Matrigel coating in each well of a regular 24-well plate and cultured for 5 days prior to GSK219 treatment in complete growth media. After 96 hr, supernatant media were removed and coat seeding TOs were lysed with lysis buffer from Sigma Total RNA purification kit for total RNA purification. RT-qPCR was performed as described (*Yang et al., 2021*). Primer sequences can be found in *Supplementary file 1*.

## Mice

The TRPV4 deficient mouse line was reported previously (*Liedtke and Friedman, 2003*). PCR genotyping was performed using tail DNA extraction. Mouse handling and usage were carried out in a strict compliance with protocols approved by the Institutional Animal Care and Use Committee at Duke University, in accordance with National Institute of Health guideline.

## Mouse placenta histological analysis

Isoflurane was used for deep anesthesia of mice. Placentas and embryos were freshly collected and fixed in 4% paraformaldehyde (PFA; Electron Microscopy Sciences, Cat. 15710) overnight at 4°C and embedded in paraffin (HistoCore Arcadia H and Arcadia C, Leica Biosystems) and then sectioned at 5 µm by using Leica RM2255 (Leica Biosystems, Buffalo Grove, IL) for histological staining. Immunofluorescence staining against MCT1 (EMD Millipore, #AB1286-I) and MCT4 (Santa Cruz Biotechnology, #sc-376140) were performed on TRPV4 WT and KO placenta by using MCT1 antibody (MilliporeSigma, AB1286-I) and MCT4 antibody (Santa Cruz Biotechnology, Inc, sc-376140). Images were collected using Zeiss 780 inverted confocal microscope.

## Fluorescence imaging of $Ca^{2+}$ and PS exposure

$Ca^{2+}$ dynamics were monitored using a calcium indicator Calbryte 520 AM (AAT Bioquest, #20701). BeWo cells were stained with 1 µM Calbryte 520 AM in DMEM-F12 for 15 min at 37°C and 5% $CO_2$. PS exposure was detected using CF 594-tagged AnV (Biotium, #29011). To monitor $Ca^{2+}$ and PS dynamics before stimulation, Calbryte 520 AM-stained cells were loaded on the cover glass and incubated in AnV (1:175 diluted in Hank's balanced salt solution) and image for 50 s. To stimulate TRPV4 activity, 20 nM or 0.2 nM TRPV4 agonist GSK101 (Sigma-Aldrich, #G0798) was used. To inhibit TRPV channel activities, 500 nM TRPV4 antagonist GSK219 (Sigma-Aldrich, #SML0694) was applied to trophoblasts.

Cells were continually imaged for another 10 min. Zeiss 780 inverted confocal microscope was used to image $Ca^{2+}$ and PS dynamics at a 5-s interval. MATLAB was used to quantify the cytosolic calcium and AnV binding.

## BeWo cell fusion and quantification of FI

After seeding cells for 24 hr, BeWo cells were treated with 30 µM forskolin (Cell Signaling Technology, #3828s) for 48 hr to induce cell fusion, with forskolin-containing media changed every 24 hr. After forskolin treatment, cells were stained with Hoechst (Invitrogen, #H3570, 1:2000) and Di-8-ANEPPS (Biotium, #61012, 2 µM) or Wheat Germ Agglutinin Alexa Fluor-488 Conjugate (Invitrogen, #W11261, 1:1000) for 15 min at 37°C and 5% $CO_2$. For each treatment group, six random fields of view were acquired using Zeiss 780 inverted confocal microscope. Cell fusion is quantified by calculating the FI as described previously (*Zhang et al., 2020*). FI is calculated as:

$$FI = \frac{\sum\limits_{1}^{6} f_i}{6} = \frac{\sum\limits_{1}^{6}(\frac{\sum nf}{N})}{6}$$

where *fi* represents the FI of each field; *nf*, the number of fused nuclei in each field; *N*, the total nuclei number in each field, respectively. To avoid the instances of cell division, cells with two nuclei were not considered as fused cells.

## Immunostaining

The mCherry-TMEM16F expressing BeWo TMEM16F KO cells were generated by transduction of mCherry-TMEM16F containing lentiviral particles as described previously (*Zhang et al., 2020*). Flag-tagged R. norvegicus TRPV4 plasmids (Addgene, #45751) were transiently transfected to HEK293T cells by using X-tremeGENE360 transfection reagent (MilliporeSigma). Medium was replaced with regular culture medium 4 hr after transfection. Ruthenium red (Sigma-Aldrich, #R2751, 10 µM) was add into the culture medium to inhibit TRPV4 activation. Cells were fixed with 1% PFA in phosphate-buffered saline (PBS) for 10 min, permeabilized with 0.1% Triton X-100 in PBS, and blocked with 5% goat serum in PBS for an hour. Coverslips were incubated in primary antibodies, anti-TRPV4 (1:200, Alomone, #ACC-034) and anti-mCherry (1:500, NOVUS, #NBP1-96752) or anti-Flag (1:1000, Sigma-Aldrich, #F1804), at 4°C overnight. Secondary antibodies, Alexa Fluor 488 or Alexa Fluor 640 fluorescence system (Molecular Probes, #35552), were used for fluorescent staining. After staining nuclei with 4',6-diamidino-2-phenylindole (DAPI), coverslips were mounted using ProLong Diamond Anti-fade Mountant (Invitrogen, #P36961) and imaged with Zeiss 780 inverted confocal microscope.

## Electrophysiology

All currents were recorded in either outside-out or whole-cell configurations on BeWo cells using an Axopatch 200B amplifier (Molecular Devices) and the pClamp software package (Molecular Devices). The glass pipettes were pulled from borosilicate capillaries (Sutter Instruments) and fire-polished using a microforge (Narishge) to reach resistance of 2–3 MΩ. The pipette solution contained (in mM): 140 CsCl, 10 HEPES, 1 $MgCl_2$, adjusted to pH 7.2 (CsOH), 0.2 or 2 mM EGTA or 2 mM BAPTA as indicated. The bath solution contained 140 CsCl, 10 HEPES, 0 or 2.5 $CaCl_2$ as indicated, adjusted to pH 7.4 (CsOH). Pharmacological reagents were applied from extracellular side including 30 nM GSK101 (Sigma-Aldrich, #G0798), 500 nM GSK219 (Sigma-Aldrich, #SML0694) as indicated. Procedures for solution application were as employed previously (*Le et al., 2019*; *Liang and Yang, 2021*). Briefly, a perfusion manifold with 100–200 µm tip was packed with eight PE10 tubes. Each tube was under separate valve control (ALA-VM8, ALA Scientific Instruments), and solution was applied from only one PE10 tube at a time onto the excised patches or whole-cell clamped cells. All experiments were at room temperature (22–25°C). All the other chemicals for solution preparation were obtained from Sigma-Aldrich.

## Patch clamp-lipid scrambling fluorometry assay

This assay combined patch-clamp recoding with phospholipid scrambling assay as reported previously (*Yu et al., 2015*; *Liang and Yang, 2021*). Briefly, cells were seeded and transfected on poly-L-lysine (Sigma-Aldrich)-coated coverslips prior to the experiments. Fluorescence-tagged (CF 594) AnV (Annexin V-CF 594, Biotium #29011) was diluted at 0.5 µg ml$^{-1}$ with extracellular recording solutions

(140 mM CsCl, 2.5 mM CaCl$_2$, 10 mM HEPES, 30 nM GSK101, pH 7.4) and then added into one of the perfusion tubes. Glassed pipettes were prepared and filled with internal solution as mentioned previously in the electrophysiology part. After focusing on the cell surface, switching the filter to CF594 (Annexin V signal). Then a whole-cell patch was achieved, following with application of designed voltage protocol at an inter-sweep interval of 5 s. Simultaneously, imaging recordings about AnV signal was initiated also with 5-s interval.

## Statistical analysis

All statistical analyses were performed with Clampfit 10.7 (Molecular Devices), Excel (Microsoft), or Prism software (GraphPad). A two-tailed Student's t-test was used for single comparisons between two groups (paired or unpaired). One-way ANOVA following by Tukey's test was used for multiple comparisons. Data were represented as mean ± standard error of the mean (SEM) unless stated otherwise. Symbols *, **, ***, ****, and ns denote statistical significance corresponding to p-value<0.05, <0.01, <0.001,<0.0001, and no significance, respectively.

## Acknowledgements

The authors appreciate Dr. Hua Pan for technical support and Augustus Lowry for critical reading of the manuscript. This work was supported by NIH-DP2GM126898 grant (to H.Y.).

## Additional information

### Competing interests

Wolfgang Liedtke: WL is a full-time executive employee of Regeneron Pharmaceuticals, Tarrytown NY. This affiliation had no influence on his contribution. Beyond this, no other conflicts of interest exist. The other authors declare that no competing interests exist.

### Funding

| Funder | Grant reference number | Author |
|---|---|---|
| National Institutes of Health | DP2GM126898 | Huanghe Yang |

The funders had no role in study design, data collection and interpretation, or the decision to submit the work for publication.

### Author contributions

Yang Zhang, Conceptualization, Data curation, Formal analysis, Investigation, Methodology, Software, Validation, Visualization, Writing – original draft, Writing – review and editing; Pengfei Liang, Data curation, Formal analysis, Investigation, Methodology, Validation, Writing – original draft, Writing – review and editing; Liheng Yang, Data curation, Formal analysis, Investigation, Methodology, Writing – original draft, Writing – review and editing; Ke Zoe Shan, Data curation, Formal analysis, Investigation, Validation, Visualization, Writing – original draft, Writing – review and editing; Liping Feng, Yong Chen, Resources; Wolfgang Liedtke, Resources, Writing – original draft; Carolyn B Coyne, Formal analysis, Methodology, Resources, Writing – original draft, Writing – review and editing; Huanghe Yang, Conceptualization, Funding acquisition, Project administration, Resources, Supervision, Visualization, Writing – original draft, Writing – review and editing

### Author ORCIDs

Yang Zhang http://orcid.org/0000-0003-3625-9965
Liheng Yang http://orcid.org/0000-0001-6842-086X
Carolyn B Coyne http://orcid.org/0000-0002-1884-6309
Huanghe Yang http://orcid.org/0000-0001-9521-9328

## Ethics

Human subjects: Placental tissues were collected under the Institutional Review Board approval (IRB# PRO00014627 of Duke University and XHEC-C-2018-089 of Xinhua Hospital). Informed consent was obtained following the IRBs.

This study was performed in strict accordance with the recommendations in the Guide for the Care and Use of Laboratory Animals of the National Institutes of Health. All of the animals were handled according to approved institutional animal care and use committee (IACUC) protocols (#A086-21-04) of Duke University.

## Decision letter and Author response

Decision letter https://doi.org/10.7554/eLife.78840.sa1
Author response https://doi.org/10.7554/eLife.78840.sa2

# Additional files

## Supplementary files

• Supplementary file 1. qPCR primer list used in this study.

• MDAR checklist

## Data availability

All study data are included in the article and/or supporting information. All the numerical data can be found in the associated 'Source data' files for each figures. The MATLAB code supporting the present study is available at GitHub https://github.com/YZ299/matlabcode/blob/main/matlabcode.m, (copy archived at swh:1:rev:be9755302543fa49eef5c3137b35a46e687a9853).

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
