## [Editor Report]

This study by Yang Zhang and collaborators shows that calcium entry caused by TRPV4 channel activation in human trophoblasts results in TMEM16F lipid scramblase activation and subsequent phosphatidylserine exposure in the outer leaflet of the plasma membrane. Multiple lines of evidence are provided to support the hypothesis that TRPV4 are required for TMEM16F activity in human trophoblasts, and thus that both membrane proteins contribute to trophoblast cell fusion in the human placenta.

---

## [Decision Letter]

**Decision letter after peer review:**

[Editors’ note: the authors submitted for reconsideration following the decision after peer review. What follows is the decision letter after the first round of review.]

Thank you for submitting the paper "Functional coupling between TRPV4 channel and TMEM16F modulates human trophoblast fusion" for consideration at *eLife*. Your submission been reviewed by three peer reviewers, one of whom is a member of our Board of Reviewing Editors, and the evaluation has been overseen by a Senior Editor. Although the work is of interest, we regret to inform you that the findings at this stage are too preliminary for further consideration at *eLife*.

Specifically, all reviewers agreed that whereas the work addresses a very relevant question, provides novel findings and the data is of high quality, a significant amount of experimental effort would still be required to solidly establish a functional interaction between TMEM16F and TRPV4 channels in human trophoblast cells, and whether this has biological relevance. Based on the discussion between the reviewers, we have put together a list of essential revisions that summarizes the concerns that were raised in review. Although there was considerable interest in the work, in our view addressing the concerns would require more than two months of effort and thus we have decided against inviting revision. We hope you find this evaluation helpful in preparing a draft for submission to another journal. If you do choose to address the essential revisions outline below, we would be willing to reconsider the work as a new submission.

Essential revisions:

1) The biological relevance for the possible interaction between TRPV4 and TMEM16F channels needs to be toned down throughout the manuscript. The authors should provide additional data in support of a biological relevance for their findings. It would be expected that TRPV4-KO mice have phenotypes associated with trophoblast fusion deficiencies if the association between TRPV4 and TMEM16F channels is as relevant as suggested – this should be at least further discussed. The authors could also use GSK101 concentrations that result in cytosolic calcium elevations that closely resemble those that would occur in vivo, or at least provide a discussion justifying the concentration of GSK101 that was used. The authors do not address the question whether the coupling of TRPV4 to TMEM16F is a general phenomenon or specific to trophoblasts.

2) The emphasis on the findings pertaining to human trophoblasts should also be toned down throughout, as most of the experiments were done in a cell line and not in native cells.

3) Group data quantitation should be provided for all relevant figures (Figure 1I and J, Figure 3B, Figure 4A-C, Figure 5, Figure S2, Figure S3A, Figure S3B, Figure S4, Figure S5, Figure S6, Figure S7). The imaging data are strictly qualitative and should be either subject to rigorous quantification with statistical analysis or eliminated. Functional assays are convincing that TRPV4 is expressed in trophoblasts, but the immunofluorescence is ambiguous. Figure 1D does not show TRPV4 convincingly on the membrane. Figure 1E needs a counterstain to show the location of the membrane and quantification of overlap between a membrane marker and TRPV4. The validation of the antibody in Figure S2 should also show staining in untransfected cells. A western blot would also be helpful.

4) It has been previously shown that the scramblase activity and ion conducing pore of TMEM16F are activated by the same ca^2+^ concentrations. The authors should show that the kinetics of the candidate TMEM16F currents occur simultaneously with activation of the scramblase activity.

5) Why is the ca^2+^ influx in KO BeWo cells much smaller than the ca^2+^ influx in wt BeWo cells? These authors have previously argued that TMEM16F is a Ca-permeable non-selective cation channel, and a plausible alternative explanation is that Ca entering via TRPV4 activates Ca influx via TMEM16F and that scrambling follows this.

6) The dynamics of Ca signaling needs further explanation. It is not clear that, in the presence of even low concentrations of EGTA, ca^2+^ through TRPV4 in excised patches could accumulate at the surface of the membrane at the high concentrations required. Indeed, the inward currents in Figure 5, as an example, are so small that, even if all the current were ca^2+^ it seems unlikely that local ca^2+^ at sufficiently high concentrations would occur. Using known information on the binding kinetics of the chelators used, the size of the currents measured, and the permeability of ca^2+^ relative to other ions, the authors could model the expected ca^2+^ kinetics in their patches to demonstrate whether the >20 µM ca^2+^ concentrations would develop as expected. There may be other ways the authors could demonstrate the actual ca^2+^ concentrations achieved on the outside of their patches.

Ca influx is dependent on voltage: at +100mV Ca influx should be negligible because Vm is approaching E(Ca). But, TMEM16F currents seem to be independent of the Ca driving force. TMEM16F currents increase with time in response to voltage steps to +100 mV, despite the fact that Ca influx is decreasing at this voltage. The amplitude of the currents are comparable to the currents elicited by 100 μm Ca in whole-cell patch clamp (compare Figure 4D and Figure S5), so this suggests that [Ca] at the TMEM16F channel remains close to 100 μm even at +100 mV. Furthermore, in Figure 5B panel 2, the amplitude of the current is independent of the duration of the prepulse at -100 mV. Maybe there is sufficient Ca influx at the holding potential to flood the cell, overwhelm native efflux mechanisms, and saturate Ca buffers, but if that is the case, this would invalidate the Ca microdomain concept developed in Figure 5.

7) The ca^2+^-imaging data, using a dye with a KD = 1.2 µM, are overinterpreted. TMEM16F is not active even at 20 µM ca^2+^ – it requires about 200 µM to be active. The authors should be more conservative in how they discuss their data given the affinities of their dye. This argument applies to the EGTA and BAPTA data.

8) The authors characterize the currents they attribute to TRPV4 as not having activation/deactivation kinetics, in contrast to the late-developing currents, which do show such kinetics. The pronounced outward rectification of the currents attributed to TRPV4 mean that they have very pronounced gating kinetics. They are just much faster than those of the late-developing currents. This must be clarified.

9) The authors should provide a control showing that the TRPV4 siRNA does not directly affect expression or activity of TMEM16F channels. This could be done by performing patch-clamp recordings with added calcium to stimulate these channels, and by showing that treatment with a calcium ionophore results in robust Annexin V binding to the outer leaflet in siRNA-treated cells.

10) The authors should further validate their TRPV4-knockdown by showing a decreased channel expression in the plasma membrane with immunofluorescence or western blots.

11) The authors should perform outside-out experiments in the presence of 2.5mM extracellular calcium and 0.2 mM EGTA in the pipette without GSK101 exposure, as in Figure 5, and show that no TMEM16F current buildup occurs after long wait times.

12) The mechanism of Forskolin is not clear. I am not aware that it has been shown that cAMP-dependent processes activate TRPV4 in trophoblasts, although TRPV4 does have consensus PKA phosphorylation sites and there is suggestive evidence in other cell types that TRPV4 is regulated by PKA-dependent phosphorylation. This should be discussed. The authors should also show whether TMEM16F knockdown reduces Forskolin-stimulated Ca signaling. The role of Forskolin on cell fusion should be discussed further – do the authors imply that the exposure of PS facilitates fusion?

13) More information is required to validate that identification of fused cells carries no subjective components, e.g. by showing the classification into fused vs non-fused for all cells in the shown representative images in Figure 6. While the TRPV4 antagonist GSK219 or siRNA reduce trophoblast fusion, they only reduce it by about 40%. The effect of TRPV4 siRNA, while statistically significant, shows a lot of scatter with significant overlap between control and siRNA data points. This suggests that other mechanisms also activate TMEM16F or that TMEM16F is not required for fusion. This should be addressed.

14) The authors must examine experimentally the potential contribution of other sources of calcium in BeWo and trophoblast cells, in particular CaV and TRPV6. Other TRPV channels are not studied because the Human Protein Atlas suggests expression is low, but the Human Protein Atlas is notoriously unreliable. Expression should be measured in the system used in this study. Ca-activated mobilization of internal Ca stores or other channels must also be considered.

15) The findings suggesting that TMEM16F and TRPV4 are co-localized should be toned down. Additional evidence should be provided to conclude that this is the case. This could involve proximity labeling, higher-resolution co-localization using fluorescence microscopy or FRET, or showing a direct interaction between the two channels. The colocalization of TRPV4 and TMEM16F (Figure S6) appears good in the selected field containing one cell, but this needs quantification over a large population of cells. Further, the colocalization is only demonstrated in cells over-expressing TMEM16F, not in native cells.

16) The writing could use improvement to make it flow better. There are missing articles and singular-plural mismatches that make the text hard to follow.

17) The introduction could be improved to make it easier for readers to follow. For those not expert in the field, the discussion of the authors' previous work, which sets up the logic for the present study, is difficult to understand. The last paragraph of the introduction goes well beyond the current manuscript and should be reined in.

*Reviewer #1:*

TMEM16F non-selective channels and lipid scramblases play many biologically relevant roles involving cell fusion, including a function in trophoblast syncytialization. However, it is unclear which pathways are utilized under physiological conditions to supply TMEM16F channels with enough cytosolic calcium to trigger their activity. In the present manuscript, Yang Zhang and collaborators show that primary human trophoblast cells express TRPV4 channels, and that cell treatment with a TRPV4-selective agonist, GSK101, increases cytosolic calcium concentration and phosphatidylserine exposure on the outer leaflet of cells. Using a human choriocarcinoma trophoblast cell line BeWo, the authors investigate the mechanism further. The authors show that the TRPV4 agonist GSK101 increases cytosolic calcium in these cells based on calcium imaging experiments and enhances currents likely mediated by TRPV4 measured in patch-clamp recordings. GSK101 treatment also enhances Annexin V binding to the plasma membrane, suggesting increased scramblase TMEM16F activity, and results in delayed activation of voltage-dependent currents that resemble TMEM16F currents. Both Annexin V exposure and voltage-dependent current buildup are absent in TMEM16F-deficient BeWo cells exposed to GSK101, as well as in cells treated with two different TRPV4 siRNAs. Together these results suggest that calcium entry via GSK101-activated TRPV4 channels stimulates TMEM16F currents and lipid scrambling activity. In outside-out patches from BeWo cells, the authors show that BAPTA in the pipette prevents the buildup of TMEM16F currents when cells are exposed to extracellular calcium and GSK101, whereas the slower chelator EGTA does not. Together with the co-localization of TMEM16F and TRPV4 channels observed using fluorescence microscopy, the latter findings suggest that both proteins are in close proximity in the plasma membrane. Finally, the authors provide evidence suggesting that the presence of TRPV4 enhances Forskolin-mediated cell fusion of BeWo cells, because treatment with a TRPV4 antagonist of with TRPV4-directred siRNA reduces cell fusion.

The possibility of a biologically relevant interaction between TRPV4 and TMEM16F in human trophoblast cells is novel and relevant. However, the biological relevance of the findings is not strongly established by the data provided in the manuscript. First, most of the findings are done in the BeWo cell line, and the experimental findings in actual human trophoblast cells are limited to showing that TRPV4 channels are expressed and that exposure of these cells to GSK101 induces calcium entry and Annexin V binding to the plasma membrane. The extent to which BeWo cells and primary human trophoblast cells are similar is not discussed. Further, a saturating concentration of a synthetic agonist is utilized to activate TRPV4 channels, but it remains unclear whether TRPV4 activity is stimulated under physiological conditions in native trophoblast cells, or whether the amount of calcium entry triggered by GSK101 reflects what occurs under physiological conditions. It remains possible that no biologically relevant connection between TMEM16F and TRPV4 channels exists, and that the findings reflect an abnormally high stimulation of TRPV4 channels leading to non-biological effects. These considerations are not reflected in the way the data and results are presented throughout the manuscript. No data is provided to support that TRPV4 channels play a role in trophoblast syncytialization in vivo.

In addition, the manuscript lacks several important controls as well as group-data quantitation of many key experiments (Figure 1I and J, Figure 3B, Figure 4A-C, Figure 5, Figure S2, Figure S3A, Figure S3B, Figure S4, Figure S5, Figure S6, Figure S7); whereas the time-course of calcium entry and Annexin V binding to WT and TMEM16F KO BeWo cells exposed to GSK101 is shown in Figure 2, no such quantitation is provided for identical experiments with cells treated with TRPV4-siRNA (Figure 4). No data is provided to show whether treatment of cells with TRPV4-siRNA reduces expression or activity of TMEM16F channels in BeWo cells, or to verify that this treatment indeed reduces the presence of TRPV4 channels, e.g. by showing reduced binding of the TRPV4-specific antibody to siRNA-treated cells or a western blot for TRPV4 channels localized at the plasma membrane. No control is provided to show whether exposure of outside-out patches to high extracellular calcium causes any TMEM16F current activation without exposure to GSK101 when long recordings as those in Figure 5 are carried out. As mentioned by the authors, BeWo cells have other calcium-permeable channels, including TRPV6 channels that are generally considered as constitutively active.

In regards to the proximity of TMEM16F and TRPV4 channels establishing functionally relevant calcium microdomains, no conclusive evidence is provided. Both the fluorescence microscopy in Figure S6 (involving an over-expressed TRPV4 construct) and the comparison between fast and slow calcium chelators in Figure 5 constitute indirect evidence, which is not reflected in the authors analysis and discussion of their data throughput the manuscript.

Finally, the results regarding cell fusion do not appear to be very robust. First, cell fusion is induced by Forskolin, which appears to suggest that TRPV4-mediated calcium entry and TMEM16F lipid scrambling are only complementary mechanisms. It is unclear from the discussion in the manuscript whether the activity of these two channels would be expected to promote cell fusion under physiological conditions without any added Forskolin. Second, there seems to be limited validation from the images in Figure 6 to support a robust effect of TRPV4-knock down or inhibition on cell fusion: the criteria for defining fused vs non-fused cells are not very clear and appear to have some degree of subjectivity, and in addition there is lots of spread in the group data and the differences between treatments are not very pronounced.

1) The biological relevance for the possible interaction between TRPV4 and TMEM16F channels needs to be toned down throughout the manuscript. The authors should provide additional data in support of a biological relevance for their findings. It would be expected that TRPV4-KO mice have phenotypes associated with trophoblast fusion deficiencies if the association between TRPV4 and TMEM16F channels is as relevant as suggested. The authors could also use GSK101 concentrations that result in cytosolic calcium elevations that closely resemble those that would occur in vivo, or at least provide a discussion justifying the concentration of GSK101 that was used.

2) The emphasis on the findings pertaining to human trophoblasts should also be toned down throughout, as most of the experiments were done in a cell line and not in native cells.

3) Group data quantitation should be provided for all relevant figures (Figure 1I and J, Figure 3B, Figure 4A-C, Figure 5, Figure S2, Figure S3A, Figure S3B, Figure S4, Figure S5, Figure S6, Figure S7).

4) The authors should provide a control showing that the siRNA does not directly affect expression or activity of TMEM16F channels. This could be done by performing patch-clamp recordings with added calcium to stimulate these channels, and by showing that treatment with a calcium ionophore results in robust Annexin V binding to the outer leaflet in siRNA-treated cells.

5) The authors should further validate their TRPV4-knockdown by showing a decreased channel expression in the plasma membrane with immunofluorescence or western blots.

6) The authors should perform outside-out experiments in the presence of 2.5mM extracellular calcium and 0.2 mM EGTA in the pipette without GSK101 exposure, as in Figure 5, and show that no TMEM16F current buildup occurs after long wait times.

7) The findings suggesting that TMEM16F and TRPV4 are co-localized should be toned down. Additional evidence should be provided to conclude that this is the case. This could involve proximity labeling, higher-resolution co-localization using fluorescence microscopy or FRET, or showing a direct interaction between the two channels.

8) The role of Forskolin, TRPV4 calcium entry and TMEM16F activity on cell fusion should be discussed further. Additionally, more information is required to validate that identification of fused cells carries no subjective components, e.g. by showing the classification into fused vs non-fused for all cells in the shown representative images in Figure 6.

*Reviewer #2:*

Zhang et al. have investigated the question of how the phospholipid scramblase TMEM16F is activated physiologically. in vitro in the lab, TMEM16F is typically activated by the Ca ionophore ionomycin or by patch clamp with high intracellular Ca concentrations, but physiological activation of TMEM16F is poorly understood. Using a combination of imaging, patch clamp, and scramblase assays, Zhang et al. show that TRPV4 and TMEM16F are expressed in human trophoblasts and in the BeWo trophoblast cell line. Increases in cytosolic Ca caused by activation of TRPV4 with a specific agonist GSK1016790A activates scrambling, TMEM16F currents, and trophoblast cell fusion.

Strengths.

These studies show that:

(1) TRPV4 and TMEM16F are expressed in both primary human trophoblasts and the trophoblast cell line WeBo. This is shown by RT-PCR, immunofluorescence, and functional assays.

(2) Ca influx via TRPV4 can activate TMEM16F in a native cell. This is an important observation because it demonstrates that TMEM16F can be activated by Ca influx mediated by an endogenous channel rather than by a Ca ionophore. Furthermore, it provides another tool that can be used to control TMEM16F experimentally.

(3) TRPV4 and TMEM16F exist in a local Ca microdomain in the cell membrane such that Ca influx through TRPV4 activates TMEM16F locally. This provides additional evidence for specific coupling between these two ion channels.

(4) Activation of TMEM16F via TRPV4-mediated Ca influx stimulates downstream processes leading to fusion of trophoblasts to form syncytiotrophoblasts in vitro. These authors had previously shown that TMEM16F is involved in trophoblast fusion, and this paper identifies the source of activating Ca that had not previously been known.

Weaknesses.

Although the paper has significant overall strengths, the paper has several major weaknesses.

(1) Although the paper shows that TRPV4 and TMEM16F are functionally coupled when TRPV4 is activated pharmacologically, the paper does not show that physiological activation of TRPV4 can activate TMEM16F. The experiments that show functional coupling between these channels utilize the TRPV4 agonist GSK101 and forskolin. These pharmacological agents are likely to activate TRPV4 to a greater extent than a physiological stimulus. Also, the mechanism of forskolin is not clear. I am not aware that it has been shown that cAMP-dependent processes activate TRPV4 in trophoblasts, although TRPV4 does have consensus PKA phosphorylation sites and there is suggestive evidence in other cell types that TRPV4 is regulated by PKA-dependent phosphorylation.

(2) Other sources of Ca are not considered. CaV and TRPV6, which have been reported to be expressed in trophoblasts, are not studied because they are thought to be unlikely to play a role, but these channels are not examined experimentally. Other TRPV channels are not studied because the Human Protein Atlas suggests expression is low, but the Human Protein Atlas is notoriously unreliable. Expression should be measured in the system used in this study. Ca-activated mobilization of internal Ca stores or other channels is not considered.

Figure 2A shows that TMEM16F KO reduces Ca influx significantly. These authors have previously argued that TMEM16F is a Ca-permeable non-selective cation channel, and a plausible alternative explanation is that Ca entering via TRPV4 activates Ca influx via TMEM16F and that scrambling follows this.

While TMEM16F knockdown reduces GSK101-stimulated Ca signaling, the effect of TMEM16F knockdown on forskolin-stimulated Ca signaling is not shown. Also, while the TRPV4 antagonist GSK219 or siRNA reduce trophoblast fusion, they only reduce it by about 40%. The effect of TRPV4 siRNA, while statistically significant, shows a lot of scatter with significant overlap between control and siRNA data points. This suggests that other mechanisms also activate TMEM16F or that TMEM16F is not required for fusion.

The authors do not address the question whether the coupling of TRPV4 to TMEM16F is a general phenomenon or specific to trophoblasts.

(3) While overall, the experimental design and execution is excellent, some experiments need additional controls, statistical analysis, or explanation.

a. Functional assays are convincing that TRPV4 is expressed in trophoblasts, but the immunofluorescence is ambiguous. Figure 1D does not show TRPV4 convincingly on the membrane. Figure 1E needs a counterstain to show the location of the membrane and quantification of overlap between a membrane marker and TRPV4. The validation of the antibody in Figure S2 should also show staining in untransfected cells. A western blot would also be helpful.

b. The colocalization of TRPV4 and TMEM16F (Figure S6) appears good in the selected field containing one cell, but this needs quantification over a large population of cells. Further, the colocalization is only demonstrated in cells over-expressing TMEM16F, not in native cells.

c. Figure 5 needs statistical analysis.

d. The dynamics of Ca signaling needs further explanation. Ca influx is dependent on voltage: at +100mV Ca influx should be negligible because Vm is approaching E(Ca). But, TMEM16F currents seem to be independent of the Ca driving force. TMEM16F currents increase with time in response to voltage steps to +100 mV, despite the fact that Ca influx is decreasing at this voltage. The amplitude of the currents are comparable to the currents elicited by 100 μm Ca in whole-cell patch clamp (compare Figure 4D and Figure S5), so this suggests that [Ca] at the TMEM16F channel remains close to 100 μm even at +100 mV. Furthermore, in Figure 5B panel 2, the amplitude of the current is independent of the duration of the prepulse at -100 mV. Maybe there is sufficient Ca influx at the holding potential to flood the cell, overwhelm native efflux mechanisms, and saturate Ca buffers, but if that is the case, this would invalidate the Ca microdomain concept developed in Figure 5.

*Reviewer #3:*

– The data, as presented, do not justify the conclusions that TRPV4 and TMEM16F are functionally coupled and that they colocalize in microdomains. This is not to say the data are not very interesting and consistent with these conclusions. Rather, the authors would be better served being more conservative in their interpretation.

– The result that a delayed cation conductance develops minutes after activation of TRPV4 is intriguing. For this result to be interpreted as activation of the ion conducting channel of TMEM16F, the authors would need to demonstrate a few things:

– It has been previously shown that the scramblase activity and ion conducing pore of TMEM16F are activated by the same ca^2+^ concentrations. The authors should show that the kinetics of the candidate TMEM16F currents occur simultaneously with activation of the scramblase activity.

– It is not clear that, in the presence of even low concentrations of EGTA, ca^2+^ through TRPV4 in excised patches could accumulate at the surface of the membrane at the high concentrations required. Indeed, the inward currents in Figure 5, as an example, are so small that, even if all the current were ca^2+^ it seems unlikely that local ca^2+^ at sufficiently high concentrations would occur. Using known information on the binding kinetics of the chelators used, the size of the currents measured, and the permeability of ca^2+^ relative to other ions, the authors could model the expected ca^2+^ kinetics in their patches to demonstrate whether the >20 µM ca^2+^ concentrations would develop as expected. There may be other ways the authors could demonstrate the actual ca^2+^ concentrations achieved on the outside of their patches.

– The ca^2+^-imaging data, using a dye with a KD = 1.2 µM, are overinterpreted. TMEM16F is not active even at 20 µM ca^2+^ – it requires about 200 µM to be active. The authors should be more conservative in how they discuss their data given the affinities of their dye. This argument applies to the EGTA and BAPTA data.

– The authors characterize the currents they attribute to TRPV4 as not having activation/deactivation kinetics, in contrast to the late-developing currents, which do show such kinetics. The pronounced outward rectification of the currents attributed to TRPV4 mean that they have very pronounced gating kinetics. They are just much faster than those of the late-developing currents.

– The imaging data are strictly qualitative and should be either subject to rigorous quantification with statistical analysis or eliminated. They add little to the present manuscript.

– Basic co-localization of two proteins in the plasma membrane on standard visible-light microscopy is not valid.

– The writing could use improvement to make it flow better. There are missing articles and singular-plural mismatches that make the text hard to follow.

– The introduction could be improved to make it easier for readers to follow. For those not expert in the field, the discussion of the authors' previous work, which sets up the logic for the present study, is difficult to understand. The last paragraph of the introduction goes well beyond the current manuscript and should be reined in.

– Why is the ca^2+^ influx in KO BeWo cells much smaller than the ca^2+^ influx in wt BeWo cells?

[Editors’ note: further revisions were suggested prior to acceptance, as described below.]

Thank you for resubmitting your work entitled "Functional coupling between TRPV4 channel and TMEM16F modulates human trophoblast fusion" for further consideration by *eLife*. Your revised article has been evaluated by Kenton Swartz (Senior Editor) and a Reviewing Editor.

The manuscript has been improved but there are some remaining issues that need to be addressed, as outlined below:

The reviewers have discussed their reviews with one another, and the Reviewing Editor has drafted this to help you prepare a revised submission. Whereas most concerns have been addressed in the revised version, reviewers still have some concerns regarding the presentation and discussion of key aspects of the results. These are outlined below.

Essential revisions:

1) The biological significance of the TRPV4-TMEM16F interaction has not been unequivocally established in the context of native human trophoblasts. This should be reflected in the way some of the conclusions are presented throughout the manuscript, by toning down certain statements, especially those related to a 'key role of TRPV4 channels in human trophoblast fusion'. Solid evidence is provided that activation of TRPV4 channels has the capacity to activate TMEM16F channels, and could potentially play an important role in trophoblast fusion, but no experimental evidence is provided to show that TRPV4 channel activity is indeed required for this process in native human cells.

The term 'human trophoblast' is used at multiple points throughout the manuscript, including in the titles of figures and Results sub-sections. The term BeWo cells (or immortalized trophoblast cell line) should be utilized instead; most of the key experiments and controls were done in these cells that could exhibit biologically relevant differences compared to non-immortalized human trophoblast cells. Particularly because the expression of TRPV4 channels in human trophoblasts seems to be low (less than 10-fold as compared to TMEM16F, Figure 1B), and other calcium channels are expressed by these cells, it is best to be cautious when drawing conclusions about the specific importance of TRPV4 channels in vivo based almost exclusively in data from immortalized cells.

2) The connection between the interplay of TMEM16F and TRPV4 in placental organoids is quite weak. Downregulation of genes is a poor readout to make the physiological inferences the authors would like to make. Are there any morphological effects on GSK219 -treated organoids, like those seen in the BeWo cells? Is it possible to measure PS exposure in the organoids? The experiment with the organoids would constitute stronger evidence for the role of TRPV4 channels in trophoblast fusion in vivo than the experiment with BeWo cells in Figure 7A-D if direct evidence were provided that inhibition of TRPV4 channels reduces cell fusion, and results in increased PS exposure. It is unclear why cell fusion was not quantified directly from the organoids, and to what extent the markers that were analyzed are indicative of cell fusion. This needs to be addressed. Despite claims to the contrary, the authors provide no evidence that inhibition of TRPV4 hinders trophoblast fusion in the organoid model. Further, the downregulation of fusogenic genes might point to a mechanism different from the TMEM16F mediated one that is proposed here. Without direct evidence connecting the data in cell lines to that in the organoid, I believe the manuscript would be strengthened by the removal of this data.

3) Calcium entry into TRPV4 siRNA-treated cells caused by saturating GSK101 (Figure S5A and B) appears to be comparable, if not higher, than in WT BeWO cells stimulated with 0.2 nM GSK101 (Figure 6A and B), yet transient and localized Annexin binding is only observed in the latter but not in the former experiments. This is confusing and needs to be discussed and explained, as it is unclear whether local and transient Annexin binding was looked at in other experiments where it might have also been observed.

4) The authors claim that at low activator concentrations PS exposure is transient. Looking at the representative images shown in Fig6, the transient PS exposure appears to revert in one location, but in the other areas, it persists. Is this just a matter of 'ending the experiment too soon', or does this represent local but irreversible PS exposure? At present, the writing indicates that all PS exposure is reversible but the data does not seem to agree. Quantification of the proportion of temporary vs persistent areas of PS exposure is important.

5) In most key figures involving fluorescence imaging only one or two cells are displayed. Although quantification is provided for more than one cell, these numbers still seem small compared to the total number of cells that can be imaged on a coverslip of in a well with seeded cells. It would strengthen the manuscript and provide a more solid representation of the findings if images were provided where a population of cells can be observed, all having similar phenotypes to the representative case that is currently shown. Otherwise, it is unclear if only cells showing the most marked phenotypes were analyzed and displayed.

6) In the response to reviewers and in the manuscript text it is implied that calcium sensitivity of TMEM16F channels is underestimated in whole-cell recordings because that experimental configuration is disruptive to cellular conditions. On the other hand, results from excised patches, which are even more disruptive to the natural conditions of cells, appear to yield higher sensitivity to calcium. Furthermore, the highest sensitivity is observed with calcium imaging, whereas pointed out in the first review, the affinity of the sensor for calcium is a determinant of what is observed. Because identifying a calcium source for TMEM16F channels that enables their activity is central to this manuscript, more clarification is required about the calcium sensitivity of TMEM16F channels. Specifically, it must be clearly stated whether it is speculated that cellular factors contribute to increasing the sensitivity of TMEM16F channels to calcium, or if cellular structures that can get disrupted in the whole cell configuration contribute to the formation of very small calcium microdomains where the concentration of the cation can increase to sufficient levels that activate TMEM16F channels. It would be important to mention here the work by the Dutzler lab (Alvadia et al., *ELife*, 2019) which showed that purified TMEM16F is activated by ca^2+^ with an EC50 ~1 uM.

*Reviewer #1:*

Many of the concerns have been addressed in the revised version of the manuscript, but I consider that there are still some issues that need to be resolved before publication, as well as some suggestions that can further strengthen the manuscript.

1) The biological significance of the TRPV4-TMEM16F interaction has not been unequivocally established in the context of native human trophoblasts. This should be reflected in the way some of the conclusions are presented throughout the manuscript, by toning down certain statements, especially those related to a 'key role of TRPV4 channels in human trophoblast fusion'. Solid evidence is provided that activation of TRPV4 channels has the capacity to activate TMEM16F channels, and could potentially play an important role in trophoblast fusion, but no experimental evidence is provided to show that TRPV4 channel activity is indeed required for this process in native human cells.

The term 'human trophoblast' is used at multiple points throughout the manuscript, including in the titles of figures and Results sub-sections. The term BeWo cells (or immortalized trophoblast cell line) should be utilized instead; most of the key experiments and controls were done in these cells that could exhibit biologically relevant differences compared to non-immortalized human trophoblast cells. Particularly because the expression of TRPV4 channels in human trophoblasts seems to be low (less than 10-fold as compared to TMEM16F, Figure 1B), and other calcium channels are expressed by these cells, it is best to be cautious when drawing conclusions about the specific importance of TRPV4 channels in vivo based almost exclusively in data from immortalized cells.

2) The experiment with the organoids would constitute stronger evidence for the role of TRPV4 channels in trophoblast fusion in vivo than the experiment with BeWo cells in Figure 7A-D if direct evidence were provided that inhibition of TRPV4 channels reduces cell fusion. It is unclear why cell fusion was not quantified directly from the organoids, and to what extent the markers that were analyzed are indicative of cell fusion. This needs to be addressed.

3) Calcium entry into TRPV4 siRNA-treated cells caused by saturating GSK101 (Figure S5A and B) appears to be comparable, if not higher, than in WT BeWO cells stimulated with 0.2 nM GSK101 (Figure 6A and B), yet transient and localized Annexin binding is only observed in the latter but not in the former experiments. This is confusing and needs to be discussed and explained, as it is unclear whether local and transient Annexin binding was looked at in other experiments where it might have also been observed.

4) In most key figures involving fluorescence imaging only one or two cells are displayed. Although quantification is provided for more than one cell, these numbers still seem small compared to the total number of cells that can be imaged on a coverslip of in a well with seeded cells. It would strengthen the manuscript and provide a more solid representation of the findings if images were provided where a population of cells can be observed, all having similar phenotypes to the representative case that is currently shown. Otherwise, it is unclear if only cells showing the most marked phenotypes were analyzed and displayed.

5) In the response to reviewers and in the manuscript text it is implied that calcium sensitivity of TMEM16F channels is underestimated in whole-cell recordings because that experimental configuration is disruptive to cellular conditions. On the other hand, results from excised patches, which are even more disruptive to the natural conditions of cells, appear to yield higher sensitivity to calcium. Furthermore, the highest sensitivity is observed with calcium imaging, whereas pointed out in the first review, the affinity of the sensor for calcium is a determinant of what is observed. Because identifying a calcium source for TMEM16F channels that enables their activity is central to this manuscript, more clarification is required about the calcium sensitivity of TMEM16F channels. Specifically, it must be clearly stated whether it is speculated that cellular factors contribute to increasing the sensitivity of TMEM16F channels to calcium, or if cellular structures that can get disrupted in the whole cell configuration contribute to the formation of very small calcium microdomains where the concentration of the cation can increase to sufficient levels that activate TMEM16F channels.

*Reviewer #3:*

This is a revision of a previously submitted manuscript. The authors addressed satisfactorily most of the first round of critiques with extensive additional experiments. I am mostly satisfied, but I have two residual concerns with what appear to be overclaims by the authors that are detailed below.

The authors claim that at low activator concentrations PS exposure is transient. Looking at the representative images shown in Fig6, the transient PS exposure appears to revert in one location, but in the other areas, it persists. Is this just a matter of 'ending the experiment too soon', or does this represent local but irreversible PS exposure? At present, the writing indicates that all PS exposure is reversible but the data does not seem to agree. Quantification of the proportion of temporary vs persistent areas of PS exposure is important.

The connection between the interplay of TMEM16F and TRPV4 in placental organoids is quite weak. Downregulation of genes is a poor readout to make the physiological inferences the authors would like to make. Are there any morphological effects on GSK219 -treated organoids, like those seen in the BeWo cells? Is it possible to measure PS exposure in the organoids? The conclusions from this section (both at the end of the results and in the discussion) need to be toned down. Despite claims to the contrary, the authors provide no evidence that inhibition of TRPV4 hinders trophoblast fusion in the organoid model. Further, the downregulation of fusogenic genes might point to a mechanism different from the TMEM16F mediated one that is proposed here. Without direct evidence connecting the data in cell lines to that in the organoid, I believe the manuscript would be strengthened by the removal of this data.

---

## [Author Response]

[Editors’ note: the authors resubmitted a revised version of the paper for consideration. What follows is the authors’ response to the first round of review.]

Essential revisions:1) The biological relevance for the possible interaction between TRPV4 and TMEM16F channels needs to be toned down throughout the manuscript. The authors should provide additional data in support of a biological relevance for their findings. It would be expected that TRPV4-KO mice have phenotypes associated with trophoblast fusion deficiencies if the association between TRPV4 and TMEM16F channels is as relevant as suggested – this should be at least further discussed. The authors could also use GSK101 concentrations that result in cytosolic calcium elevations that closely resemble those that would occur in vivo, or at least provide a discussion justifying the concentration of GSK101 that was used. The authors do not address the question whether the coupling of TRPV4 to TMEM16F is a general phenomenon or specific to trophoblasts.

We thank the reviewer for raising these critical points.

1. We have conducted more experiments (Figures 6, S11 and 7E-F) to support the biological relevance of TRPV4-TMEM16F coupling in trophoblast (please see discussions below). We also substantially revised the manuscript and paid attention to not over claim.

2. As the reviewer suggested, we carefully examined the histology of the placentas from the Trpv4 deficient (KO) mice. However, we did not observe obvious defects on the gross morphology of the placentas (Figure S13A-B). Using MCT1 and MCT2 as markers for the SynT-1 and SynT-2 syncytiotrophoblast layers of the mouse labyrinth, no obvious change in the integrity of the syncytiotrophoblast layers is observed between the WT and Trpv4 KO placentas (Figure S13C-D). The lack of fusion defect in the mouse placentas is likely due to the distinct expression profiles of TRPV4 in the human and mouse placentas. According to a recent systematic mapping of Trp channel expression in the mouse placenta (1), Trpv4 is only expressed in the maternal decidua but not the junctional zone nor the labyrinth, where cytotrophoblasts fuse and form the two syncytiotrophoblast layers (1). Interestingly, the differences in expression profiles are not just limited to TRPV4. Trpv6, a well-documented ca^2+^ transporter in human trophoblasts (2, 3) (Figure S1), is only weakly expressed in the mouse labyrinth and more prominently expressed in the intraplacental yolk sac (1, 2). Tprv2, on the other hand, is highly expressed in the mouse labyrinth (1) yet only weakly expressed in human cytotrophoblasts and syncytiotrophoblasts according to the Human Protein Atlas (Figure S1). The species-specific differential expressions of TRPV channels are possibly rooted in the fact that the placenta is a comparatively more recently evolved organ (4). Therefore, TRPV4 might play a key role in human placentas and a more marginal role in mouse placenta. Future studies are therefore needed to systematically compare the physiological functions of other TRPV channels in human and murine trophoblasts.

3. It is unfortunate that the species-dependent expression of Trpv4 in the murine placenta prevents us from studying TRPV4-TMEM16F coupling in trophoblast fusion using mouse as an animal model. To overcome this obstacle, we tried a different approach using a novel human trophoblast organoid model recently developed in Dr. Carolyn Coyne’s lab (5). We found that pharmacological inhibition of TRPV4 with GSK219, a specific TRPV4 inhibitor, can dramatically reduce the expression of the placental syncytiotrophoblast marker genes, *CSH1* and *CGB*, in the organoids (Figure 7). *CSH1* and *CGB* encode human placental lactogen (hPL) and hormone-specific β-subunit of human chorionic gonadotropin (βhCG), respectively. Both proteins are specifically synthesized by syncytiotrophoblasts. Our new experiment using the human trophoblast organoids thus demonstrates that TRPV4 indeed plays an important role in mediating human trophoblast fusion, and further supports the biological relevance of our finding. We have added the TRPV4 KO mouse placenta and human trophoblast organoid results together with corresponding discussions in the revised manuscript.

4. Following the reviewer’s suggestion, we applied extremely low concentration of GSK101 (0.2nM) to BeWo cells to mimic potential ca^2+^ elevation induced by physiological TRPV4 activation. Different from high concentration of GSK101 that causes sustained elevation of ca^2+^, 0.2 nM GSK101 induced transient increase of calcium that quickly decayed (Figure 6, video 3). Concomitant with the ca^2+^ dynamics following mild TRPV4 activation, we found that there is local AnV binding at some membrane protrusions of the treated BeWo cells (White box in Figure 6A), which was not observed in TMEM16F knockout BeWo cells (Figure S12). Interestingly, the fluorescence signal from the bound AnV gradually disappeared after calcium level decreased. This is consistent with the observation by the Nagata lab in their 2010 Nature paper (6). The authors discovered that when the Ba/F3 B-cell line was stimulated with 1 µM A23187 in 0 Ca, internal ca^2+^ store release can transiently activate lipid scramblase and induce reversible PS exposure. This is similar to what we observed in the 0.2 nM GSK101-stimulated BeWo trophoblast cell line. Our finding thus demonstrated that 1) TRPV4-mediated TMEM16F scramblase activation can happen under more physiologically relevant ca^2+^; 2) non-physiologically high ca^2+^ (20 -200 µM) is not required to activate native TMEM16F scramblases in trophoblasts (see our response to Essential Revision #7 for more details); 3) TMEM16F scramblase is more likely activated transiently and at subcellular locations under physiological conditions. The local and transient collapse of lipid asymmetry is reversible, suggesting that TMEM16F mediated lipid scrambling can be highly dynamic under physiological conditions. We have included this new result in the revised manuscript and added corresponding discussions. We hope that our finding of spatiotemporal TMEM16F activation can shine new light on understanding TMEM16F’s physiological roles including trophoblast fusion.

5. Our in vitro experiments demonstrated that the coupling between TRPV4 and TMEM16F plays an important role in mediating trophoblast fusion. Although this particular coupling in trophoblasts could be a cell type specific phenomenon that requires the co-expression of TRPV4 and TMEM16F, we believe that the coupling between ca^2+^ permeable channels (including but not limited to TRPV4) and TMEM16F should be a general principle for ca^2+^-dependent activation of TMEM16F scramblases under physiological conditions. Future studies are needed to examine this hypothesis in different cell types. We have included this discussion in our Discussion section.

2) The emphasis on the findings pertaining to human trophoblasts should also be toned down throughout, as most of the experiments were done in a cell line and not in native cells.

We appreciate the suggestion. We have conducted the following experiments to strengthen our conclusion in human trophoblasts.

1. Our results in Figure S6 show that TRPV4 and TMEM16F are functionally coupled in human primary trophoblasts.

2. We also conducted functional experiments using a novel human trophoblast organoid model. We showed that pharmacologically inhibiting TRPV4 indeed dramatically hinders trophoblast fusion in the organoids as evidenced by the significantly reduced *CSH1 and CGB* expression, two marker genes of syncytiotrophoblasts (Figure 7E-F). This experiment indicates that TRPV4 indeed plays an important role in regulating human trophoblast fusion. Please see our response to Essential Revisions #1.3 for more details.

We have thoroughly revised the manuscript accordingly.

3) Group data quantitation should be provided for all relevant figures (Figure 1I and J, Figure 3B, Figure 4A-C, Figure 5, Figure S2, Figure S3A, Figure S3B, Figure S4, Figure S5, Figure S6, Figure S7). The imaging data are strictly qualitative and should be either subject to rigorous quantification with statistical analysis or eliminated. Functional assays are convincing that TRPV4 is expressed in trophoblasts, but the immunofluorescence is ambiguous. Figure 1D does not show TRPV4 convincingly on the membrane. Figure 1E needs a counterstain to show the location of the membrane and quantification of overlap between a membrane marker and TRPV4. The validation of the antibody in Figure S2 should also show staining in untransfected cells. A western blot would also be helpful.

We apologize for missing the quantitative analysis for some of the results. We have added the missing information in the revision. The following are our changes.

1. We added statistical analysis for all relevant figures listed (new Figure 1J, Figure 3D-E, Figure 4B, 4E, Figure 5C-D, Figure 6C, Figure 7F, Figure S3B, Figure S5B, C, E, Figure S6B, Figure S7B-C and E-F, Figure S9B-C, Figure S10B, S10D-E, Figure S11B-C).

2. We validated the TRPV4 antibody using Flag-tagged TRPV4 in HEK293T cells. The immunofluorescence of TRPV4 colocalizes with anti-Flag signals in the TRPV4-Flag transfected cells (Figure S2A) but not in the untransfected cells (Figure S2B), validating the specificity of this anti-TRPV4 antibody. As the validation with immunofluorescence gave us clean results, a western blot was not performed.

3. We reperformed the immunofluorescence staining of TRPV4 using the validated antibody on human placental tissue (Figure 1D). Our result clearly showed that TRPV4 is expressed in both the single nucleated cytotrophoblasts and the multinucleated syncytiotrophoblasts. Membrane expression of TRPV4 in the trophoblasts was also observed (Figure 1D, right). In addition, we also co-labelled the BeWo cell membrane using a membrane marker FM1-43 (7) (Figure S2C). FM1-43 signal (red) and immunofluorescence staining of TRPV4 (green) overlap on the plasma membrane, supporting our patch clamp finding that TRPV4 and TMEM16F are located in spatial proximity on BeWo cell surface (Figure 5).

4) It has been previously shown that the scramblase activity and ion conducing pore of TMEM16F are activated by the same ca^2+^ concentrations. The authors should show that the kinetics of the candidate TMEM16F currents occur simultaneously with activation of the scramblase activity.

To address this concern, we simultaneously monitored TMEM16F current and lipid scrambling after GSK101 application on WT BeWo cells using whole cell patch clamp-lipid scramblase fluorometry (PCLSF) (8, 9). Our result showed that TMEM16F current and lipid scrambling activity appeared at similar time under whole cell configuration (Figure S8). Consistent with previous reports, there was a long lag time to reliably detect PS exposure and TMEM16F current following GSK101 application and TRPV4 activation (before t2). The time course for TMEM16F activation is similar to the previous reports using whole-cell patch clamping or whole-cell PCLSF, in which high concentration of ca^2+^ directly dilates into the recorded cells from recording pipette. The exact mechanism underlying the long delay of TMEM16F activation under whole cell configuration is still not entirely clear. Please see our response in Essential Revisions #7 and the revised manuscript for more discussion on this topic.

5) Why is the ca^2+^ influx in KO BeWo cells much smaller than the ca^2+^ influx in wt BeWo cells? These authors have previously argued that TMEM16F is a Ca-permeable non-selective cation channel, and a plausible alternative explanation is that Ca entering via TRPV4 activates Ca influx via TMEM16F and that scrambling follows this.

We thank the reviewer for raising this very interesting question. The ca^2+^ influx in KO BeWo cells is slightly weaker than the ca^2+^ influx in WT BeWo cells (Figure 2C). Our previous patch clamp characterization demonstrated that TMEM16F itself has the capability to permeate ca^2+^ (10). A recently report from the Jan lab also supports that ca^2+^ indeed can permeate through active TMEM16F using an imaging assay (11). Although the exact reason still needs to be further investigated, we agree with the reviewer that TMEM16F in trophoblasts may indeed serve as a ca^2+^-permeable, non-selective channel, which is activated by TRPV4-mediated ca^2+^ influx and in turn contributes to more ca^2+^ influx. Therefore, both TRPV4 and TMEM16F can contribute to sustained ca^2+^ increase in WT BeWo cells. On the other hand, GSK101-stimulated ca^2+^ entry in TMEM16F KO BeWo cells may be solely contributed by TRPV4 but not TMEM16F channels. Of course, we cannot rule out the possibility that genetic ablation of TMEM16F in BeWo cell may reduce TRPV4 expression or attenuate its response to GSK101. These possibilities are worth further investigating in future studies.

6) The dynamics of Ca signaling needs further explanation. It is not clear that, in the presence of even low concentrations of EGTA, ca^2+^ through TRPV4 in excised patches could accumulate at the surface of the membrane at the high concentrations required. Indeed, the inward currents in Figure 5, as an example, are so small that, even if all the current were ca^2+^ it seems unlikely that local ca^2+^ at sufficiently high concentrations would occur. Using known information on the binding kinetics of the chelators used, the size of the currents measured, and the permeability of ca^2+^ relative to other ions, the authors could model the expected ca^2+^ kinetics in their patches to demonstrate whether the >20 µM ca^2+^ concentrations would develop as expected. There may be other ways the authors could demonstrate the actual ca^2+^ concentrations achieved on the outside of their patches.

We appreciate the comment. The following are our response and clarification.

1. There is a mis-conception in the field with regard to the ca^2+^ concentration required for TMEM16F activation. Based on previous whole cell recordings, >20 µM ca^2+^ is needed to activate TMEM16F current. However, under physiological conditions, cells barely can achieve such high concentration of ca^2+^, even within ca^2+^ microdomains (12, 13). How TMEM16F plays a role under physiological conditions is thus difficult to understand. Our previous inside-out patch measurements showed that ca^2+^ as low as 2 µM can reliably activated TMEM16F current (10). To test if physiologically relevant ca^2+^ level could activate TMEM16F scramblase in intact cells, we applied ultra low concentration of GSK101 (0.2 nM) to BeWo cells. As shown in Figure 6, mild TRPV4 activation only triggers transient ca^2+^ increase followed by local PS exposure in some membrane processes. On the other hand, the same GSK101 stimulation only triggers ca^2+^ transients without local PS exposure in TMEM16F knockout BeWo cells (Figure S12). Our new results with physiologically relevant ca^2+^ are consistent with the observation by the Nagata lab in 2010 (6).The authors found that ca^2+^-ionophore-induced store ca^2+^ release can induce reversible PS exposure in Ba/F3 cells. Based on our previous inside-out patch measurement (10), our new result using 0.2 nM GSK101 and the observation from the Nagata lab (6), it is clear that TMEM16F activation under physiological conditions requires much lower ca^2+^ level than the nonphysiological 20-200 µM ca^2+^ range. The overestimated of ca^2+^ concentration needed for TMEM16F activation is likely due to the artifacts under the invasive whole cell recording, which can disrupt intracellular environment. Please see our responses to Essential Revision #1, #7 and revised manuscript for more detailed discussions.

2. In Figure 5, we used excised outside-out patch instead of whole-cell recording. The advantages of outside-out patch clamping are that 1) TMEM16F current can be instantaneously elicited without the complication from the long delay under whole cell configuration (please see our discussion in Response to Essential Revision #7 for more details); 2) fast channel rundown (likely due to PIP2 depletion (14)) can be largely prevented; 3) different from excised inside-out patches that are usually perfused with intracellular solution, the pipette solution remains static thereby ca^2+^ influx through TRPV4 channels will not be instantaneously washed away from patch surface. Considering the high single channel conductance and high ca^2+^ permeability of TRPV4, ca^2+^ will be high within the nanodomain near the intracellular mouth of a TRPV4 channel, which will decay within microdomain and then completely diffuse into the bulk solution. Therefore, when the pipette solution only had 0.2 mM EGTA, the locally high ca^2+^ can instantaneously activate the TMEM16F channels within microdomains from the TRPV4 channels (Figure 5B #2). When EGTA in the pipette solution is increased to 2 mM, TMEM16F channel activation requires much prolonged prepulse at -100 mV to allow more ca^2+^ entry through TRPV4 channels (Figure 5B #3). These functional experiments thus support that ca^2+^ entry through TRPV4 channels indeed can transiently accumulated on the intracellular cell surface and activate TMEM16F within the microdomains. In order to facilitate the readers to understand our experimental design and interpret our results, we have included cartoon diagrams for each experimental condition in Figure 5B.

3. The small GSK101-induced current in 0 ca^2+^ should be derived from TRPV4 channels. According to the reported single channel conductance of ~50-60 pS for inward TRPV4 current (15), the estimated number of TRPV4 in each outside-out patch was about 7. This is consistent with the relatively low expression level of TRPV4 using immunostaining (Figure S2C). It is known that TRPV4 channel has large single channel conductance of 50-60 pS and is highly permeable to ca^2+^ with (PCa/PNa = 6-10) (15-17). Therefore, when GSK101 binds to the channel, it is plausible that there will be robust ca^2+^ flux through TRPV4 channels in our outside-out patches before ca^2+^ diffuses into the bulk pipette solution. The local ca^2+^ near TRPV4 channels is high enough to activate TMEM16F channels in the vicinity as evidence by the incapability of 2 mM EGTA to prevent TMEM16F channel opening (Figure 5B, #3). We believe that our results using different ca^2+^ chelators clearly demonstrate the functional coupling between TRPV4 and TMEM16F. Estimation of net ca^2+^ going through each TRPV4 channel seems beyond the scope of this study.

Ca influx is dependent on voltage: at +100mV Ca influx should be negligible because Vm is approaching E(Ca). But, TMEM16F currents seem to be independent of the Ca driving force. TMEM16F currents increase with time in response to voltage steps to +100 mV, despite the fact that Ca influx is decreasing at this voltage. The amplitude of the currents are comparable to the currents elicited by 100 μm Ca in whole-cell patch clamp (compare Figure 4D and Figure S5), so this suggests that [Ca] at the TMEM16F channel remains close to 100 μm even at +100 mV. Furthermore, in Figure 5B panel 2, the amplitude of the current is independent of the duration of the prepulse at -100 mV. Maybe there is sufficient Ca influx at the holding potential to flood the cell, overwhelm native efflux mechanisms, and saturate Ca buffers, but if that is the case, this would invalidate the Ca microdomain concept developed in Figure 5.

We apologize that we did not clearly describe the holding voltage for the figures mentioned. In these experiments, we did not constantly hold the membrane potential at +100mV. Instead, we ran a voltage ramp protocol every 5 s from -100mV (to induce enough ca^2+^ influx) to +100mV (to monitor 16F activity). Between each ramp, the membrane potential was held at -60 mV, which also promotes ca^2+^ entry. Therefore, ca^2+^ should have entered the recorded cells during the long holding period at -60 mV and gradually accumulated over several minutes of recording. This is likely the reason why the GSK101 induced whole cell TMEM16F current (Figure 4D) is comparable to the current elicited by 100 µM Ca (previous Figure S5A, now revised Figure S7A).

For Figure 5B, we used outside-out patch recording, not whole cell recording. Our pipette solution did not contain ATP. So, ca^2+^ efflux system is not expected to be functional. The effects that we observed should be the summation of ca^2+^ influx through TRPV channels, the buffering capacity and kinetics of EGTA (microdomain) and BAPTA (nanodomain), as well as the diffusion rate of ca^2+^ into the bulk pipette solution (bulk diffusion). The volume of pipette solution is enormous compared with the amount of ca^2+^ influx, and there is no physical barrier to prevent ca^2+^ to diffuse into the bulk pipette solution. Therefore, ca^2+^ entry through a TRPV4 channel in an outside out membrane patch is expected to readily diffuse away from the mouth of the channel without sufficient accumulation, even in 0.2 mM EGTA (panel 2). Here we plotted all six outside-out recordings under this condition (Author response image 1). The immediate activation of TMEM16F current after the first 0.1 s prepulse at -100 mV with GSK101 stimulation clearly demonstrates that TMEM16F stays in the close proximity to TRPV4 channels so that ca^2+^ influx through TRPV4 channels can readily activated nearby TMEM16F channels. However, there were no or minimal increase of TMEM16F current when the prepulse was prolonged from 0.1 s to 1 s (Author response image 1). Lack of ca^2+^ accumulation near the patched membrane is likely the main reason why we did not see obvious increase of TMEM16F current amplitude. Nevertheless, when EGTA concentration was increased to 2 mM (Figure 2B, panel 3), an obvious delay of TMEM16F activation indicates that higher EGTA within the microdomain of a TRPV4 channel can effectively chelate ca^2+^ so that only prolonged prepulse can bring enough ca^2+^ to overcome the chelating effect of 2 mM EGTA. On the other hand, 2 mM fast ca^2+^ chelator BAPTA completely prevented TMEM16F activation even when the prepulse was prolonged to 1 s. All these results under outside-out configuration thus demonstrate that TRPV-TMEM16F coupling happens within ca^2+^ microdomains in the trophoblasts.

**Author response image 1. sa2fig1:** Six different outside-out patch recordings from wildtype (WT) BeWo cells with 0. 2 EGTA in the pipette and 2.5mM ca^2+^ in the extracellular solution. 30nM GSK101 was applied through a perfusion system, the protocol used here is the same as Figure 5A.

7) The ca^2+^-imaging data, using a dye with a KD = 1.2 µM, are overinterpreted. TMEM16F is not active even at 20 µM ca^2+^ – it requires about 200 µM to be active. The authors should be more conservative in how they discuss their data given the affinities of their dye. This argument applies to the EGTA and BAPTA data.

We appreciate the reviewer asked this important question. This is a critical yet unsettled question in the field, ie. how high the ca^2+^ concentration is needed to activate TMEM16F lipid scramblase under physiological conditions.

1. Utilizing TMEM16F’s channel activity, its EC_50_ values have been estimated to range from ~3 to 100 µM, depending on the configuration and ionic conditions of the patch clamp recording (18-25). Our previous inside-out patch clamp recording of the heterologously expressed TMEM16F in HEK293 cells showed that the EC_50_ is about 14 µM, whereas the endogenous TMEM16F current in megakaryocytes has much higher ca^2+^ sensitivity with EC_50_ of ~3 µM (10). Ca^2+^ concentration as low as 2 µM can activate TMEM16F current under inside-out configuration.

2. The statement that TMEM16F channel activation needs >20 µM ca^2+^ is mainly derived from whole-cell patch clamp recordings (26, 27). However, multiple labs report that there is an unusual ~5-15 minute lag time before TMEM16F current activation even under high ca^2+^ (20 – 200 µM), which was not observed in excised inside-out or outside-out patch recordings. Given the measured EC_50_ values using inside-out patches, the long lag time under whole cell configuration cannot be explained by the delay of ca^2+^ diffusion from pipette solution to the patched cell (usually equilibrate within a couple of minutes). The exact reason for the delayed activation under whole cell configuration is unclear, even though a recent publication showed that the actin cytoskeleton may play some role in negatively regulating TMEM16F (28). Regardless of the underlying mechanism for the long delay under whole cell configuration, it is clear that disruption of native environment by breaking in whole cell configuration is likely the major reason. Therefore, the reported ca^2+^ range for TMEM16F current activation based on whole cell recordings does not reflect, and even underestimates the ca^2+^ sensitivity of TMEM16F scramblase under physiological conditions.

3. Unfortunately, the exact calcium concentration required for native TMEM16F scramblase activation has not been established. Based the following evidence and reasoning, we believe that the calcium level required to activate TMEM16F scramblase might be much lower than the value obtained under whole cell patch conditions.

4. Low concentrations of ionophores (e.g. 1 µM A23187 in Suzuki, et al., Nature 2010 (6) and Yang et al. Cell 2012 (10)) can trigger obvious PS exposure in Ba/F3 cells and mouse platelets. In the Ba/F3 cells, even ca^2+^ release from internal store can promote mild PS exposure (6). Interestingly, the low concentration ionophore-triggered PS exposure was reversible when the PS exposed Ba/F3 cells were incubated with BAPTA-AM at 37 °C for 5 min or cultured them in ca^2+^-free medium at 37 °C for 12 h. When we addressed the Essential Revisions #1, we also found that low concentration of GSK101 (0.2 nM)-triggered ca^2+^ elevation can induce transient PS exposure in some membrane processes of BeWo cells (Figure 6). The exposed PS rapidly disappeared from cell surface, likely due to PS internalization by flippases (6). Low concentrations of ca^2+^ ionophore only can trigger limited intracellular ca^2+^ increase (0.5 µM ionomycin results in ~0.5 µM ca^2+^ in platelet (29)). Although we didn’t quantify free ca^2+^ concentration in the 0.2 nM GSK101 experiment, the rapid decay of ca^2+^ level (Figure 6) suggests that sustained global ca^2+^ elevation in this condition is also very limited. Therefore, the observed reversible PS exposure mediated by TMEM16F in Ba/F3 and BeWo cells suggest that the activation of TMEM16F scramblase under physiological conditions does not require extremely high ca^2+^ of 20-200 µM, which is non-physiological.

5. The above observations also suggest that TMEM16F lipid scramblase can be partially activated in a spatiotemporal fashion under physiological conditions. However, due to the low sensitivity of the fluorescence-based scrambling assays, most of the studies including ours monitored global PS exposure, and the local and transient PS exposure has not been thoroughly investigated. Therefore, the calcium level required to activate TMEM16F scramblase under physiological conditions might have been overestimated.

6. In our hands, 1-2 µM of ionomycin can robustly activated TMEM16F in our stable HEK293 cell line, which induces global PS exposure and membrane remodeling (20, 30). According to numerous reports, this range of ionomycin only can induce several µM of intracellular ca^2+^ (2 µM ionomycin induces 5 µM ca^2+^ in platelets (29)), which is still much lower than 20 – 200 µM ca^2+^ estimated by whole cell patch clamping.

7. According to our EGTA and BAPTA experiments (Figure 5), we estimated that TMEM16F is located within the microdomain (EGTA sensitive) but not nanodomain (BAPTA sensitive) of TRPV4 channels. As calcium level decays rapidly from nanodomain (~20 µM) to microdomain (a few µM) (31), TMEM16F scramblase can hardly be activated if its activation requires 20-200 µM ca^2+^.

We have included corresponding discussions in the revised manuscript.

8) The authors characterize the currents they attribute to TRPV4 as not having activation/deactivation kinetics, in contrast to the late-developing currents, which do show such kinetics. The pronounced outward rectification of the currents attributed to TRPV4 mean that they have very pronounced gating kinetics. They are just much faster than those of the late-developing currents. This must be clarified.

We appreciate the reviewer pointing out the differences in the activation/deactivation kinetics for the early-developing current (TRPV4) and late-developing current (TMEM16F). Yes. GSK101-induced outward rectifying current is indeed fast in kinetics (32). TMEM16F current, on the other hand, shows much slower kinetics even in the presence of high ca^2+^ (0.1 to 1 mM) (8, 10). When we knockdown TRPV4 in BeWo cells, we observed significant reduction of this early-developed current with fast kinetic (Figure S5C-E), suggesting that this current is indeed mediated by TRPV4. We have clarified the kinetic differences between TRPV4 and TMEM16F channels in our revised manuscript.

9) The authors should provide a control showing that the TRPV4 siRNA does not directly affect expression or activity of TMEM16F channels. This could be done by performing patch-clamp recordings with added calcium to stimulate these channels, and by showing that treatment with a calcium ionophore results in robust Annexin V binding to the outer leaflet in siRNA-treated cells.

Thanks for asking for this important control experiment. Patch clamp-lipid scramblase fluorometry has been done on the cells transfected with control-siRNA and two TRPV4 siRNAs. To quantify TMEM16F expression level and minimize the interference by the long delay time before TMEM16F activation, 1 mM Ca was included in the pipette (33). Our results show that ca^2+^-induced TMEM16F current amplitudes are comparable between the control and TRPV4 siRNAs treated groups, suggesting that the expression of TMEM16F is not altered after TRPV4 knockdown (Figure S10C-F). In addition, CaPLSase activity is also similar between the control and knockdown groups (Figure S10A-B), indicating that the scramblase function and expression of TMEM16F are not altered after knocking down of TRPV4.

10) The authors should further validate their TRPV4-knockdown by showing a decreased channel expression in the plasma membrane with immunofluorescence or western blots.

We thank the reviews for the suggestion. We conduct the immunofluorescence staining of TRPV4 in TRPV4-knockdown BeWo cells. The results showed a significant reduction of TRPV4 expression on the plasma membrane (Figure S4). In addition to immunofluorescence, our ca^2+^ imaging (Figure S5A-B) and patch clamp recording (Figure S5C-E) also show that GSK101-induced TRPV4 current and ca^2+^ influx are diminished in TRPV4 siRNA knockdown BeWo cells (Figure S5). All these results support efficient knockdown of TRPV4 in BeWo cells.

11) The authors should perform outside-out experiments in the presence of 2.5mM extracellular calcium and 0.2 mM EGTA in the pipette without GSK101 exposure, as in Figure 5, and show that no TMEM16F current buildup occurs after long wait times.

This experiment has been done as suggested (Figure S11). Our results show that in the absence of GSK101, there is no TMEM16F current buildup after long wait time when extracellular solution contains 0 ca^2+^ or 2.5 mM ca^2+^. This further supports our conclusion that GSK101-induced activation of TMEM16F requires ca^2+^ influx through TRPV4.

12) The mechanism of Forskolin is not clear. I am not aware that it has been shown that cAMP-dependent processes activate TRPV4 in trophoblasts, although TRPV4 does have consensus PKA phosphorylation sites and there is suggestive evidence in other cell types that TRPV4 is regulated by PKA-dependent phosphorylation. This should be discussed. The authors should also show whether TMEM16F knockdown reduces Forskolin-stimulated Ca signaling. The role of Forskolin on cell fusion should be discussed further – do the authors imply that the exposure of PS facilitates fusion?

We thank the reviewer for raising the interesting questions.

1. Forskolin-induced cAMP signaling is known to play a critical role in BeWo trophoblast cell differentiation and fusion. The predominant model is that forskolin-induced PKA activation and phosphorylation of ERK1/2 and p38 MAPK can upregulate Syncytins, the critical fusogenes in trophoblasts (34), to facilitate trophoblast fusion (35). As the reviewer pointed out, TRPV4 in trophoblasts is likely also phosphorylated downstream of PKA, which may alter TRPV4 function or surface expression after forskolin treatment. This will be an interesting direction for future investigations of TRPV4 in trophoblast biology. We have included corresponding discussion in the revised manuscript.

2. We apologize that we do not quite understand the rationale to test if TMEM16F knockdown reduces Forskolin-stimulated Ca signaling. We feel the results are likely difficult to interpret and may be irrelevant to the main conclusion of the manuscript. Please advise.

3. In addition to Syncytin-mediated trophoblast fusion, membrane remodeling also plays an essential role in trophoblast fusion. In our recently publication, we used both in vitro and in vivo approaches to show that TMEM16F-mediated PS exposure is indispensable for trophoblast fusion (36). Genetic ablation of TMEM16F not only prevents BeWo cell fusion in vitro, but also causes deficient formation of the Syncytin-2 layer in mouse placentas in vivo. It is still unclear the relationship between TMEM16F-and syncytin-mediated trophoblast fusion. This needs to be further investigated in the future.

13) More information is required to validate that identification of fused cells carries no subjective components, e.g. by showing the classification into fused vs non-fused for all cells in the shown representative images in Figure 6. While the TRPV4 antagonist GSK219 or siRNA reduce trophoblast fusion, they only reduce it by about 40%. The effect of TRPV4 siRNA, while statistically significant, shows a lot of scatter with significant overlap between control and siRNA data points. This suggests that other mechanisms also activate TMEM16F or that TMEM16F is not required for fusion. This should be addressed.

We appreciate the suggestions and comments.

1. We have updated previous Figure 6A, 6C (current Figure 7A, 7C). To facilitate the readers to distinguish the fused vs unfused cells, the nuclei of unfused cells are marked with white dotted circles and fused cells are marked with red dotted circles.

2. Our recent publication shows that TMEM16F is indispensable for trophoblast fusion (36). The detailed mechanism is still under investigation. In order to understand how TMEM16F controls trophoblast fusion, it is important to dissect how TMEM16F is activated under physiological conditions.

3. We agree with the reviewer. TRPV4 is very likely not the sole calcium source for TMEM16F activation. There must be other types of calcium-permeable channels that can provide calcium for TMEM16F activation. In supporting this hypothesis, a BioRxiv preprint recently reported that the gain-of-function mutant TRPC6 channel can enhance PS exposure in platelets, presumably through activation of TMEM16F (doi: https://doi.org/10.1101/2022.02.01.478727). This new finding further supports our conclusion that activation of TMEM16F by calcium-permeable channels can be a generalize mechanism in difference cells. We discuss this in our revised manuscript (line 239-242) “Our current study suggests that functional coupling between ca^2+^ permeable channels and TMEM16F can be a generalized mechanism for TMEM16F activation in different cell types. Ca^2+^ permeable channels other than TRPV4 may serve as ca^2+^ sources for TMEM16F activation. A recent study in platelets shows additional evidence to support this hypothesis. The authors demonstrated that a gain-of-function mutant TRPC6 channel can enhance PS exposure in platelets. As TRPC6 is a ca^2+^ channel and TMEM16F is the major ca^2+^-activated lipid scramblase, the enhanced PS exposure is likely through enhanced TRPC6-TMEM16F coupling. Future studies are needed to further test this hypothesis in different cell types.”

14) The authors must examine experimentally the potential contribution of other sources of calcium in BeWo and trophoblast cells, in particular CaV and TRPV6. Other TRPV channels are not studied because the Human Protein Atlas suggests expression is low, but the Human Protein Atlas is notoriously unreliable. Expression should be measured in the system used in this study. Ca-activated mobilization of internal Ca stores or other channels must also be considered.

We appreciate the reviewer’s suggestion and concerns.

1. We disagree with the reviewer’s comment on the reliability of the Human Protein Atlas. Some of their immunohistochemistry results may not be accurate, depending on the antibodies. However, their transcriptome results have been widely accepted and used (37-40) (more than 2,000 citations over the past few years). Based on our experience, their transcriptome data are reliable. The data we used in Figure S1 are from their single cell RNAseq database not from immunohistochemistry. Our immunostaining and functional tests of both TMEM16F (36) and TRPV4 (Figure 1D, S2C) validate the accuracy of their single cell RNAseq results. The high expression of *TRPV6* transcripts (Figure S1) is also consistent with previous reports (2, 3). In addition, we also searched other gene expression database including Genevestigator, and published human trophoblast single cell RNAseq results (41, 42). We found consistent gene expression profiles for TRPV channels.

2. In this study, we want to demonstrate a physiological mechanism of TMEM16F activation. Our main message is that TRPV4 is functionally expressed in human trophoblasts, where its opening can activate TMEM16F scramblase and help trophoblast fusion. To reach this conclusion, we have used TRPV4 specific agonist and antagonist, as well as gene silencing. We believe our conclusion is fully supported by our results. Screening all calcium channels including IP3Rs that mediate calcium store release in trophoblasts is highly important for future studies. However, we believe this request is beyond the scope of this study. We sincerely wish the reviewer would agree.

3. Nevertheless, we still conducted the following experiment and tried our best to address this question.

A) L-type CaV and TRPV6 channels are two major calcium channels previously reported to be expressed in trophoblasts (2, 3, 43, 44). Unfortunately, there is no TRPV6-specific agonist or antagonist. Therefore, we cannot use pharmacological tools to test its role in TMEM16F activation. However, due to its fast inactivation kinetics upon ca^2+^ entry (45), TRPV6 has been believed to be involved in trans-placental ca^2+^ transport (CaT) instead of sustained ca^2+^ signaling (46, 47). Based on this, we believe that it is less likely that TRPV6 acts as a major ca^2+^ source for TMEM16F activation in trophoblasts.

In order to investigate whether L-type CaV channels could contribute to the activation of TMEM16F in trophoblast and regulate trophoblast fusion, we applied nifedipine, an L-type CaV channel inhibitor, to BeWo cells (Author response image 2). We found that 1 μm of nifedipine has no effect on forskolin-induced BeWo cell fusion, suggesting that L-type Cav channels may not play an important role in trophoblast fusion.

**Author response image 2. sa2fig2:** Nifedipine (1 µM) has no effect on forskolin-induced BeWo cell fusion. (A) Representative images of control and nifedipine treated BeWo cells after 48-hour forskolin treatment. Nuclei and membranes were labelled with Hoechst (blue) and Di-8-ANEPPS (green), respectively. (B) Quantification of nifedipine effect on forskolin-induced BeWo cell fusion. Unpaired two-sided Student’s t-test. ns: not significant. Error bars indicate ± SEM. Each dot represents the average of fusion indexes of six random fields from one cover glass.

15) The findings suggesting that TMEM16F and TRPV4 are co-localized should be toned down. Additional evidence should be provided to conclude that this is the case. This could involve proximity labeling, higher-resolution co-localization using fluorescence microscopy or FRET, or showing a direct interaction between the two channels. The colocalization of TRPV4 and TMEM16F (Figure S6) appears good in the selected field containing one cell, but this needs quantification over a large population of cells. Further, the colocalization is only demonstrated in cells over-expressing TMEM16F, not in native cells.

We appreciate the suggestions.

1. As we explained in the manuscript, the available validated polyclonal TRPV4 and TMEM16F antibodies were raised in rabbits. They are not compatible to co-stain endogenous TRPV4 and TMEM16F.

2. We imaged immunofluorescence of endogenous TRPV4 and overexpressed TMEM16F-mCherry in a large population of BeWo cells (Author response image 3). Colocalization of TRPV4 and TMEM16F-mCherry on cell surface is still apparent.

**Author response image 3. sa2fig3:** TRPV4 and TMEM16F are spatially close to each other on BeWo cell membrane. Immunofluorescence of the endogenous TRPV4 (anti-TRPV4, red) and the heterologously expressed mCherry-tagged TMEM16F (anti-mCherry) in BeWo TMEM16F knockout cells.

3. We agree with the reviewer that the colocalization under confocal microscopy is not sufficient to support TRPV4-TMEM16F coupling within nm range. Additional evidence is needed to prove their functional coupling. This was the reason why we included the electrophysiology results in Figure 5, in which we utilized the different kinetics of BAPTA and EGTA in chelating ca^2+^ in nano- and micro-domains to estimate the spatial relationship between TRPV4 and TMEM16F. Utilization of the chalators’ different buffering kinetics to distinguish calcium micro- vs nano-domains is a widely used, quantitative biophysical method (12, 13). To our knowledge, the sensitivity of this method is not less than the few other methods suggested by the reviewer.

4. Our electrophysiological results in Figure 5 give a clear estimation of TRPV4 and TMEM16F spatial relationship. Co-immunolocalization or proximity ligation assay that is limited by the capability of the antibodies is no longer needed. To avoid misunderstanding, we have changed the “co-localized” to “spatial proximity” in the manuscript, and removed previous co-immunostaining results.

16) The writing could use improvement to make it flow better. There are missing articles and singular-plural mismatches that make the text hard to follow.

We apologize for this. The grammar has been thoroughly checked.

17) The introduction could be improved to make it easier for readers to follow. For those not expert in the field, the discussion of the authors' previous work, which sets up the logic for the present study, is difficult to understand. The last paragraph of the introduction goes well beyond the current manuscript and should be reined in.

We appreciate the suggestion. The introduction has been thoroughly revised.

Reviewer #1:TMEM16F non-selective channels and lipid scramblases play many biologically relevant roles involving cell fusion, including a function in trophoblast syncytialization. However, it is unclear which pathways are utilized under physiological conditions to supply TMEM16F channels with enough cytosolic calcium to trigger their activity. In the present manuscript, Yang Zhang and collaborators show that primary human trophoblast cells express TRPV4 channels, and that cell treatment with a TRPV4-selective agonist, GSK101, increases cytosolic calcium concentration and phosphatidylserine exposure on the outer leaflet of cells. Using a human choriocarcinoma trophoblast cell line BeWo, the authors investigate the mechanism further. The authors show that the TRPV4 agonist GSK101 increases cytosolic calcium in these cells based on calcium imaging experiments and enhances currents likely mediated by TRPV4 measured in patch-clamp recordings. GSK101 treatment also enhances Annexin V binding to the plasma membrane, suggesting increased scramblase TMEM16F activity, and results in delayed activation of voltage-dependent currents that resemble TMEM16F currents. Both Annexin V exposure and voltage-dependent current buildup are absent in TMEM16F-deficient BeWo cells exposed to GSK101, as well as in cells treated with two different TRPV4 siRNAs. Together these results suggest that calcium entry via GSK101-activated TRPV4 channels stimulates TMEM16F currents and lipid scrambling activity. In outside-out patches from BeWo cells, the authors show that BAPTA in the pipette prevents the buildup of TMEM16F currents when cells are exposed to extracellular calcium and GSK101, whereas the slower chelator EGTA does not. Together with the co-localization of TMEM16F and TRPV4 channels observed using fluorescence microscopy, the latter findings suggest that both proteins are in close proximity in the plasma membrane. Finally, the authors provide evidence suggesting that the presence of TRPV4 enhances Forskolin-mediated cell fusion of BeWo cells, because treatment with a TRPV4 antagonist of with TRPV4-directred siRNA reduces cell fusion.The possibility of a biologically relevant interaction between TRPV4 and TMEM16F in human trophoblast cells is novel and relevant. However, the biological relevance of the findings is not strongly established by the data provided in the manuscript. First, most of the findings are done in the BeWo cell line, and the experimental findings in actual human trophoblast cells are limited to showing that TRPV4 channels are expressed and that exposure of these cells to GSK101 induces calcium entry and Annexin V binding to the plasma membrane. The extent to which BeWo cells and primary human trophoblast cells are similar is not discussed. Further, a saturating concentration of a synthetic agonist is utilized to activate TRPV4 channels, but it remains unclear whether TRPV4 activity is stimulated under physiological conditions in native trophoblast cells, or whether the amount of calcium entry triggered by GSK101 reflects what occurs under physiological conditions. It remains possible that no biologically relevant connection between TMEM16F and TRPV4 channels exists, and that the findings reflect an abnormally high stimulation of TRPV4 channels leading to non-biological effects. These considerations are not reflected in the way the data and results are presented throughout the manuscript. No data is provided to support that TRPV4 channels play a role in trophoblast syncytialization in vivo.

We appreciate the criticisms from the reviewer, which are invaluable to guide our revision. We have conducted the following experiments to demonstrate the physiological function of TRPV4 and TRPV4-TMEM16F coupling in trophoblasts. Briefly, (1) 0.2 nM GSK101 stimulates transient ca^2+^ increase followed by local PS exposure (Figure 6), suggesting that physiologically relevant ca^2+^ level is sufficient to activate native TMEM16F in subcellular regions and implying the TRPV4-TMEM16F coupling happens in a spatiotemporal fashion under physiological conditions. (2) TRPV4-TMEM16F coupling happens in human primary trophoblast cells (Figure S6). (3) Pharmacological inhibition of TRPV4 in a novel human trophoblast organoid model suppresses trophoblast fusion (Figure 7E-F). Please see our response to Essential Revisions #1 and the revised manuscript for more detailed discussions.

In addition, the manuscript lacks several important controls as well as group-data quantitation of many key experiments (Figure 1I and J, Figure 3B, Figure 4A-C, Figure 5, Figure S2, Figure S3A, Figure S3B, Figure S4, Figure S5, Figure S6, Figure S7); whereas the time-course of calcium entry and Annexin V binding to WT and TMEM16F KO BeWo cells exposed to GSK101 is shown in Figure 2, no such quantitation is provided for identical experiments with cells treated with TRPV4-siRNA (Figure 4). No data is provided to show whether treatment of cells with TRPV4-siRNA reduces expression or activity of TMEM16F channels in BeWo cells, or to verify that this treatment indeed reduces the presence of TRPV4 channels, e.g. by showing reduced binding of the TRPV4-specific antibody to siRNA-treated cells or a western blot for TRPV4 channels localized at the plasma membrane. No control is provided to show whether exposure of outside-out patches to high extracellular calcium causes any TMEM16F current activation without exposure to GSK101 when long recordings as those in Figure 5 are carried out. As mentioned by the authors, BeWo cells have other calcium-permeable channels, including TRPV6 channels that are generally considered as constitutively active.

We apologize for missing the quantitative analysis for our imaging results. We have made corresponding corrections and conducted control experiments. Please see our response to Essential Revision #3 for more details.

In regards to the proximity of TMEM16F and TRPV4 channels establishing functionally relevant calcium microdomains, no conclusive evidence is provided. Both the fluorescence microscopy in Figure S6 (involving an over-expressed TRPV4 construct) and the comparison between fast and slow calcium chelators in Figure 5 constitute indirect evidence, which is not reflected in the authors analysis and discussion of their data throughput the manuscript.

We agree with the reviewer that the co-immunostaining results from previous Figure S6 cannot serve as conclusive evidence to demonstrate microdomain coupling between these two proteins (please note TMEM16F-mCherry not TRPV4 was overexpressed to overcome the incompatibility issue of the TRPV4 and TMEM16F antibodies). Therefore, we decided to remove the costaining results from the revised manuscript, despite the fact that we conducted more experiments to validate the specificity of the TRPV4 antibody (Figure S2) and to image a large population of cells (Response Figure #4).

On the other hand, using different concentrations of ca^2+^ chelators with fast and slow kinetics has been a widely used approach to estimate the relative distance between a ca^2+^ channel and a ca^2+^ sensing channel (12, 13). Utilizing the high sensitivity of patch clamp technique, this approach can distinguish the relative distance within 100 nm. Therefore, we believe the distance estimated using this approach should have comparable, if not better, accuracy to other imaging or biochemical approaches. Please see our response to Essential Revisions #15 for more details. We have included more discussions in the revised manuscript.

Finally, the results regarding cell fusion do not appear to be very robust. First, cell fusion is induced by Forskolin, which appears to suggest that TRPV4-mediated calcium entry and TMEM16F lipid scrambling are only complementary mechanisms. It is unclear from the discussion in the manuscript whether the activity of these two channels would be expected to promote cell fusion under physiological conditions without any added Forskolin. Second, there seems to be limited validation from the images in Figure 6 to support a robust effect of TRPV4-knock down or inhibition on cell fusion: the criteria for defining fused vs non-fused cells are not very clear and appear to have some degree of subjectivity, and in addition there is lots of spread in the group data and the differences between treatments are not very pronounced.

We appreciate the questions and concerns.

1) We apologize for the confusion. Cell fusion is a complicated cellular process, which involves in the coordinated actions of different proteins and membrane remodeling. Despite the extensive investigations in recent years, detailed cell fusion mechanisms are still largely unclear. We hope that our current finding can shine light on understanding how ca^2+^ and the ca^2+^ activated lipid scramblase contribute to membrane remolding during cell fusion. Future investigations are needed to examine the relationship between the known fusogenes and scramblase-mediated membrane remodeling during cell fusion.

2) Forskolin-induced cAMP signaling is known to play a critical role in BeWo trophoblast cell differentiation and fusion. Nevertheless, human primary trophoblasts can spontaneously fuse during culturing without requiring forskolin stimulation. Unfortunately, the COVID restrictions prevented us to obtain human placentas to further examine TRPV4-TMEM16F coupling using human primary trophoblasts. Instead, we utilized human trophoblast organoids, which are readily available from Dr. Carolyn Coyne’s lab at Duke. Similar to primary trophoblasts, the organoid trophoblasts can also spontaneously fuse without forskolin (Figure 7F, DMSO control). However, pharmacological inhibition of TRPV4 can largely abolish the organoids to express *CSH1* and *CGB*, two important markers for syncytiotrophoblasts. This experiment thus further support our observations of the fusion defects using TRPV4 inhibitor or siRNAs in BeWo cells (Figure 7A-D).

3) The definition of fused and nonfused cells and fusion index quantification have been described in the method section of the manuscript. We have used this same method and criteria to quantify BeWo cell fusion in two publications (36, 48). Cell fusion is a dynamic process. To reduce bias, we randomly picked six fields from a coverglass and averaged all the fusion indexes to obtain one data point shown in Figure 7B and 7D. To avoid the instances of cell division, cells with two nuclei were not considered as fused cells. We also updated previous Figure 6A, 6C (current Figure 7A, 7C) in order to facilitate the readers to distinguish the fused vs unfused cells. The nuclei of unfused cells are marked with white dotted circles and fused cells are marked with red dotted circles.

1) The biological relevance for the possible interaction between TRPV4 and TMEM16F channels needs to be toned down throughout the manuscript. The authors should provide additional data in support of a biological relevance for their findings. It would be expected that TRPV4-KO mice have phenotypes associated with trophoblast fusion deficiencies if the association between TRPV4 and TMEM16F channels is as relevant as suggested. The authors could also use GSK101 concentrations that result in cytosolic calcium elevations that closely resemble those that would occur in vivo, or at least provide a discussion justifying the concentration of GSK101 that was used.

Please see our response to Essential revisions # 5 for more detailed discussion.

2) The emphasis on the findings pertaining to human trophoblasts should also be toned down throughout, as most of the experiments were done in a cell line and not in native cells.

We have conducted experiments using human primary trophoblasts and human trophoblast organoids. Please see our response to Essential revisions # 2 for more detailed discussion.

3) Group data quantitation should be provided for all relevant figures (Figure 1I and J, Figure 3B, Figure 4A-C, Figure 5, Figure S2, Figure S3A, Figure S3B, Figure S4, Figure S5, Figure S6, Figure S7).

Quantifications have been included in the revised manuscript. Please see our response to Essential revisions # 3 for more details.

4) The authors should provide a control showing that the siRNA does not directly affect expression or activity of TMEM16F channels. This could be done by performing patch-clamp recordings with added calcium to stimulate these channels, and by showing that treatment with a calcium ionophore results in robust Annexin V binding to the outer leaflet in siRNA-treated cells.

This has been done and included as Figure S10. Please see our response to Essential revisions # 9 for more details.

5) The authors should further validate their TRPV4-knockdown by showing a decreased channel expression in the plasma membrane with immunofluorescence or western blots.

This has been done and included as Figures S4 and S5. Please see our response to Essential revisions # 10 for more details.

6) The authors should perform outside-out experiments in the presence of 2.5mM extracellular calcium and 0.2 mM EGTA in the pipette without GSK101 exposure, as in Figure 5, and show that no TMEM16F current buildup occurs after long wait times.

This has been done and included as Response Figure 2. Please see our response to Essential revisions # 11 for more details.

7) The findings suggesting that TMEM16F and TRPV4 are co-localized should be toned down. Additional evidence should be provided to conclude that this is the case. This could involve proximity labeling, higher-resolution co-localization using fluorescence microscopy or FRET, or showing a direct interaction between the two channels.

We have removed the description of TMEM16F-TRPV4 colocalization from the revised manuscript. Proximity labeling, higher-resolution co-localization or FRET was not performed. Please see our response to Essential revisions # 15 for our justification.

8) The role of Forskolin, TRPV4 calcium entry and TMEM16F activity on cell fusion should be discussed further. Additionally, more information is required to validate that identification of fused cells carries no subjective components, e.g. by showing the classification into fused vs non-fused for all cells in the shown representative images in Figure 6.

For our quantification of fusion indexes, please see our response to Reviewer 1’s Public Review, the last comment.

We also updated previous Figure 6A, 6C (current Figure 7A, 7C) in order to facilitate the readers to distinguish the fused vs unfused cells. The nuclei of unfused cells are marked with white dotted circles and fused cells are marked with red dotted circles.

Reviewer #2:Zhang et al. have investigated the question of how the phospholipid scramblase TMEM16F is activated physiologically. in vitro in the lab, TMEM16F is typically activated by the Ca ionophore ionomycin or by patch clamp with high intracellular Ca concentrations, but physiological activation of TMEM16F is poorly understood. Using a combination of imaging, patch clamp, and scramblase assays, Zhang et al. show that TRPV4 and TMEM16F are expressed in human trophoblasts and in the BeWo trophoblast cell line. Increases in cytosolic Ca caused by activation of TRPV4 with a specific agonist GSK1016790A activates scrambling, TMEM16F currents, and trophoblast cell fusion.Strengths.These studies show that:(1) TRPV4 and TMEM16F are expressed in both primary human trophoblasts and the trophoblast cell line WeBo. This is shown by RT-PCR, immunofluorescence, and functional assays.(2) Ca influx via TRPV4 can activate TMEM16F in a native cell. This is an important observation because it demonstrates that TMEM16F can be activated by Ca influx mediated by an endogenous channel rather than by a Ca ionophore. Furthermore, it provides another tool that can be used to control TMEM16F experimentally.(3) TRPV4 and TMEM16F exist in a local Ca microdomain in the cell membrane such that Ca influx through TRPV4 activates TMEM16F locally. This provides additional evidence for specific coupling between these two ion channels.(4) Activation of TMEM16F via TRPV4-mediated Ca influx stimulates downstream processes leading to fusion of trophoblasts to form syncytiotrophoblasts in vitro. These authors had previously shown that TMEM16F is involved in trophoblast fusion, and this paper identifies the source of activating Ca that had not previously been known.

We appreciate the reviewer’s positive comments.

Weaknesses.Although the paper has significant overall strengths, the paper has several major weaknesses.(1) Although the paper shows that TRPV4 and TMEM16F are functionally coupled when TRPV4 is activated pharmacologically, the paper does not show that physiological activation of TRPV4 can activate TMEM16F. The experiments that show functional coupling between these channels utilize the TRPV4 agonist GSK101 and forskolin. These pharmacological agents are likely to activate TRPV4 to a greater extent than a physiological stimulus. Also, the mechanism of forskolin is not clear. I am not aware that it has been shown that cAMP-dependent processes activate TRPV4 in trophoblasts, although TRPV4 does have consensus PKA phosphorylation sites and there is suggestive evidence in other cell types that TRPV4 is regulated by PKA-dependent phosphorylation.

We appreciate the important comments.

1. TRPV4 is a polymodal sensor that can be activated by various stimulations including temperature, osmolarity, endocannabinoids, etc. In addition, there are likely other endogenous trophoblast proteins can also sense these stimulations. Therefore, it will be challenging to specifically dissect out the effects of physiological activation of TRPV4. Species-different expression of Tprv4 in the mouse placenta further prevents us to dissect its physiological functions in trophoblast fusion in vivo. At this moment, the highly specific and potent TRPV4 agonists and antagonists seems to be the best solution to explore its physiological roles in trophoblast fusion. Our new experiments using ultra low GSK101 (0.2 nM) to mimic physiological TRPV4 activation show that TRPV4-TMEM16F coupling can happen under physiological conditions (Figure 6).

2. Please see our response to Essential Revisions #12 for detailed discussions for forskolin effects.

(2) Other sources of Ca are not considered. CaV and TRPV6, which have been reported to be expressed in trophoblasts, are not studied because they are thought to be unlikely to play a role, but these channels are not examined experimentally. Other TRPV channels are not studied because the Human Protein Atlas suggests expression is low, but the Human Protein Atlas is notoriously unreliable. Expression should be measured in the system used in this study. Ca-activated mobilization of internal Ca stores or other channels is not considered.

Please see our response to Essential Revision #14.

Figure 2A shows that TMEM16F KO reduces Ca influx significantly. These authors have previously argued that TMEM16F is a Ca-permeable non-selective cation channel, and a plausible alternative explanation is that Ca entering via TRPV4 activates Ca influx via TMEM16F and that scrambling follows this.

We appreciate the reviewer’s thoughtful suggestion on this phenomenon. Please see our response to Essential Revisions #5 for detailed discussion.

While TMEM16F knockdown reduces GSK101-stimulated Ca signaling, the effect of TMEM16F knockdown on forskolin-stimulated Ca signaling is not shown. Also, while the TRPV4 antagonist GSK219 or siRNA reduce trophoblast fusion, they only reduce it by about 40%. The effect of TRPV4 siRNA, while statistically significant, shows a lot of scatter with significant overlap between control and siRNA data points. This suggests that other mechanisms also activate TMEM16F or that TMEM16F is not required for fusion.

Thanks for the great comments.

Based on our limited understanding on TMEM16F biology, the lipid scramblase can play multifaceted roles in a cell, owning to its capability to alter phospholipid cross-bilayer distribution and change membrane environment. Our understanding what it can do is still in the primitive stage, including its effects on forskolin-stimulated Ca signaling. Because forskolin-stimulated Ca signaling can be directly or indirectly affected by TMEM16F knockdown (sorry, we used CRISPR-Cas9 TMEM16F knockout cells not knockdown in this study), we think it will be challenging to clearly dissect it out within a short period of time. In addition, this is beyond the scope of this study. We hope the reviewer would agree.

For our response to the remaining comments, please see our discussion in Response to Essential Revisions # 13.

The authors do not address the question whether the coupling of TRPV4 to TMEM16F is a general phenomenon or specific to trophoblasts.

Our in vitro experiments demonstrated that the coupling between TRPV4 and TMEM16F plays an important role in mediating trophoblast fusion. Although this particular coupling in trophoblasts could be a cell type specific phenomenon that requires the co-expression of TRPV4 and TMEM16F, we believe that the coupling between ca^2+^ permeable channels (including but not limited to TRPV4) and TMEM16F should be a general principle for ca^2+^-dependent activation of TMEM16F scramblases under physiological conditions. Future studies are needed to examine this hypothesis in different cell types. This has been included in Response to Essential Revisions #1. We have also included this in our Discussion section.

(3) While overall, the experimental design and execution is excellent, some experiments need additional controls, statistical analysis, or explanation.a. Functional assays are convincing that TRPV4 is expressed in trophoblasts, but the immunofluorescence is ambiguous. Figure 1D does not show TRPV4 convincingly on the membrane. Figure 1E needs a counterstain to show the location of the membrane and quantification of overlap between a membrane marker and TRPV4. The validation of the antibody in Figure S2 should also show staining in untransfected cells. A western blot would also be helpful.

We appreciate all the suggestions. We have repeated TRPV4 immunofluorescence for human placental villi with better resolution. In addition to cytosolic expression, TRPV4 expression on the apical side of the syncytiotrophoblasts is evident in the enlarged view of the revised Figure 1D.

Previous Figure 1E (TRPV4 staining in BeWo cells) has been moved to Figure S2C with counterstain of a membrane marker FM1-43.

New Figure S2A-B show the validation of the TRPV4 antibody.

b. The colocalization of TRPV4 and TMEM16F (Figure S6) appears good in the selected field containing one cell, but this needs quantification over a large population of cells. Further, the colocalization is only demonstrated in cells over-expressing TMEM16F, not in native cells.

Quantifications have been done on a large population of cells (Response Figure 3). We didn’t co-stain TRPV4 and TMEM16F in native cells because both antibodies are raised in rabbits. Therefore, we had to overexpress a mCherry-tagged TMEM16F to do the co-staining. As we explained in Response to Essential Revisions #15, we decided to rely on our patch clamp recording under different ca^2+^ chelators instead of immunostaining to demonstrate TRPV4-TMEM16F coupling. Please see our discussion in Response to Essential Revisions #15 for more details.

c. Figure 5 needs statistical analysis.

Statistics has been included in revised Figure 5C-D.

d. The dynamics of Ca signaling needs further explanation. Ca influx is dependent on voltage: at +100mV Ca influx should be negligible because Vm is approaching E(Ca). But, TMEM16F currents seem to be independent of the Ca driving force. TMEM16F currents increase with time in response to voltage steps to +100 mV, despite the fact that Ca influx is decreasing at this voltage. The amplitude of the currents are comparable to the currents elicited by 100 μm Ca in whole-cell patch clamp (compare Figure 4D and Figure S5), so this suggests that [Ca] at the TMEM16F channel remains close to 100 μm even at +100 mV. Furthermore, in Figure 5B panel 2, the amplitude of the current is independent of the duration of the prepulse at -100 mV. Maybe there is sufficient Ca influx at the holding potential to flood the cell, overwhelm native efflux mechanisms, and saturate Ca buffers, but if that is the case, this would invalidate the Ca microdomain concept developed in Figure 5.

We apologize that we did not clearly describe the holding voltage for the figures mentioned. The reviewer is correct. The holding voltage of -60 mV in the constant presence of GSK101 favors ca^2+^ influx and accumulation under whole-cell configuration. This is likely the reason why the GSK101 stimulated TMEM16F current is comparable with direct ca^2+^ activated current. For Figure 5B, #2, we used outside-out patch, which eliminates the long delay of TMEM16F activation under whole cell configuration with unclear mechanism. The volume of pipette solution is enormous compared with the amount of ca^2+^ influx, and there is no physical barrier to prevent ca^2+^ to diffuse into the bulk pipette solution. Therefore, ca^2+^ entry through a TRPV4 channel in an outside out membrane patch is expected to readily diffuse away from the mouth of the channel without sufficient accumulation, even in 0.2 mM EGTA (panel 2). Please see our response to Essential Revisions #6B for details.

Reviewer #3:– The data, as presented, do not justify the conclusions that TRPV4 and TMEM16F are functionally coupled and that they colocalize in microdomains. This is not to say the data are not very interesting and consistent with these conclusions. Rather, the authors would be better served being more conservative in their interpretation.

We appreciated the comments. We have provided multiple lines of evidence in the revised manuscript to support that TRPV4 and TMEM16F are functionally coupled, and their coupling can happen within microdomains in trophoblasts, including imaging and electrophysiology assays, pharmacological tools and gene silencing. We have thoroughly revised the manuscript to avoid over interpretation.

– The result that a delayed cation conductance develops minutes after activation of TRPV4 is intriguing. For this result to be interpreted as activation of the ion conducting channel of TMEM16F, the authors would need to demonstrate a few things:– It has been previously shown that the scramblase activity and ion conducing pore of TMEM16F are activated by the same ca^2+^ concentrations. The authors should show that the kinetics of the candidate TMEM16F currents occur simultaneously with activation of the scramblase activity.

We have included a new experiment using patch clamp-lipid scrambling fluorometry to monitor TMEM16F channel and scramblase activities simultaneously upon GSK101 stimulation (Figure S8). Please see Response to Essential Revision #4 for details.

– It is not clear that, in the presence of even low concentrations of EGTA, ca^2+^ through TRPV4 in excised patches could accumulate at the surface of the membrane at the high concentrations required. Indeed, the inward currents in Figure 5, as an example, are so small that, even if all the current were ca^2+^ it seems unlikely that local ca^2+^ at sufficiently high concentrations would occur. Using known information on the binding kinetics of the chelators used, the size of the currents measured, and the permeability of ca^2+^ relative to other ions, the authors could model the expected ca^2+^ kinetics in their patches to demonstrate whether the >20 µM ca^2+^ concentrations would develop as expected. There may be other ways the authors could demonstrate the actual ca^2+^ concentrations achieved on the outside of their patches.

Please see our response to Essential Revisions #6A for details.

– The ca^2+^-imaging data, using a dye with a KD = 1.2 µM, are overinterpreted. TMEM16F is not active even at 20 µM ca^2+^ – it requires about 200 µM to be active. The authors should be more conservative in how they discuss their data given the affinities of their dye. This argument applies to the EGTA and BAPTA data.

We appreciate the reviewer asked this important question. Please see our Response to Essential Revisions #7 for more detailed discussion. In summary, our previous and current results together with the previous report in Ba/Fe cells (6) suggest that ca^2+^ required to activate TMEM16F under physiological conditions should be lower than the estimated >20 µM ca^2+^ range based on whole-cell patch clamp recording of TMEM16F current. It is very likely this invasive method disrupts native cellular environment (as evidenced by the abnormally long lag time), leading to underestimated ca^2+^ sensitivity for TMEM16F activation. In addition, the ca^2+^ dye was only used to monitor ca^2+^ increase after TRPV4 activation in this study. We did not intend to use the fluorescence intensity to quantify the exact ca^2+^ concentration. EGTA and BAPTA can distinguish ca^2+^ microdomains and nanodomains with wide range of ca^2+^ concentrations (12, 13). We believe that TMEM16F’s ca^2+^ sensitivity falls within this range, as evidenced by the differential TMEM16F activation in the presence of different concentrations of EGTA and BAPTA (Figure 5),

– The authors characterize the currents they attribute to TRPV4 as not having activation/deactivation kinetics, in contrast to the late-developing currents, which do show such kinetics. The pronounced outward rectification of the currents attributed to TRPV4 mean that they have very pronounced gating kinetics. They are just much faster than those of the late-developing currents.

We appreciate the reviewer pointing out the differences in the activation/deactivation kinetics for the early-developing current (TRPV4) and late-developing current (TMEM16F). Yes. GSK101-induced outward rectifying current is indeed fast in kinetics (32). TMEM16F current, on the other hand, shows much slower kinetics even in the presence of high ca^2+^ (0.1 to 1 mM) (8, 10). When we knockdown TRPV4 in BeWo cells, we observed significant reduction of this early-developed current with fast kinetic (Figure S5C-E), suggesting that this current is indeed mediated by TRPV4. We have clarified the kinetic differences between TRPV4 and TMEM16F channels in our revised manuscript. (Response to Essential Revisions #8)

– The imaging data are strictly qualitative and should be either subject to rigorous quantification with statistical analysis or eliminated. They add little to the present manuscript.

We have added quantifications to the imaging results in the revised manuscript.

– Basic co-localization of two proteins in the plasma membrane on standard visible-light microscopy is not valid.

We agree. We have removed the results using immune-colocalization from the revised manuscript. Instead, we estimated the spatial relationship between TRPV4 and TMEM16F using patch clamp recording in the presence of different concentrations of EGTA and BAPTA (Figure 5). Please see our response to Essential Revisions # 15 and our revised manuscript for more details.

– The writing could use improvement to make it flow better. There are missing articles and singular-plural mismatches that make the text hard to follow.

We have thoroughly checked grammar issues in the manuscript (response to Essential Revisions #16).

– The introduction could be improved to make it easier for readers to follow. For those not expert in the field, the discussion of the authors' previous work, which sets up the logic for the present study, is difficult to understand. The last paragraph of the introduction goes well beyond the current manuscript and should be reined in.

Please see our response to Essential Revision #17 for details.

– Why is the ca^2+^ influx in KO BeWo cells much smaller than the ca^2+^ influx in wt BeWo cells?

Please see our response to Essential Revision #5 for details.

[Editors’ note: what follows is the authors’ response to the second round of review.]

Essential revisions:1) The biological significance of the TRPV4-TMEM16F interaction has not been unequivocally established in the context of native human trophoblasts. This should be reflected in the way some of the conclusions are presented throughout the manuscript, by toning down certain statements, especially those related to a 'key role of TRPV4 channels in human trophoblast fusion'. Solid evidence is provided that activation of TRPV4 channels has the capacity to activate TMEM16F channels, and could potentially play an important role in trophoblast fusion, but no experimental evidence is provided to show that TRPV4 channel activity is indeed required for this process in native human cells.The term 'human trophoblast' is used at multiple points throughout the manuscript, including in the titles of figures and Results sub-sections. The term BeWo cells (or immortalized trophoblast cell line) should be utilized instead; most of the key experiments and controls were done in these cells that could exhibit biologically relevant differences compared to non-immortalized human trophoblast cells. Particularly because the expression of TRPV4 channels in human trophoblasts seems to be low (less than 10-fold as compared to TMEM16F, Figure 1B), and other calcium channels are expressed by these cells, it is best to be cautious when drawing conclusions about the specific importance of TRPV4 channels in vivo based almost exclusively in data from immortalized cells.

We appreciate the additional comments. Here are our responses.

a) We agree with the reviewer. TRPV4 is likely only one of several calcium channels that activate TMEM16F and modulate trophoblast fusion. We focused on TRPV4, a previously unknown calcium channel in trophoblasts, to demonstrate the general principle for TMEM16F activation under physiological conditions. We have carefully checked and revised the manuscript to make sure that we did not overclaim.

b) The reason why we emphasize human trophoblasts is to highlight the species-dependent expression of TRPV4 in human and mice. As mouse trophoblasts do not express Trpv4 (Figure 7—figure supplement), we had to use in vitro human trophoblast models. In addition to use BeWo cell line, we also used human primary trophoblasts and human trophoblast organoids to further validate our findings.

To be more accurate, we have thoroughly checked our manuscript to make sure that we clearly defined which human cells we used (BeWo cell, human primary trophoblasts, human trophoblast organoids) in each conditions, including subsection titles and figure legends. Please see our revised manuscript for details.

2) The connection between the interplay of TMEM16F and TRPV4 in placental organoids is quite weak. Downregulation of genes is a poor readout to make the physiological inferences the authors would like to make. Are there any morphological effects on GSK219 -treated organoids, like those seen in the BeWo cells? Is it possible to measure PS exposure in the organoids? The experiment with the organoids would constitute stronger evidence for the role of TRPV4 channels in trophoblast fusion in vivo than the experiment with BeWo cells in Figure 7A-D if direct evidence were provided that inhibition of TRPV4 channels reduces cell fusion, and results in increased PS exposure. It is unclear why cell fusion was not quantified directly from the organoids, and to what extent the markers that were analyzed are indicative of cell fusion. This needs to be addressed. Despite claims to the contrary, the authors provide no evidence that inhibition of TRPV4 hinders trophoblast fusion in the organoid model. Further, the downregulation of fusogenic genes might point to a mechanism different from the TMEM16F mediated one that is proposed here. Without direct evidence connecting the data in cell lines to that in the organoid, I believe the manuscript would be strengthened by the removal of this data.

We thank the reviewer for the suggestions. The following are our further clarifications.

a) As we described in the manuscript, *CSH1* and *CGB* are widely used placental trophoblast fusion marker genes, which are indicative of cell fusion (1-7). *CSH1* and *CGB* encode human placental lactogen (hPL) and hormone-specific β-subunit of human chorionic gonadotropin (βhCG), respectively. Both proteins are specifically synthesized by syncytiotrophoblasts. It is a consensus in the field that the extent of *CSH1* and *CGB* expression directly reports trophoblast fusion efficiency. In fact, commercial pregnancy strip tests detect hCG secreted from syncytiotrophoblasts to report pregnancy. Pharmacological inhibition of TRPV4 by GSK219 dramatically reduces the expression of *CSH1* and *CGB* (Figure 7F). Our result thus strongly supports that inhibiting TRPV4 hinders trophoblast fusion in the human trophoblast organoids.

b) Human trophoblast organoids are newly developed research tools. There is no good way available to quantify fusion index in the 3D system thus far. We tried to use anti-CD46 and anti-CD71 to label the fused cells. Unfortunately, the CD46 and CD71 antibodies also labeled cytotrophoblasts in the human placenta organoids. We also tried to use the membrane labeling method that we developed for quantifying BeWo cell fusion (8, 9). Unfortunately, the membrane marker failed to label the membrane of human placenta organoids. Therefore, we had to use the syncytialization markers *CSH1* and *CGB* to quantify the fusion efficiency in the human placenta organoids. Please see our response a) for details. A reliable imaging method to quantify trophoblast fusion in organoids needs to developed in future studies.

c) Due to the detection limit using fluorescently labeled PS-binding proteins including Annexin V, we have not been able to successfully capture PS exposure during cell-cell fusion even in 2D BeWo cell culture. Measuring PS exposure in the organoids seems even more challenging. We wish a more sensitive method to monitor PS exposure can be developed in the near future so that we can detect PS exposure during fusion in real time.

d) *SYN1* is the most important fusogene for trophoblast fusion. However, we did not observe *SYN1* downregulation as the reviewer suspected. Instead, we even observed slight upregulation of *SYN1* expression in GSK219-treated organoids (Figure 7F). This paradoxical finding further supports our previous speculation that TMEM16F-mediated PS exposure indeed plays a critical role in controlling trophoblast fusion, likely independent of the syncytin fusogenes (9). Our current study thus adds new insights in understanding trophoblast fusion. Detailed mechanism needs to be further investigated.

Based on these points, we believe that keeping the human trophoblast organoid results seems appropriate. We hope the reviewer would agree.

3) Calcium entry into TRPV4 siRNA-treated cells caused by saturating GSK101 (Figure S5A and B) appears to be comparable, if not higher, than in WT BeWO cells stimulated with 0.2 nM GSK101 (Figure 6A and B), yet transient and localized Annexin binding is only observed in the latter but not in the former experiments. This is confusing and needs to be discussed and explained, as it is unclear whether local and transient Annexin binding was looked at in other experiments where it might have also been observed.

We thank the reviewer for the comments.

Although the calcium levels are comparable in siRNA-treated cells with high GSK101 (Figure 1— figure supplement 5A) and in WT cells treated with low GSK101 (Figure 6A-B), there is a fundamental difference between these two conditions. In the siRNA treated cells, TRPV4 expression was largely suppressed, resulting in global reduction of channel density on cell surface. It is likely high GSK101 can hardly induce high calcium in spatially restricted regions to activate local TMEM16F scramblase. However, for siRNA untreated BeWo cells, TRPV4 expression and its density in some local domains is intact. Low GSK101 (0.2 nM) is likely still able to trigger local calcium that is high enough to induce transient TMEM16F activation in microdomain, as we shown in Figure 6. We have revised the manuscript accordingly to make this point clear (Line230-239).

4) The authors claim that at low activator concentrations PS exposure is transient. Looking at the representative images shown in Fig6, the transient PS exposure appears to revert in one location, but in the other areas, it persists. Is this just a matter of 'ending the experiment too soon', or does this represent local but irreversible PS exposure? At present, the writing indicates that all PS exposure is reversible but the data does not seem to agree. Quantification of the proportion of temporary vs persistent areas of PS exposure is important.

We appreciate the reviewer’s careful examination. Please look at the pre-GSK101 image at time 0 (Figure 6A and the supplementary video 3), the seemingly ‘persistent’ PS positive spots were already there before TRPV4 activation. At 1 o’clock and 3 o’clock directions far away from the cell body, the AnV spots were due to PS positive debris/microparticles in the cell culture. At 6 o’clock direction, there were two PS positive spots, too. They were loosely attached to the cell surface, but not part of the cell. These PS positive debris/microparticles are very common in cell culture when imaging with fluorescence AnV at high resolution. Different from these “persistent” ‘artifacts’, the AnV binding signal at the 9 o’clock direction was not present at time 0. It transiently appeared after GSK101 stimulation, and then rapidly disappeared. This was ‘real’ calcium-induced lipid scrambling. Therefore, our observation using low GSK101 is consistent with the balance between lipid scramblases and flippases under physiological conditions. When local calcium increases, transient TMEM16F activation will lead to temporary collapse of membrane lipid asymmetry in some subcellular localizations. When calcium level fades down, ATP-dependent flippases will restore the lipid asymmetry and transport PS back into the cytosolic leaflet, resulting in AnV unbinding from cell surface. We believe the reversible, spatiotemporal lipid scrambling activity is a reasonable explanation for understanding TMEM16F’s physiological functions.

To avoid confusing the readers, we have labeled the ‘persistent’ PS positive debris/microparticles in the revised Figure 6 and add the descriptions in the figure legends.

5) In most key figures involving fluorescence imaging only one or two cells are displayed. Although quantification is provided for more than one cell, these numbers still seem small compared to the total number of cells that can be imaged on a coverslip of in a well with seeded cells. It would strengthen the manuscript and provide a more solid representation of the findings if images were provided where a population of cells can be observed, all having similar phenotypes to the representative case that is currently shown. Otherwise, it is unclear if only cells showing the most marked phenotypes were analyzed and displayed.

Showing high magnification images in the manuscript is solely for demonstration purpose so that the reader can see more details. We have also included statistics in our resubmission. To address the concern, Author response image 4 shows a lower magnification image with a population of cells upon GSK101 stimulation. In this figure, 20 nM GSK101 triggers ca^2+^ influx and PS exposure (labeled by AnV) in all WT BeWo cells but fails to trigger calcium increase in TRPV4 siRNA2 treated BeWo cells.

**Author response image 4. sa2fig4:** Ca^2+^ influx through TRPV4 activates TMEM16F scramblase (left) and siRNA knockdown of TRPV4 (right) abolishes GSK101-induced ca^2+^ influx and subsequent TMEM16F CaPLSase activation.

6) In the response to reviewers and in the manuscript text it is implied that calcium sensitivity of TMEM16F channels is underestimated in whole-cell recordings because that experimental configuration is disruptive to cellular conditions. On the other hand, results from excised patches, which are even more disruptive to the natural conditions of cells, appear to yield higher sensitivity to calcium. Furthermore, the highest sensitivity is observed with calcium imaging, whereas pointed out in the first review, the affinity of the sensor for calcium is a determinant of what is observed. Because identifying a calcium source for TMEM16F channels that enables their activity is central to this manuscript, more clarification is required about the calcium sensitivity of TMEM16F channels. Specifically, it must be clearly stated whether it is speculated that cellular factors contribute to increasing the sensitivity of TMEM16F channels to calcium, or if cellular structures that can get disrupted in the whole cell configuration contribute to the formation of very small calcium microdomains where the concentration of the cation can increase to sufficient levels that activate TMEM16F channels. It would be important to mention here the work by the Dutzler lab (Alvadia et al., ELife, 2019) which showed that purified TMEM16F is activated by ca^2+^ with an EC50 ~1 uM.

We thank the reviewers’ the further comments. As we explained in our previous response letter, the discrepancy between low calcium sensitivity reported in whole cell recording and higher calcium sensitivity measured in excised patches and fluorescence imaging assay for lipid scrambling is an important yet unsolved question in the field. Regardless of the reason for this discrepancy (which is not the task of this manuscript), we believe that it is very likely that the calcium sensitivity estimated by whole cell recording has been dramatically underestimated. Our reasoning is listed below.

(1) Under physiological conditions, 20-200 μm sustained global calcium elevation barely can happen in a healthy cell. (2) Calcium ionophore induced calcium increase in most of lipid scrambling assays used in different labs should be below 20-200 uM. (3) Now we show that 0.2 nM GSK101 can very mildly increase intracellular calcium and transiently induced PS exposure in BeWo cells. As the reviewer pointed out, the calcium dye we used has kd of 1.2 uM. Therefore, calcium increase under this condition should be way lower than 20-200 uM. (4) In addition to our own inside out patch measure of EC50 of 13.6 μm for heterologously expressed TMEM16F current and 5.1 μm for endogenous TMEM16F in megakaryocytes at +60 mV (10), the recent work by the Dutzler lab (11) also showed that purified TMEM16F is activated by ca^2+^ with an EC50 ~1 uM. Therefore, the exact calcium sensitivity for TMEM16F channels and lipid scramblases varies depending on the methods and conditions.

We revised lines 297-310 to further clarify.

Reviewer #1:Many of the concerns have been addressed in the revised version of the manuscript, but I consider that there are still some issues that need to be resolved before publication, as well as some suggestions that can further strengthen the manuscript.1) The biological significance of the TRPV4-TMEM16F interaction has not been unequivocally established in the context of native human trophoblasts. This should be reflected in the way some of the conclusions are presented throughout the manuscript, by toning down certain statements, especially those related to a 'key role of TRPV4 channels in human trophoblast fusion'. Solid evidence is provided that activation of TRPV4 channels has the capacity to activate TMEM16F channels, and could potentially play an important role in trophoblast fusion, but no experimental evidence is provided to show that TRPV4 channel activity is indeed required for this process in native human cells.The term 'human trophoblast' is used at multiple points throughout the manuscript, including in the titles of figures and Results sub-sections. The term BeWo cells (or immortalized trophoblast cell line) should be utilized instead; most of the key experiments and controls were done in these cells that could exhibit biologically relevant differences compared to non-immortalized human trophoblast cells. Particularly because the expression of TRPV4 channels in human trophoblasts seems to be low (less than 10-fold as compared to TMEM16F, Figure 1B), and other calcium channels are expressed by these cells, it is best to be cautious when drawing conclusions about the specific importance of TRPV4 channels in vivo based almost exclusively in data from immortalized cells.

We appreciate the additional comments. Here are our responses.

a) We agree with the reviewer. TRPV4 is likely only one of several calcium channels that activate TMEM16F and modulate trophoblast fusion. We focused on TRPV4, a previously unknown calcium channel in trophoblasts, to demonstrate the general principle for TMEM16F activation under physiological conditions. We have carefully checked and revised the manuscript to make sure that we did not overclaim.

b) The reason why we emphasize human trophoblasts is to highlight the species-dependent expression of TRPV4 in human and mice. As mouse trophoblasts do not express Trpv4 (Figure 7—figure supplement 1), we had to use in vitro human trophoblast models. In addition to use BeWo cell line, we also used human primary trophoblasts and human trophoblast organoids to further validate our findings.

To be more accurate, we have thoroughly checked our manuscript to make sure that we clearly defined which human cells we used (BeWo cell, human primary trophoblasts, human trophoblast organoids) in every conditions, including subsection titles and figure legends. Please see our revised manuscript for details.

2) The experiment with the organoids would constitute stronger evidence for the role of TRPV4 channels in trophoblast fusion in vivo than the experiment with BeWo cells in Figure 7A-D if direct evidence were provided that inhibition of TRPV4 channels reduces cell fusion. It is unclear why cell fusion was not quantified directly from the organoids, and to what extent the markers that were analyzed are indicative of cell fusion. This needs to be addressed.

We thank the comments. The following are our clarifications.

a) As we described in the manuscript, *CSH1* and *CGB* are widely used placental trophoblast fusion marker genes, which are indicative of cell fusion (1-7). *CSH1* and *CGB* encode human placental lactogen (hPL) and hormone-specific β-subunit of human chorionic gonadotropin (βhCG), respectively. Both proteins are specifically synthesized by syncytiotrophoblasts. It is a consensus in the field that the extent of *CSH1* and *CGB* expression directly reports trophoblast fusion efficiency. In fact, commercial pregnancy strip tests detect hCG secreted from syncytiotrophoblasts to report pregnancy. Pharmacological inhibition of TRPV4 by GSK219 dramatically reduces the expression of *CSH1* and *CGB* (Figure 7F). Our result thus strongly supports that inhibiting TRPV4 hinders trophoblast fusion in the human trophoblast organoids.

b) Human trophoblast organoids are newly developed research tools. There is no good way available to quantify fusion index in the 3D system thus far. We tried to use anti-CD46 and anti-CD71 to label the fused cells. Unfortunately, the CD46 and CD71 antibodies also labeled cytotrophoblasts in the human placenta organoids. We also tried to use the membrane labeling method that we developed for quantifying BeWo cell fusion (8, 9). Unfortunately, the membrane marker failed to label the membrane of human placenta organoids. Therefore, we had to use the syncytialization markers *CSH1* and *CGB* to quantify the fusion efficiency in the human placenta organoids. Please see our response a) for details. A reliable imaging method to quantify trophoblast fusion in organoids needs to developed in future studies.

3) Calcium entry into TRPV4 siRNA-treated cells caused by saturating GSK101 (Figure S5A and B) appears to be comparable, if not higher, than in WT BeWO cells stimulated with 0.2 nM GSK101 (Figure 6A and B), yet transient and localized Annexin binding is only observed in the latter but not in the former experiments. This is confusing and needs to be discussed and explained, as it is unclear whether local and transient Annexin binding was looked at in other experiments where it might have also been observed.

We thank the reviewer for the comments.

Although the calcium levels are comparable in siRNA-treated cells with high GSK101 (Figure 1— figure supplement 5A) and in WT cells treated with low GSK101 (Figure 6A-B), there is a fundamental difference between these two conditions. In the siRNA treated cells, TRPV4 expression was largely suppressed, resulting in global reduction of channel density on cell surface. It is likely high GSK101 can hardly induce high calcium in spatially restricted regions to activate local TMEM16F scramblase. However, for siRNA untreated BeWo cells, TRPV4 expression and its density in some local domains is intact. Low GSK101 (0.2 nM) is likely still able to trigger local calcium that is high enough to induce transient TMEM16F activation in microdomain, as we shown in Figure 6. We have revised the manuscript accordingly to make this point clear.

4) In most key figures involving fluorescence imaging only one or two cells are displayed. Although quantification is provided for more than one cell, these numbers still seem small compared to the total number of cells that can be imaged on a coverslip of in a well with seeded cells. It would strengthen the manuscript and provide a more solid representation of the findings if images were provided where a population of cells can be observed, all having similar phenotypes to the representative case that is currently shown. Otherwise, it is unclear if only cells showing the most marked phenotypes were analyzed and displayed.

Showing high magnification images in the manuscript is solely for demonstration purpose so that the reader can see more details. We have also included statistics in our resubmission. To address the concern, shown below is a lower magnification image with a population of cells upon GSK101 stimulation. In this figure, 20 nM GSK101 triggers ca^2+^ influx and PS exposure (labeled by AnV) in all WT BeWo cells but fails to trigger calcium increase in TRPV4 siRNA2 treated BeWo cells.

5) In the response to reviewers and in the manuscript text it is implied that calcium sensitivity of TMEM16F channels is underestimated in whole-cell recordings because that experimental configuration is disruptive to cellular conditions. On the other hand, results from excised patches, which are even more disruptive to the natural conditions of cells, appear to yield higher sensitivity to calcium. Furthermore, the highest sensitivity is observed with calcium imaging, whereas pointed out in the first review, the affinity of the sensor for calcium is a determinant of what is observed. Because identifying a calcium source for TMEM16F channels that enables their activity is central to this manuscript, more clarification is required about the calcium sensitivity of TMEM16F channels. Specifically, it must be clearly stated whether it is speculated that cellular factors contribute to increasing the sensitivity of TMEM16F channels to calcium, or if cellular structures that can get disrupted in the whole cell configuration contribute to the formation of very small calcium microdomains where the concentration of the cation can increase to sufficient levels that activate TMEM16F channels.

We thank the reviewers’ the further comments. As we explained in our previous response letter, the discrepancy between low calcium sensitivity reported in whole cell recording and higher calcium sensitivity measured in excised patches and fluorescence imaging assay for lipid scrambling is an important yet unsolved question in the field. Regardless of the reason for this discrepancy (which is not the task of this manuscript), we believe that it is very likely that the calcium sensitivity estimated by whole cell recording has been dramatically underestimated. Our reasoning is listed below.

(1) Under physiological conditions, 20-200 μm sustained global calcium elevation barely can happen in a healthy cell. (2) Calcium ionophore induced calcium increase in most of lipid scrambling assays used in different labs should be below 20-200 uM. (3) Now we show that 0.2 nM GSK101 can very mildly increase intracellular calcium and transiently induced PS exposure in BeWo cells. As the reviewer pointed out, the calcium dye we used has kd of 1.2 uM. Therefore, calcium increase under this condition should be way lower than 20-200 uM. (4) In addition to our own inside out patch measure of EC50 of 13.6 μm for heterologously expressed TMEM16F current and 5.1 μm for endogenous TMEM16F in megakaryocytes at +60 mV (10), the recent work by the Dutzler lab (11) also showed that purified TMEM16F is activated by ca^2+^ with an EC50 ~1 uM. Therefore, the exact calcium sensitivity for TMEM16F channels and lipid scramblases varies depending on the methods and conditions.

We revised lines 297-310 to further clarify

Reviewer #3:This is a revision of a previously submitted manuscript. The authors addressed satisfactorily most of the first round of critiques with extensive additional experiments. I am mostly satisfied, but I have two residual concerns with what appear to be overclaims by the authors that are detailed below.The authors claim that at low activator concentrations PS exposure is transient. Looking at the representative images shown in Fig6, the transient PS exposure appears to revert in one location, but in the other areas, it persists. Is this just a matter of 'ending the experiment too soon', or does this represent local but irreversible PS exposure? At present, the writing indicates that all PS exposure is reversible but the data does not seem to agree. Quantification of the proportion of temporary vs persistent areas of PS exposure is important.

We appreciate the reviewer’s careful examination. Please look at the pre-GSK101 image at time 0 (Figure 6A and Video 3), the seemingly “persistent” PS positive spots were already there before TRPV4 activation. At 1 o’clock and 3 o’clock directions far away from the cell body, the AnV spots were due to PS positive debris/microparticles in the cell culture. At 6 o’clock direction, there were two PS positive spots, too. They were loosely attached to the cell surface, but not part of the cell. These PS positive debris/microparticles are very common in cell culture when imaging with fluorescence AnV at high resolution. Different from these “persistent” ‘artifacts’, the AnV binding signal at the 9 o’clock direction was not present at time 0. It transiently appeared after GSK101 stimulation, and then rapidly disappeared. This was calcium-induced lipid scrambling. Therefore, our observation using low GSK101 is consistent with the balance between lipid scramblases and flippases under physiological conditions. When local calcium increases, transient TMEM16F activation will lead to temporary collapse of membrane lipid asymmetry in some subcellular localizations. When calcium level fades down, ATP-dependent flippases will restore the lipid asymmetry, resulting in AnV unbinding from cell surface. We believe the reversible, spatiotemporal lipid scrambling activity is a reasonable explanation for understanding TMEM16F’s physiological functions.

To avoid confusing the readers, we have labeled the ‘persistent’ PS positive debris/microparticles in the revised Figure 6 and add the descriptions in the figure legends.

The connection between the interplay of TMEM16F and TRPV4 in placental organoids is quite weak. Downregulation of genes is a poor readout to make the physiological inferences the authors would like to make. Are there any morphological effects on GSK219 -treated organoids, like those seen in the BeWo cells? Is it possible to measure PS exposure in the organoids? The conclusions from this section (both at the end of the results and in the discussion) need to be toned down. Despite claims to the contrary, the authors provide no evidence that inhibition of TRPV4 hinders trophoblast fusion in the organoid model. Further, the downregulation of fusogenic genes might point to a mechanism different from the TMEM16F mediated one that is proposed here. Without direct evidence connecting the data in cell lines to that in the organoid, I believe the manuscript would be strengthened by the removal of this data.

We thank the reviewer for the suggestion. The following are our further clarifications.

a) As we described in the manuscript, *CSH1* and *CGB* are widely used placental trophoblast fusion marker genes, which are indicative of cell fusion (1). *CSH1* and *CGB* encode human placental lactogen (hPL) and hormone-specific β-subunit of human chorionic gonadotropin

(βhCG), respectively. Both proteins are specifically synthesized by syncytiotrophoblasts (7). It is a consensus in the field that the extent of *CSH1* and *CGB* expression directly reports trophoblast fusion efficiency. In fact, commercial pregnancy strip tests detect hCG secreted from syncytiotrophoblasts to report pregnancy. Pharmacological inhibition of TRPV4 by GSK219 dramatically reduces the expression of *CSH1* and *CGB* (Figure 7F). Our result thus strongly supports that inhibiting TRPV4 hinders trophoblast fusion in the human trophoblast organoids.

b) Human trophoblast organoids are a newly developed research tool. There is no good way available to quantify fusion index in the 3D system thus far. We tried to use anti-CD46 and anti-CD71 to label the fused cells. Unfortunately, the CD46 and CD71 antibodies also labeled cytotrophoblasts in the human placenta organoids. We also tried to use the membrane labeling method that we developed for quantifying BeWo cell fusion (8, 9). Unfortunately, the membrane marker failed to label the membrane of human placenta organoids. Therefore, we have to use the syncytialization markers *CSH1* and *CGB* to quantify the fusion efficiency in the human placenta organoids. Please see our response a) for details.

c) Due to the detection limit using fluorescently labeled PS binding proteins including Annexin V, we have not been able to successfully capture PS exposure during cell-cell fusion even in 2D BeWo cell culture. Measuring PS exposure in the organoids will be even more challenging. We wish a more sensitive method to monitor PS exposure can be developed in the near future so that we can detect PS exposure during fusion in real time.

d) *SYN1* is the most important fusogene for trophoblast fusion. However, we did not observe *SYN1* downregulation as the reviewer suspected. Instead, we even observed slight upregulation of *SYN1* expression in GSK219-treated organoids (Figure 7F). This paradoxical finding further supports our previous speculation that TMEM16F-mediated PS exposure indeed plays a critical role in controlling trophoblast fusion, likely independent of the syncytin fusogenes (9). Our current study thus adds new insights in understanding trophoblast fusion. Detailed mechanism needs to be further investigated.

Based on these points, we believe that keeping the human trophoblast organoid results seems appropriate. We hope the reviewer would agree.

References:

1. K. De Clercq *et al.*, Mapping the expression of transient receptor potential channels across murine placental development. *Cell Mol Life Sci* 10.1007/s00018-021-03837-3 (2021).

2. Y. Suzuki *et al.*, Calcium channel TRPV6 is involved in murine maternal-fetal calcium transport. *J Bone Miner Res* 23, 1249-1256 (2008).

3. C. Fecher-Trost *et al.*, Maternal Transient Receptor Potential Vanilloid 6 (Trpv6) Is Involved In Offspring Bone Development. *J Bone Miner Res* 34, 699-710 (2019).

4. R. M. Roberts, J. A. Green, L. C. Schulz, The evolution of the placenta. *Reproduction* 152, R179-189 (2016).

5. L. Yang, C. Megli, C. B. Coyne, Innate immune signaling in trophoblast and decidua organoids defines differential antiviral defenses at the maternal-fetal interface. *bioRxiv* 10.1101/2021.03.29.437467, 2021.2003.2029.437467 (2021).

6. J. Suzuki, M. Umeda, P. J. Sims, S. Nagata, Calcium-dependent phospholipid scrambling by TMEM16F. *Nature* 468, 834-U135 (2010).

7. T. Hirama *et al.*, Membrane curvature induced by proximity of anionic phospholipids can initiate endocytosis. *Nat Commun* 8, 1393 (2017).

8. P. Liang, H. Yang, Molecular underpinning of intracellular pH regulation on TMEM16F. *J Gen Physiol* 153 (2021).

9. K. Yu *et al.*, Identification of a lipid scrambling domain in ANO6/TMEM16F. *eLife* 4 (2015).

10. H. Yang *et al.*, TMEM16F forms a ca^2+^-activated cation channel required for lipid scrambling in platelets during blood coagulation. *Cell* 151, 111-122 (2012).

11. T. W. Han *et al.*, Chemically induced vesiculation as a platform for studying TMEM16F activity. *P Natl Acad Sci USA* 116, 1309-1318 (2019).

12. B. Fakler, J. P. Adelman, Control of KCa Channels by Calcium Nano/Microdomains. *Neuron* 59, 873-881 (2008).

13. E. Eggermann, I. Bucurenciu, S. P. Goswami, P. Jonas, Nanodomain coupling between Ca(2)(+) channels and sensors of exocytosis at fast mammalian synapses. *Nat Rev Neurosci* 13, 7-21 (2011).

14. W. Ye *et al.*, Phosphatidylinositol-(4, 5)-bisphosphate regulates calcium gating of small-conductance cation channel TMEM16F. *Proceedings of the National Academy of Sciences of the United States of America* 115, E1667-E1674 (2018).

15. J. A. Filosa, X. Yao, G. Rath, TRPV4 and the regulation of vascular tone. *J Cardiovasc Pharmacol* 61, 113-119 (2013).

16. D. E. Clapham, D. Julius, C. Montell, G. Schultz, International Union of Pharmacology. XLIX. Nomenclature and structure-function relationships of transient receptor potential channels. *Pharmacol Rev* 57, 427-450 (2005).

17. T. Voets *et al.*, Molecular determinants of permeation through the cation channel TRPV4. *Journal of Biological Chemistry* 277, 33704-33710 (2002).

18. S. Feng *et al.*, Cryo-EM Studies of TMEM16F Calcium-Activated Ion Channel Suggest Features Important for Lipid Scrambling. *Cell Rep* 28, 567-579 e564 (2019).

19. S. Grubb *et al.*, TMEM16F (Anoctamin 6), an anion channel of delayed Ca(2+) activation. *J Gen Physiol* 141, 585-600 (2013).

20. T. Le *et al.*, An inner activation gate controls TMEM16F phospholipid scrambling. *Nat Commun* 10, 1846 (2019).

21. D. M. Nguyen, L. S. Chen, W. P. Yu, T. Y. Chen, Comparison of ion transport determinants between a TMEM16 chloride channel and phospholipid scramblase. *J Gen Physiol* 151, 518-531 (2019).

22. P. Scudieri *et al.*, Ion channel and lipid scramblase activity associated with expression of TMEM16F/ANO6 isoforms. *J Physiol* 593, 3829-3848 (2015).

23. T. Shimizu *et al.*, TMEM16F is a component of a ca^2+^-activated Cl^-^ channel but not a volume-sensitive outwardly rectifying Cl^-^ channel. *Am J Physiol Cell Physiol* 304, C748-759 (2013).

24. H. Yang *et al.*, TMEM16F forms a ca^2+^-activated cation channel required for lipid scrambling in platelets during blood coagulation. *Cell* 151, 111-122 (2012).

25. W. Ye, T. W. Han, M. He, Y. N. Jan, L. Y. Jan, Dynamic change of electrostatic field in TMEM16F permeation pathway shifts its ion selectivity. *ELife* 8 (2019).

26. S. Stabilini, A. Menini, S. Pifferi, Anion and Cation Permeability of the Mouse TMEM16F Calcium-Activated Channel. *Int J Mol Sci* 22 (2021).

27. S. Grubb *et al.*, TMEM16F (Anoctamin 6), an anion channel of delayed ca^2+^ activation. *J Gen Physiol* 141, 585-600 (2013).

28. H. Lin, J. Roh, J. H. Woo, S. J. Kim, J. H. Nam, TMEM16F/ANO6, a ca^2+^-activated anion channel, is negatively regulated by the actin cytoskeleton and intracellular MgATP. *Biochemical and Biophysical Research Communications* 503, 2348-2354 (2018).

29. W. K. Pollock, T. J. Rink, Thrombin and Ionomycin Can Raise Platelet Cytosolic Ca-2+ to ΜM Levels by Discharge of Internal Ca-2+ Stores – Studies Using Fura-2. *Biochemical and Biophysical Research Communications* 139, 308-314 (1986).

30. T. Le, S. C. Le, Y. Zhang, P. Liang, H. Yang, Evidence that polyphenols do not inhibit the phospholipid scramblase TMEM16F. *Journal of Biological Chemistry* 295, 12537-12544 (2020).

31. B. Fakler, J. P. Adelman, Control of K(Ca) channels by calcium nano/microdomains. *Neuron* 59, 873-881 (2008).

32. S. Loukin, X. Zhou, Z. Su, Y. Saimi, C. Kung, Wild-type and brachyolmia-causing mutant TRPV4 channels respond directly to stretch force. *J Biol Chem* 285, 27176-27181 (2010).

33. S. C. Le, P. Liang, A. J. Lowry, H. Yang, Gating and Regulatory Mechanisms of TMEM16 Ion Channels and Scramblases. *Frontiers in Physiology* 12 (2021).

34. S. Mi *et al.*, Syncytin is a captive retroviral envelope protein involved in human placental morphogenesis. *Nature* 403, 785-789 (2000).

35. M. Delidaki, M. Gu, A. Hein, M. Vatish, D. K. Grammatopoulos, Interplay of cAMP and MAPK pathways in hCG secretion and fusogenic gene expression in a trophoblast cell line. *Mol Cell Endocrinol* 332, 213-220 (2011).

36. Y. Zhang *et al.*, TMEM16F phospholipid scramblase mediates trophoblast fusion and placental development. *Sci Adv* 6, eaba0310 (2020).

37. M. Karlsson *et al.*, A single-cell type transcriptomics map of human tissues. *Sci Adv* 7 (2021).

38. E. Sjostedt *et al.*, An atlas of the protein-coding genes in the human, pig, and mouse brain. *Science* 367 (2020).

39. M. Uhlen *et al.*, A pathology atlas of the human cancer transcriptome. *Science* 357 (2017).

40. M. Uhlen *et al.*, A genome-wide transcriptomic analysis of protein-coding genes in human blood cells. *Science* 366 (2019).

41. C. Azar *et al.*, RNA-Seq identifies genes whose proteins are transformative in the differentiation of cytotrophoblast to syncytiotrophoblast, in human primary villous and BeWo trophoblasts. *Sci Rep* 8, 5142 (2018).

42. X. Zhang, M. Pavlicev, H. N. Jones, L. J. Muglia, Eutherian-Specific Gene TRIML2 Attenuates Inflammation in the Evolution of Placentation. *Mol Biol Evol* 37, 507-523 (2020).

43. R. Moreau, A. Hamel, G. Daoud, L. Simoneau, J. Lafond, Expression of calcium channels along the differentiation of cultured trophoblast cells from human term placenta. *Biol Reprod* 67, 1473-1479 (2002).

44. Y. Suzuki *et al.*, TRPV6 Variants Interfere with Maternal-Fetal Calcium Transport through the Placenta and Cause Transient Neonatal Hyperparathyroidism. *Am J Hum Genet* 102, 1104-1114 (2018).

45. A. K. Singh, L. L. McGoldrick, E. C. Twomey, A. I. Sobolevsky, Mechanism of calmodulin inactivation of the calcium-selective TRP channel TRPV6. *Science Advances* 4 (2018).

46. C. Fecher-Trost, U. Wissenbach, P. Weissgerber, TRPV6: From identification to function. *Cell Calcium* 67, 116-122 (2017).

47. M. K. C. van Goor, J. G. J. Hoenderop, J. van der Wijst, TRP channels in calcium homeostasis: from hormonal control to structure-function relationship of TRPV5 and TRPV6. *Bba-Mol Cell Res* 1864, 883-893 (2017).

48. Y. Zhang, H. Yang, A simple and robust fluorescent labeling method to quantify trophoblast fusion. *Placenta* 77, 16-18 (2019).